# Asynchrony between virus diversity and antibody selection limits influenza virus evolution

**Dylan H Morris[1]\*, Velislava N Petrova[2], Fernando W Rossine[1], Edyth Parker[3,4], Bryan T Grenfell[1,5], Richard A Neher[6], Simon A Levin[1], Colin A Russell[4]\***

[1]Department of Ecology & Evolutionary Biology, Princeton University, Princeton, United States; [2]Department of Human Genetics, Wellcome Trust Sanger Institute, Cambridge, United Kingdom; [3]Department of Veterinary Medicine, University of Cambridge, Cambridge, United Kingdom; [4]Department of Medical Microbiology, Academic Medical Center, University of Amsterdam, Amsterdam, Netherlands; [5]Fogarty International Center, National Institutes of Health, Bethesda, United States; [6]Biozentrum, University of Basel, Basel, Switzerland

**Abstract** Seasonal influenza viruses create a persistent global disease burden by evolving to escape immunity induced by prior infections and vaccinations. New antigenic variants have a substantial selective advantage at the population level, but these variants are rarely selected within-host, even in previously immune individuals. Using a mathematical model, we show that the temporal asynchrony between within-host virus exponential growth and antibody-mediated selection could limit within-host antigenic evolution. If selection for new antigenic variants acts principally at the point of initial virus inoculation, where small virus populations encounter well-matched mucosal antibodies in previously-infected individuals, there can exist protection against reinfection that does not regularly produce observable new antigenic variants within individual infected hosts. Our results provide a theoretical explanation for how virus antigenic evolution can be highly selective at the global level but nearly neutral within-host. They also suggest new avenues for improving influenza control.

**\*For correspondence:**
dylan@dylanhmorris.com (DHM);
c.a.russell@amc.uva.nl (CAR)

## Introduction

Antibody-mediated immunity exerts evolutionary selection pressure on the antigenic phenotype of seasonal influenza viruses (*Hensley et al., 2009*; *Archetti and Horsfall, 1950*). Influenza virus infections and vaccinations induce neutralizing antibodies that can prevent reinfection with previously encountered virus antigenic variants, but such reinfections nonetheless occur (*Clements et al., 1986*; *Memoli et al., 2020*; *Javaid et al., 2020*). At the human population level, accumulation of antibody-mediated immunity creates selection pressure favoring antigenic novelty. Circulating antigenic variants typically go extinct rapidly following the population-level emergence of a new antigenic variant, at least for A/H3N2 viruses (*Smith et al., 2004*).

New antigenic variants like those that result in antigenic cluster transitions (*Smith et al., 2004*) and warrant updating the composition of seasonal influenza virus vaccines are likely to be produced in every infected host. Seasonal influenza viruses have high polymerase error rates (on the order of $10^{-5}$ mutations/nucleotide/replication [*Nobusawa and Sato, 2006*]), reach large within-host virus population sizes (as many as $10^{10}$ virions [*Perelson et al., 2012*]), and can be altered antigenically by single amino acid substitutions in the hemagglutinin (HA) protein (*Koel et al., 2013*; *Linderman et al., 2014*).

In the absence of antibody-mediated selection pressure, de novo generated antigenic variants should constitute a tiny minority of the total within-host virus population. Such minority variants are unlikely to be transmitted onward or detected with current next-generation sequencing (NGS) methods. But selection pressure imposed by the antibody-mediated immune response in previously exposed individuals could promote these variants to sufficiently high frequencies to make them easily transmissible and NGS detectable. The potential for antibody-mediated antigenic selection can be readily observed in infections of vaccinated mice (*Hensley et al., 2009*) and in virus passage in eggs in the presence of immune sera (*Davis et al., 2018*).

Surprisingly, new antigenic variants are rarely observed in human seasonal influenza virus infections, even in recently infected or vaccinated hosts (*Debbink et al., 2017*; *Dinis et al., 2016*; *McCrone et al., 2018*; *Sobel Leonard et al., 2016*; *Han et al., 2019*; *Valesano et al., 2019*; *Javaid et al., 2020*; *Figure 1A,B*). These observations contradict existing models of within-host influenza virus evolution (*Luo et al., 2012*; *Volkov et al., 2010*) and pathogen immune escape generally (*Kennedy and Read, 2017*), which model strong within-host antibody selection from the beginning of infection and therefore predict that new antigenic variants will be at consensus or fixation in detectable reinfections of previously immune hosts. This raises a fundamental dilemma. If within-host antibody selection is strong, why do new antigenic variants appear so rarely? If this selection is weak, how can there be protection against reinfection and resulting strong population-level selection?

We hypothesized that influenza virus antigenic evolution is limited by asynchrony between virus diversity and antibody-mediated selection pressure. Antibody immunity at the point of transmission in previously-infected or vaccinated individuals should reduce the initial probability of reinfection (*Le Sage et al., 2020*); secretory IgA antibodies on mucosal surfaces (sIgA) are likely to play a large role (*Wang et al., 2017*, see Appendix Section A2). But if viruses are not blocked at the point of transmission and successfully infect host cells, an antibody-mediated recall response takes multiple days to mount (*Coro et al., 2006*; *Lam and Baumgarth, 2019*, see detailed review in Appendix Section A2). Virus titer—and virus shedding (*Lau et al., 2010*)—may peak before the production of new antibodies has even begun, leaving limited opportunity for within-host immune selection. If immune selection pressure is strong at the point of transmission but weak during virus exponential growth, new antigenic variants could spread rapidly at the population level without being readily selected during the timecourse of a single infection. Moreover, prior work has established tight population bottlenecks at the point of influenza virus transmission (*McCrone et al., 2018*; *Xue and Bloom, 2019*). With a tight transmission bottleneck and weak selection during virus exponential growth, antigenic diversity generated during any particular infection will most likely be lost, slowing the accumulation of population-level antigenic diversity.

We used a mathematical model to investigate our hypothesis that realistically-timed antibody-mediated immune dynamics slow within-host antigenic evolution. We found three modeling results: (1) antibody neutralization at the point of inoculation can protect experienced hosts against reinfection and explain new antigenic variants' population-level selective advantage. (2) If successful reinfection occurs, the delay between the start of virus replication and the mounting of a recall antibody response renders within-host antigenic evolution nearly neutral, even in experienced hosts. (3) It is therefore reasonable that substantial population immunity may need to accumulate before new antigenic variants are likely to observed in large numbers at the population level, whereas effective within-host selection would predict that they should be readily observable even before they proliferate.

Our modeling results suggest a plausible mechanism that can explain otherwise poorly-reconciled empirical patterns, and should motivate further experimental investigation of the mechanisms of immune protection and natural selection on influenza virus antigenic phenotypes at the point of transmission.

## Model overview

Our model reflects the following biological realities: (1) Seasonal influenza virus infections of otherwise healthy individuals typically last 5–7 days (*Suess et al., 2012*); (2) In influenza virus-naive individuals, it can take up to 7 days for anti-influenza virus antibodies to start being produced (*Wrammert et al., 2008*), effectively resulting in no selection (*Figure 1A*); (3) In previously infected ('experienced') individuals, sIgA antibodies can neutralize inoculated virions before they can infect

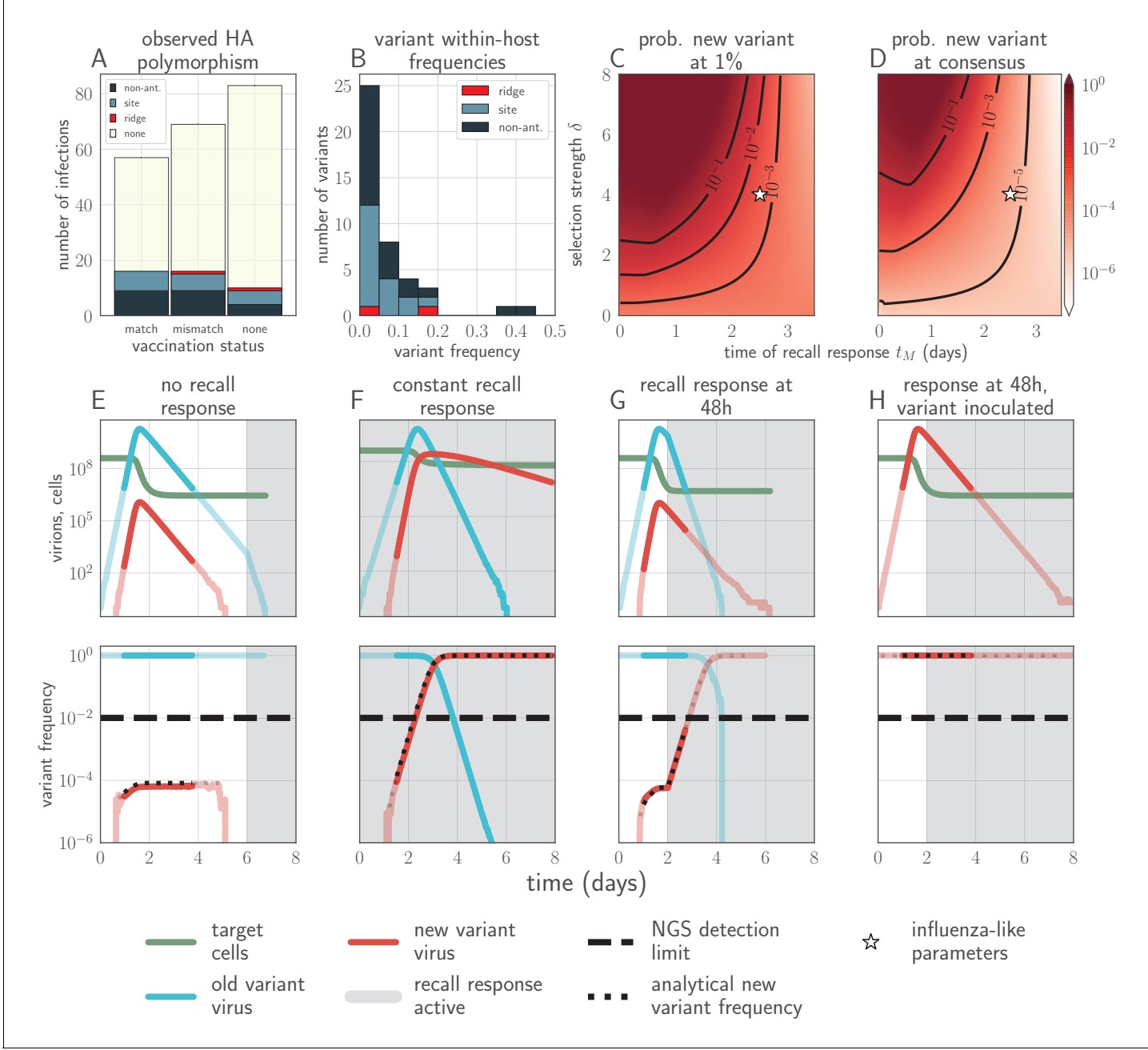

**Figure 1.** Empirical within-host influenza virus variant frequencies and model within-host evolutionary dynamics. (A, B) meta-analysis of A/H3N2 viruses from next-generation sequencing studies of naturally-infected individuals (*Debbink et al., 2017*; *McCrone et al., 2018*). (A) Fraction of infections with one or more observed amino acid polymorphisms in the hemagglutinin (HA) protein, stratified by likelihood of affecting antigenicity: infections with a substitution in the 'antigenic ridge' of 7 key amino acid positions found by *Koel et al., 2013* in red, infections with a substitution in a classically-defined 'antigenic site', (*Wiley et al., 1981*) in blue, infections with HA substitutions only in non-antigenic regions in gray, infections with no HA substitutions in cream. Infections grouped by whether individuals had been (left) vaccinated in a year that the vaccine matched the circulating strain, (center) vaccinated in a year that the vaccine did not match the circulating strain, or (right) not vaccinated. (B) Distribution of plotted polymorphic sites from (A) by within-host frequency of the minor variant. (C, D) heatmaps showing model probability of new antigenic variant selection to the NGS detection threshold of 1% (C) and to 50% (D) by 3 days post infection given the strength of immune selection $\delta$, the antibody response time $t_M$ and a founding population composed of old variant virions. Probabilities calculated from *Equation 27* in the Materials and methods. Calculated with $c_w = 1, c_m = 0$, but for $t_M > 1$, replication selection probabilities are approximately equal for all $c_w, c_m, k$ trios that yield a given $\delta$ (see Materials and methods). Star denotes a plausible influenza-like parameter regime: 25% escape from sterilizing-strength immunity ($c_w = 1, c_m = 0.75, k = 20$) with a recall response at 2.5 days post infection. Black lines are probability contours. (E–H) example model trajectories. Upper row: absolute counts of virions and target cells. Lower row: variant frequencies for old antigenic variant (blue) and new variant (red). Dashed line shows 1% frequency, the detection limit of NGS. Dotted line

*Figure 1 continued on next page*

Figure 1 continued

shows an analytical prediction for new variant frequency according to *Equations 15 and 16* (see Materials and methods). Model scenarios: (**E**) naive; (**F**) experienced with $t_M = 0$; (**G**) experienced with $t_M = 2$; (**H**) experienced with $t_M = 2$ and new antigenic variant virion inoculated. Lines faded when infection is below 5% transmission probability—approximately $10^7$ virions with default parameters. All parameters as in *Table 1* unless otherwise stated.

host cells; (*Wang et al., 2017*) (4) However, if an inoculated virion manages to cause an infection in an experienced individual, it takes 2–5 days for the infected host to mount a recall adaptive immune response, including producing new antibodies (*Coro et al., 2006*; *Zuccarino-Catania et al., 2014*) (see Appendix Section A2 for further discussion of motivating immunology). Importantly, this contrasts with previous within-host models of virus evolution, which have assumed that antibody-mediated neutralization of virions during virus replication is strong from the point of inoculation onward and is the mechanism of protection against reinfection (*Luo et al., 2012*; *Volkov et al., 2010*). It also reflects new animal model evidence of sterilizing antibody immunity (*Le Sage et al., 2020*). We discuss existing models and hypotheses for the rarity of population-level influenza antigenic variation in Appendix Section A7.

Our model can be parameterized to reflect different hypothesized immune mechanisms, different host immune statuses, and different durations of infection. In the model, virions $V_i$ of antigenic type $i$ infect target cells $C$, replicate, mutate to a new antigenic type $j$ at a rate $\mu_{ij}$, and decay at a rate $d_v$. We model the innate immune response implicitly as depletion of infectible cells. We model the antibody-mediated immune response as an increase $k$ in the virion decay rate in the presence of well-matched antibodies. To model partial antibody cross reactivity, we scale $k$ by a parameter $c_i \in [0, 1]$; $c_i$ reflects the binding strength of the host's best-matched antibodies to antigenic type $i$. So in the presence of an antibody response, virions $V_i$ of type $i$ decay at a rate $d_v + c_i k$.

The model can accommodate $N_v$ antigenic variants $i = 1, 2, ...N_v$ linked by an arbitrary network of possible substitutions and corresponding mutation probabilities $\mu_{ij}$, but in practice we typically consider two, the new variant $m$ and the old variant $w$, and neglect back-mutation from new variant to old variant ($\mu_{wm} > 0$, but $\mu_{mw} = 0$).

To assess the importance of transmission bottlenecks, initial virus diversity, and sIgA antibody neutralization in virus evolution, we model the point of transmission as a series of stochastic events which may ultimately lead to one of more virions invading cells and initiating an infection. The recipient host is inoculated with a random sample of within-host virus diversity from the transmitting host. In experienced hosts, this inoculum is probabilistically thinned by host antibodies. The founding population that initiates the infection is then randomly sampled from among any remaining virions.

Mathematically, we model the number of inoculated virions as Poisson-distributed with a mean $v$, so if variant $i$ has frequency $f_i$ within the transmitting host, the number of variant $i$ virions inoculated is Poisson-distributed with mean $v f_i$. The virions then encounter antibodies, which we interpret as sIgA but can be understood to be any antigen-specific antibody-mediated protection that precedes cell infection; each virion of variant $i$ is independently neutralized with a probability $\kappa_i$. This probability depends upon the strength of protection against homotypic reinfection $\kappa$ and the sIgA cross immunity between variants $\sigma_{ij}$, ($0 \leq \sigma_{ij} \leq 1$). So if a host with antibodies to variant $j$ is challenged with variant $i$, those virions will be neutralized with a probability $\kappa_i = \kappa \sigma_{ij}$. For simplicity, we assume the same homotypic protection level across all variants and hosts, though in practice there may be variation in the immunogenicity of individual variants and in the strength of responses generated by individual hosts. We typically fix host immune histories to test the effect of host immune history on selective dynamics. When necessary, we can model a novel (non-recall) antibody response to a strain $i$ by designating the host as experienced to $i$ at some time $t_N^i$ post-infection (see Materials and methods).

The model is continuous-time and stochastic: cell infection, virion production, virus mutation, and virion decay are stochastic events that occur at rates governed by the current state of the system, with exponentially distributed (memoryless) waiting times. The system is approximately well-mixed: we track counts of virions and cells without an explicit account of space within the upper respiratory tract. We treat infected and dead or removed cells implicitly.

**Table 1.** Model parameters, default values, and sources/justifications.

| Parameter | Meaning | Units | Value | Source or justification |
|---|---|---|---|---|
| $t_M$ | time post-infection of antibody response in experienced hosts | days | 2 | literature (see review in Appendix Section A2) |
| $t_N^w$ | time post-infection of a novel immune response to the old antigenic variant | days | 6 | literature (see review in Appendix Section A2) |
| $p_C$ | per-capita growth rate of target cells at low density | $\frac{1}{\text{days}}$ | 0 | ignored on the timescale of a single infection |
| $C_{\max}$ | maximum number of target cells | cells | $4 \times 10^8$ | standard in the modeling literature (*Baccam et al., 2006*; *Luo et al., 2012*; *Hadjichrysanthou et al., 2016*) |
| $\mathcal{R}_0$ | within-host basic reproduction number for the virus | unitless | 5 | empirical fits of target cell models (*Hadjichrysanthou et al., 2016*) |
| $r_w$ | average number of infectious virions produced by a cell infected with old antigenic variant virus | virions | 100 | literature (*Frensing et al., 2016*) |
| $r_m$ | average number of infectious virions produced by a cell infected with new antigenic variant virus | virions | 100 | no within-host deleteriousness for new antigenic variants |
| $\mu_{wm}$ | probability of mutation from old variant to new variant | unitless | $0.33 \times 10^{-5}$ | literature (*Nobusawa and Sato, 2006*) |
| $\mu_{mw}$ | probability of mutation from new variant to old variant | unitless | 0 | back-mutation neglected |
| $\beta$ | rate of infectious contact between virions and target cells per cell per virion | $\frac{1}{\text{virions cells days}}$ | calculated | from $\mathcal{R}_0$ |
| $\ell$ | number of target cells lost per infectious contact | cells | 1 | one cell lost per cell infection |
| $d_v$ | exponential decay rate of infectious virions | $\frac{1}{\text{days}}$ | 4 | empirical fits of target cell models (*Hadjichrysanthou et al., 2016*) and modeling literature (*Baccam et al., 2006Luo et al., 2012*) |
| $k$ | additional per-virion neutralization rate in the presence of a well-matched antibody response | $\frac{1}{\text{days}}$ | 6 | varied to test hypotheses |
| $c_w$ | fractional cross reactivity during viral replication between host antibodies and the old antigenic variant | unitless | 0 or 1 | naive or homotypically reinfected hosts |
| $c_m$ | fractional cross reactivity during viral replication between host antibodies and the new antigenic variant | unitless | 0 | full escape variant |
| $\kappa_w$ | probability that an individual old antigenic variant virion inoculated into an experienced host is neutralized in the respiratory tract mucosa | unitless | set from $z_w$ | calculated from *Equation 38* |
| $\kappa_m$ | probability that an individual new antigenic variant virion inoculated into an experienced host is neutralized in the respiratory tract mucosa | unitless | $\sigma \kappa_w$ | reduced relative to $\kappa_w$ by immune escape |
| $\sigma$ | fractional cross immunity at the sIgA bottleneck between old antigenic variant and new antigenic variant | unitless | 0 | full escape variant |
| $v$ | number of virions encountering sIgA | virions | $10 \times b$ | |
| $b$ | size of final/cell infection bottleneck | virions | 1 | NGS studies (*McCrone et al., 2018*; *Xue and Bloom, 2019*) |
| $V_{50}$ | viral load at which there is a fifty percent transmission probability | virions | $10^8$ | chosen to give realistic transmission window (*Tsang et al., 2015*) and based on prior modeling studies (*Russell et al., 2012*) |
| $\theta$ | transmission threshold for threshold model | virions | $10^7$ | chosen to be consistent with $V_{50}$ |

Parameterized as in *Table 1*, the model captures key features of influenza infections: rapid peaks approximately 2 days post-infection and slower declines with clearance approximately a week post-infection, with faster clearance in experienced hosts.

We give a full mathematical description of the model in the Materials and methods.

In addition to analyzing this within-host model, we explored the between-host and population-level implications of these within-host dynamics using a simple transmission chain model and an SIR-like population-level model, which we describe in the Materials and methods.

## Results

### Realistically-timed immune kinetics limit otherwise rapid adaptation during exponential growth

In our model, sufficiently strong antibody neutralization during the virus exponential growth period can potentially stop replication and block onward transmission, but this mechanism of protection results in detectable new antigenic variants in each observable homotypic reinfection, since the infection is terminated rapidly unless it generates a new antigenic variant that substantially escapes antibody neutralization (*Figure 2*).

If there is antibody neutralization throughout virus exponential growth and it is not sufficiently strong to control the infection, this facilitates the establishment of new antigenic variants: variants

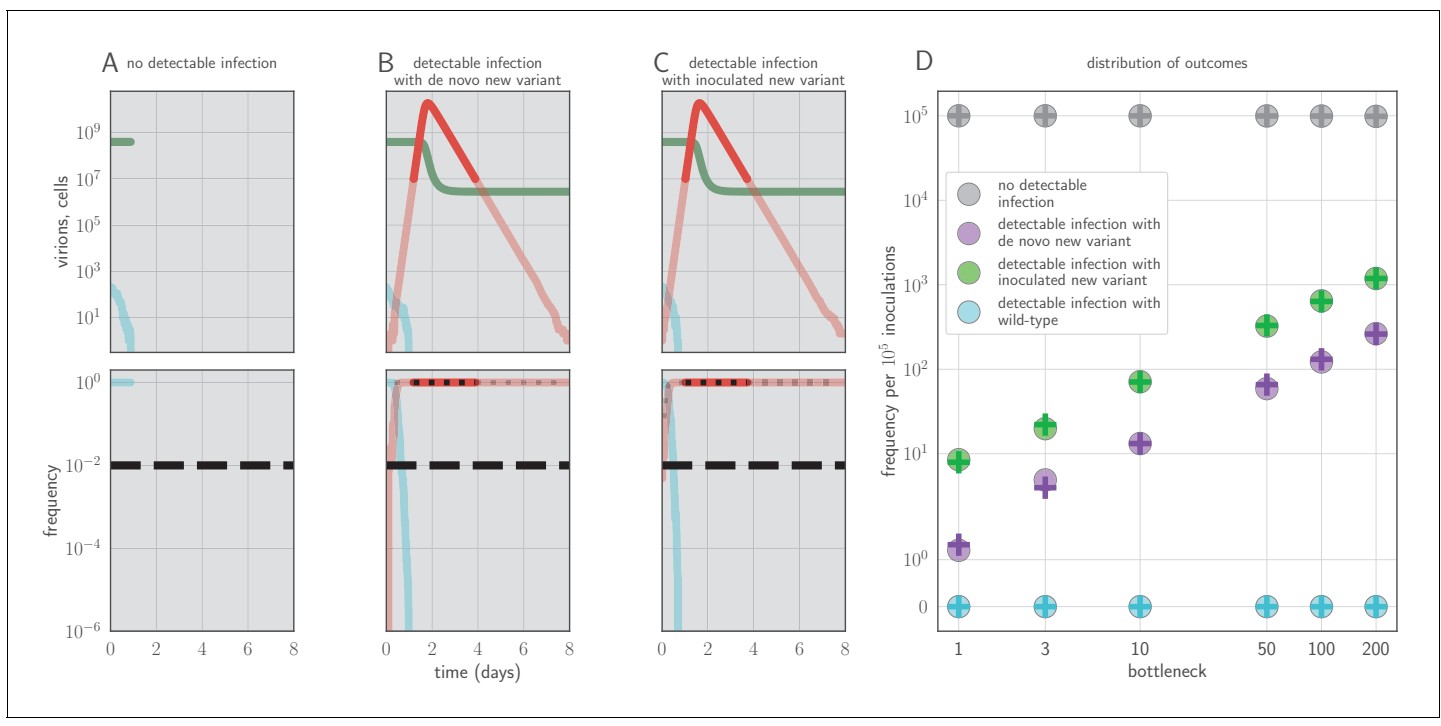

**Figure 2.** Example timecourses and distribution of outcomes when antibody immunity is active from the start of infection and sufficient to prevent detectable reinfection. $t_M = 0$, $k = 20$, yielding $\mathcal{R}^w(t=0)<1$ for the old antigenic variant but $\mathcal{R}^m(t=0)>1$ for the new antigenic variant, where $\mathcal{R}^i(t)$ is the within-host effective reproduction number for variant $i$ at time $t$ (see Materials and methods). No mucosal antibody neutralization ($z_w = z_m = 0$); protection is only via neutralization during replication. Example timecourses from simulations with founding population (bottleneck) $b = 200$. Since neutralization during replication takes the place of mucosal sIgA neutralization, $b$ here should be understood as comparable to the parameter $v$ in models with sIgA neutralization. (A–C) Top panels: absolute abundances of target cells (green), old antigenic variant virions (blue), and new antigenic variant virions (red). Bottom panels: frequencies of virion types. Black dotted line is an analytical prediction for the new antigenic variant frequency given the time of first appearance. Black dashed line is the threshold for NGS detection. (D) Frequencies of no infection, de novo new antigenic variant infection, inoculated new antigenic variant infection, and old antigenic variant infection per $10^5$ inoculations of an immune host by a naive host. Circles are frequencies from simulation runs ($10^6$ runs for bottlenecks 1–10, $10^5$ runs for bottlenecks 50–200). Plus-signs are analytical model predictions for frequencies (see Appendix Section A3.6), with $f_{\text{int}}$ set equal to the average from donor-host stochastic simulations for the given bottleneck. Parameters as in *Table 1* unless otherwise stated.

can be generated de novo and then selected to detectable and easily transmissible frequencies (*Figure 1C,D,F*, sensitivity analysis in *Appendix 1—figure 3*). We term selection on a replicating within-host virus population 'replication selection'. Virus phenotypes that directly affect fitness independent of immune system interactions are likely to be subject to replication selection.

Adding a realistic delay to antibody production of two days post-infection (*Lam and Baumgarth, 2019*) curtails antigenic replication selection. There is no focused antibody-mediated response during the virus exponential growth phase, and so the infection is dominated by old antigenic variant virus (*Figure 1C,D,G*, sensitivity analysis in *Appendix 1—figure 3*). Antigenic variant viruses begin to be selected to higher frequencies late in infection, once a memory response produces high concentrations of cross-reactive antibodies. But by the time this happens in typical infections, both new antigenic variant and old antigenic variant populations have peaked and begun to decline due to innate immunity and depletion of infectible cells, so new antigenic variants remain too rare to be detectable with NGS (*Figure 1G*).

We find that replication selection of antigenic novelty to detectable levels becomes likely only if infections are prolonged, and virus antigenic diversity and antibody selection pressure therefore coincide (*Figure 1G*, see also Figure 7). This can explain existing observations: within-host adaptive antigenic evolution can be seen in prolonged infections of immune-compromised hosts (*Xue et al., 2017*), and prolonged influenza infections show large within-host effective population sizes (*Lumby et al., 2020*).

## Neutralization of virions at the point of transmission provides host protection and population-level selection without rapid within-host adaptation

Adding antibody neutralization of virions at the point of inoculation (e.g. by mucosal sIgA) to our model produces realistic levels of protection against reinfection, and when reinfections do occur, they are overwhelmingly old antigenic variant reinfections. New antigenic variants that arise during these reinfections remain undetectably rare, reproducing observations from natural human infections (*Figures 1A–D,G–H*, *Figure 3B–C*; *Debbink et al., 2017*; *Dinis et al., 2016*; *McCrone et al., 2018*; *Sobel Leonard et al., 2016*). The combination of mucosal sIgA protection and a realistically-timed antibody recall response explains how there can exist immune protection against reinfection—and thus a population-level selective advantage for new antigenic variants—without observable within-host antigenic selection in typical infections of experienced hosts.

## Tight bottlenecks lead to loss of generated diversity and mean new variants reach consensus through founder effects

Regardless of host immune status, an antigenic variant that has been generated de novo within a host must survive a series of population bottlenecks if it is to infect other individuals. To found a new infection, virions must be expelled from a currently infected host (excretion bottleneck), must enter another host (inter-host bottleneck), must escape mucus on the surface of the airway epithelium (mucus bottleneck), must avoid neutralization by sIgA antibodies on mucosal surfaces (sIgA bottleneck), and must infect a cell early enough to form a detectable fraction of the resultant infection (cell infection bottleneck) (*Figure 3A*). The sum of all of these effects is the net bottleneck and typically results in infections being initiated by a single genomic variant (*McCrone et al., 2018*; *Xue and Bloom, 2019*; *Ghafari et al., 2020*). That said, bottlenecks resulting from direct contact transmission may be substantially wider than those associated with respiratory droplet or aerosol transmission (*Varble et al., 2014*) and more human studies are required to quantify these differences.

We find that because antigenic variants appear at very low within-host frequencies when generated de novo and undergo minimal or no replication selection, new antigenic variants most commonly reach detectable levels within hosts through founder effects at the point of inter-host transmission: a low-frequency antigenic variant generated in one host survives the net bottleneck to found the infection of a second host (*Figure 3D*).

Given that influenza bottlenecks are thought to be on the order of a single virion (*McCrone et al., 2018*), any replication-competent mutant that founds an infection should occur at NGS-detectable levels, and likely at consensus. But for the same reason, these founder effects are rare events. The sampling process that produces these founder effects could be a purely neutral. It

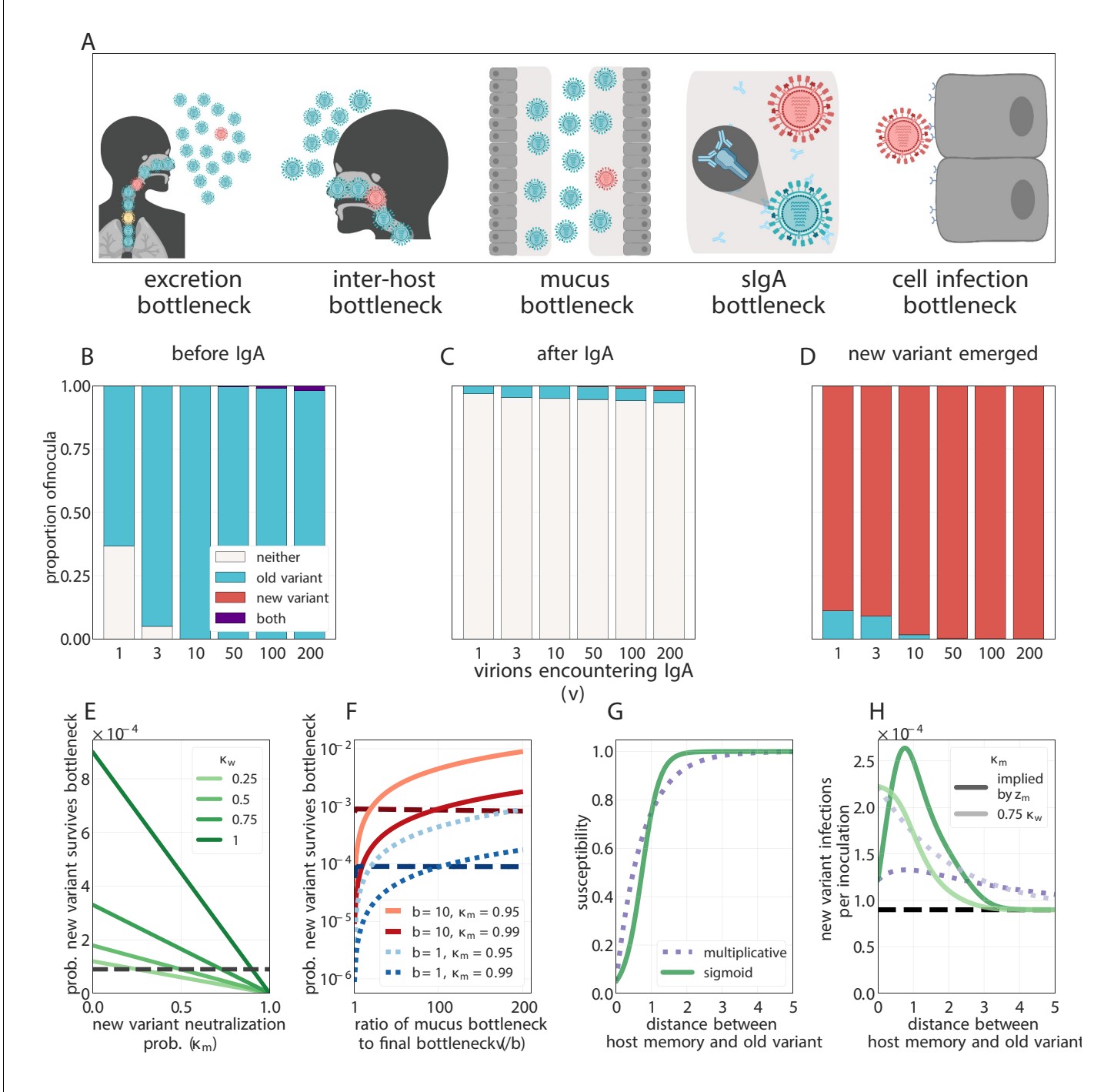

**Figure 3.** Selection for antigenic variants at the point of transmission (inoculation selection). (A) Schematic of bottlenecks faced by both old antigenic variant (blue) and new antigenic variant (other colors) virions at the point of virus transmission. Key parameters for inoculation selection are the mucus bottleneck size $v$—the mean number of virions that encounter sIgA—and the cell infection bottleneck size $b$. (B–D) Effect of sIgA selection at the point of inoculation with $b = 1$. (B, C) Analytical model distribution of virions inoculated into an immune host immediately before (B) and after (C) mucosal neutralization/the sIgA bottleneck. $f_{mt}$ set to mean of stochastic simulations. (D) Distribution of founding virion populations (after the cell infection bottleneck) among individuals who developed detectable new antigenic variant infections in stochastic simulations. (E–F) Analytical model probability that a variant survives the final bottleneck. Dashed horizontal lines indicate probability in naive hosts. $f_{mt} = 9 \times 10^{-5}$ (the approximate mean in stochastic simulations for $b = 1$). (E) Variant probability of surviving all bottlenecks, as a function of old antigenic variant neutralization probability $\kappa_w$ and new antigenic variant mucosal neutralization probability $\kappa_m$. (F) New antigenic variant survival probability as a function of the ratio of $v$ to $b$. (G, H) Effect of host susceptibility model on the appearance of antigenic novelty. Per-inoculation rates of new variants surviving the bottleneck (H) depend on

*Figure 3 continued on next page*

Figure 3 continued

host immune status and on the relationship between virus antigenic phenotype and host susceptibility $(1 - z_w)$ (G). Plotted with $f_{mt} = 9 \times 10^{-5}$, and 25% susceptibility to (75% protection against, $z_1 = 0.75$) a variant one antigenic cluster away from host memory. Unless noted, parameters for all plots as in **Table 1**.

is likely quite close to neutral in a truly naive recipient host who does not possess well-matched antibodies to the inoculated old variant: all inoculated virions, regardless of antigenic phenotype, have an equal chance of becoming part of the new infection's founding population. If there are $v$ virions that compete to found the infection and $b = 1$, then each virion founds the infection with probability $1/v$.

In our model, new antigenic variants therefore survive the transmission bottleneck upon inoculation into a naive host with a probability approximately equal to the donor-host variant frequency $f_{mt}$ times the bottleneck size $b$ (see Materials and methods). New antigenic variant infections of naive hosts should then occur on the order of 1 in $10^5$ or 1 in $10^4$ such infections given biologically plausible parameters (*Figure 3E–H*).

But if different virions have different chances of being neutralized at the point of transmission, the founding process may be selective. Among the virions that encounter antibodies, those that are less likely to be neutralized have a higher than average chance of undergoing stochastic promotion to consensus while those that are more likely to be neutralized have a lower than average chance. A new antigenic variant may then be disproportionately likely to survive the net bottleneck (*Figure 3*, Appendix Section A4.2). We term this potential selection on inoculated diversity 'inoculation selection'. Neutralization at the point of transmission thus not only gives new antigenic variant infections their transmission advantage (population-level selection) but may also increase the rate at which these new antigenic variant infections arise (inoculation selection).

There is some suggestive evidence of differential survival of particular (not necessarily antigenic) influenza genetic variants at the point of transmission from experiments in ferrets (*Wilker et al., 2013*; *Moncla et al., 2016*). But as Lumby and colleagues (*Lumby et al., 2018*) point out in a reanalysis of those experiments, it is difficult empirically to distinguish selection that occurs at the point of transmission from selection that occurs during early replication in the recipient host because of the challenges associated with sampling the small virus populations present at the earliest stages of infection. Here we define inoculation selection as selection on the bottlenecked virus population that establishes infection in the recipient host before any virus replication has taken place in that host.

## Inoculation selection depends on degree of founding competition and degree of immune escape

The strength of inoculation selection depends on the ratio of the number of virions that compete to found an infection in the absence of well-matched sIgA antibodies (the mucus bottleneck size $v$) to the number of virions that actually found an infection (the final cell infection bottleneck size $b$). The larger this $v/b$ ratio is, the more inoculation selection in experienced hosts facilitates the survival of new antigenic variants (*Figure 3F*, *Appendix 1—figure 1*).

When new antigenic variant immune escape is incomplete due to partial cross-reactivity with previous antigenic variants, increased antibody neutralization is a double-edged sword for new antigenic variant virions. Competition to found the infection from old antigenic variant virions is reduced, but the new antigenic variant is itself at greater risk of being neutralized. The impact of inoculation selection therefore depends on the degree of similarity between previously encountered viruses and the new antigenic variant. An experienced recipient host could facilitate the survival of large-effect antigenic variants (like those seen at antigenic cluster transitions [*Smith et al., 2004*]) while impeding the survival of variants that provide less substantial immune escape (*Figure 3E*, *Appendix 1—figure 1*).

Inoculation selection is limited by the low frequency $f_{mt}$ of new antigenic variants in transmitting donor hosts (due to weak replication selection), the potentially small mucus bottleneck size $v$, and the fact that some hosts previously infected with the old variant or similar antigenic variants might not possess well-matched antibodies due to original antigenic sin (*Davenport and Hennessy, 1956*), antigenic seniority (*Lessler et al., 2012*), immune backboosting (*Fonville et al., 2014*), or

other sources of individual-specific variation in antibody production (*Lee et al., 2019*). These factors combined make selection and onward transmission of new variants rare.

## Immune hosts can facilitate the appearance of new variants without producing rapid diversification

Onward transmission of new variants can be facilitated by natural selection—replication selection, inoculation selection, or both. The degree of facilitation depends principally on four quantities: (1) $\delta\tau$, the product of the replication selection fitness difference $\delta = k(c_w - c_m)$ and the time under replication selection $\tau$. This determines the degree to which the new variant is promoted by replication selection prior to transmission (increasing $f_{mt}$). (2) $\kappa_w$, the sIgA neutralization probability for the old variant. This must be large enough to reduce competition for the final bottleneck. (3) $v/b$, the ratio of the number of virions that encounter sIgA $v$ to the cell infection (final) bottleneck size $b$. This determines the degree of competition to found the infection, and thus sets the maximum potential strength of inoculation selection when $\kappa_w$ is large: a $\frac{v}{b}$-fold improvement over drift for small $b$. (4) $1 - \kappa_m$, how likely the new variant is to avoid neutralization at the sIgA bottleneck; this scales down the maximum inoculation selective strength set by $v/b$. Inoculation selection can impede new variant survival relative to drift if $1 - \kappa_m$ is small enough.

When $\delta\tau$ is small and $\kappa_w$, $v/b$, and $1 - \kappa_m$ are large, new variant survival is most facilitated by inoculation selection. When the opposite is true, replication selection is most important. And there are parameter regimes in which both replication and inoculation selection provide a substantial improvement over drift (*Figure 4*, see Appendix Section A4.6 for mathematical intuition for these results).

At realistic parameter values and assuming all individuals develop well-matched antibodies to previously encountered antigenic variants, only ~1 to 2 in $10^4$ inoculations of an experienced host results in a new antigenic variant surviving the bottleneck (*Figure 4E,H*, *Figure 4*). This rate is likely to be an overestimate due to the factors mentioned above. Moreover, it is only about 2- to 3-fold higher than the rate of bottleneck survival in naive hosts, where new antigenic variant infections should occasionally occur via neutral stochastic founder effects. In short, even in the presence of experienced hosts, antigenic selection is inefficient and most generated antigenic diversity is lost at the point of transmission. Because of these inefficiencies, new antigenic variants can be generated in every infected host without producing explosive antigenic diversification at the population level.

## Inoculation selection produces realistically noisy between-host evolution

To investigate the between-host consequences of adaptation given weak replication selection, tight bottlenecks, and possible inoculation selection, we simulated transmission chains according to our within-host model, allowing the virus to evolve in a 1-dimensional antigenic space (*Smith et al., 2004*; *Bedford et al., 2012*) until a generated antigenic mutant became the majority within-host variant. When all hosts in a model transmission chain are naive, antigenic evolution is non-directional and recapitulates the distribution of within-host mutations (*Figure 5A*). When antigenic selection is constant throughout infection and even a moderate fraction of hosts are experienced, antigenic evolution is unrealistically adaptive: the virus evolves directly away from existing immunity and large-effect antigenic changes are observed frequently (*Figure 5B,C*). When the model incorporates both mucosal antibodies and realistically-timed recall responses, major antigenic variants appear only rarely and the overall distribution of emerged variants better mimics empirical observations (*Figure 5D*)—most notably, the phenomenon of quasi-neutral diversification within an antigenic cluster seen in Figures 1 and 2 of *Smith et al., 2004*. A simple analytical model (see Methods) in which generated antigenic mutants fix according to their replication and inoculation-selective advantages also displays this behavior (*Figure 5B–H*).

In particular, we note that whereas an immediate recall response would predict strong near-constant directed evolution of virus antigenic phenotypes away from existing immunity (*Figure 5B,C*), a realistically-timed recall response predicts that small-effect, drift-like antigenic substitutions will be observed. Even substitutions that move a virus 'backward' in antigenic space— so that it is more readily neutralized by existing antibodies than the ancestral variant—can be observed thanks to the large role of stochasticity at the point of transmission. That said, there is a slight bias favoring forward substitutions, especially those of sufficiently large effect to create a substantial selective

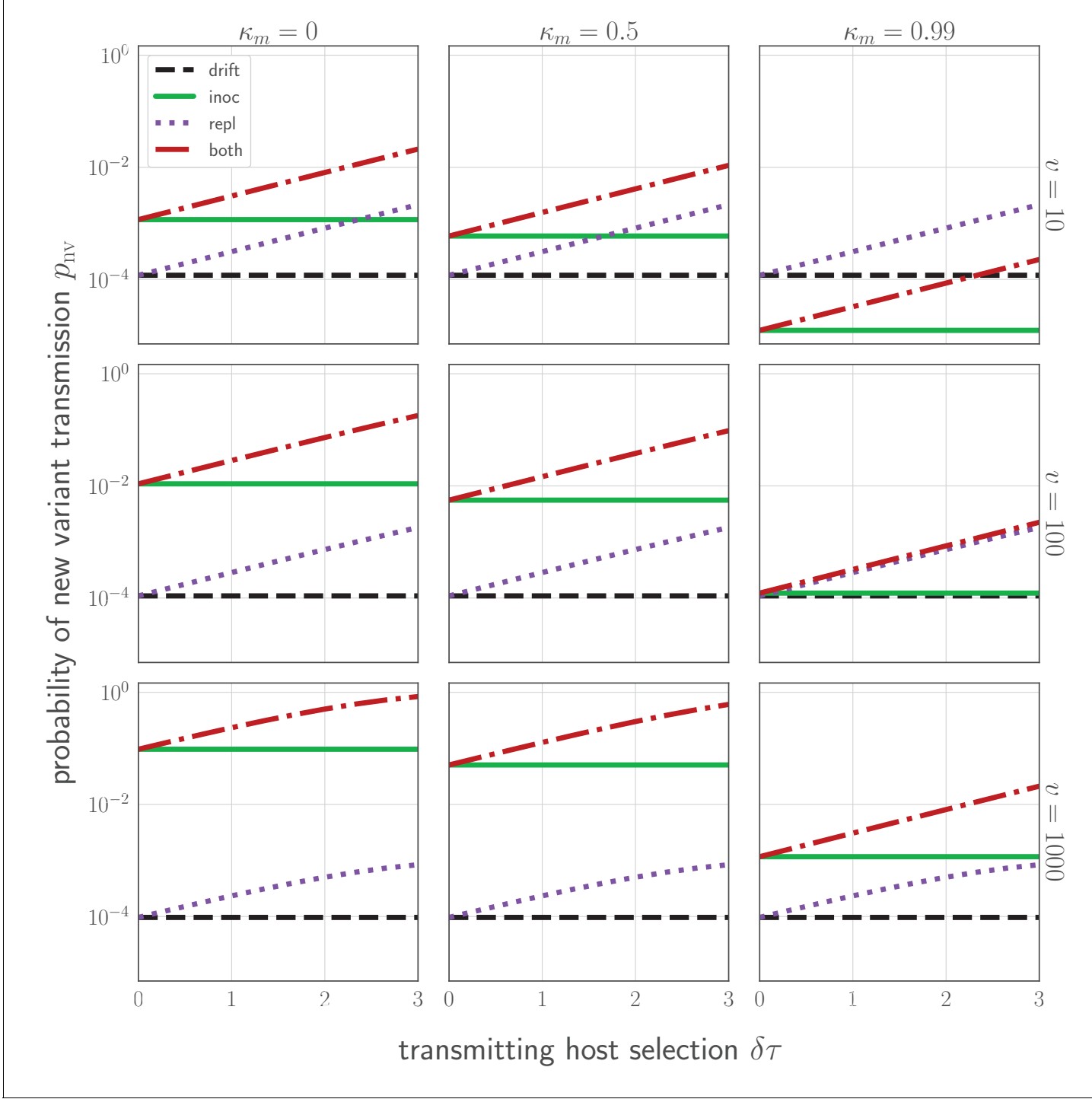

**Figure 4.** Probability $p_{nv}$ of a new variant surviving the transmission bottleneck as a function of donor-host replication selection and recipient host inoculation selection. Calculated according to *Equation 43*, and plotted as a function of degree of replication selection in the donor host $\delta\tau$, the product of the selection strength $\delta$ and the time duration $\tau = \max\{0, t_t - t_M\}$ between the onset of the antibody response at $t_M$ and the transmission event at $t_t$. Black dashed line: neutral (drift) expectation, where $\delta = 0$ in the donor host and the recipient host does not neutralize either the old or the new variant at the point of transmission ($\kappa_w = \kappa_m = 0$). Purple dotted line: replication selection only: $\delta\tau$ as given in the donor host, but a naive recipient host. Green solid line: inoculation selection only: $\delta = 0$ in the donor host, but a recipient host with well-matched antibodies to the old variant ($\kappa_w = 1$), with varying degrees of immune escape ($\kappa_m$ as given in the columns). Red dot-dashed line: combination of both replication selection in the donor host as before and inoculation selection in the recipient host as before. Plotted with a final bottleneck of $b = 1$ and a mean of $v$ virions encountering sIgA as given in the rows. Parameters as in *Table 1* unless otherwise noted.

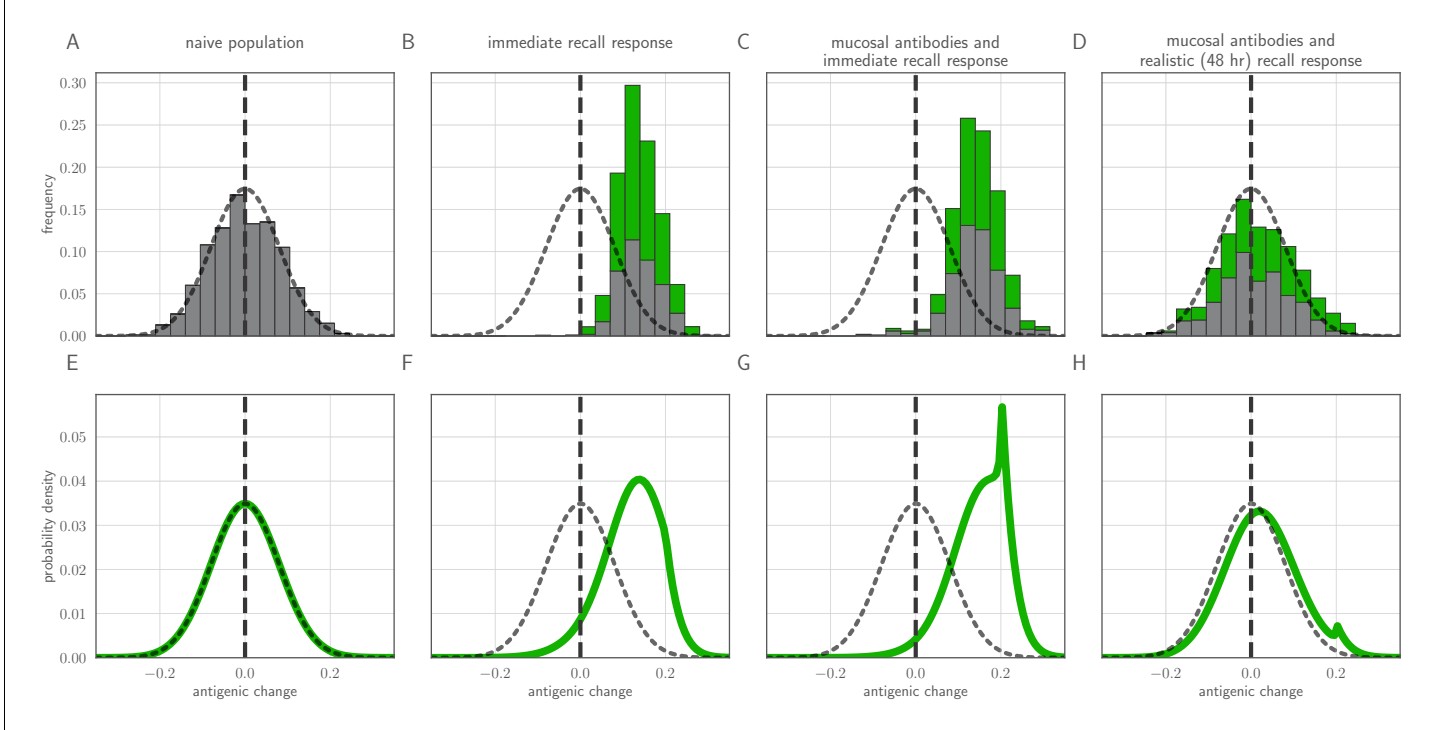

**Figure 5.** Distribution of mutant effects given replication and inoculation selection. Distribution of antigenic changes along 1000 simulated transmission chains (A–D) and from an analytical model (E–H). In (A,E) all naive hosts, in other panels a mix of naive hosts and experienced hosts. Antigenic phenotypes are numbers in a 1-dimensional antigenic space and govern both sIgA cross immunity $\sigma$ and replication cross immunity $c$. A distance of $\geq 1$ corresponds to no cross immunity between phenotypes and a distance of 0 to complete cross immunity. Gray line gives the shape of Gaussian within-host mutation kernel. Histograms show frequency distribution of observed antigenic change events and indicate whether the change took place in a naive (gray) or experienced (green) host. In (B–D) distribution of host immune histories is 20% of individuals previously exposed to phenotype −0.8, 20% to phenotype −0.5, 20% to phenotype 0 and the remaining 40% of hosts naive. In (E), naive hosts inoculate naive hosts. In (F–H) hosts with history −0.8 inoculate hosts with history −0.8. Initial variant has phenotype 0 in all sub-panels. Model parameters as in *Table 1*, except $k = 25$. Spikes in densities occur at 0.2 as this is the point of full escape in a host previously exposed to phenotype −0.8.

advantage over the ancestral variant (*Figure 5D*). Coupled with the plausible assumption that large-effect substitutions are rarer than small-effect substitutions (here captured qualitatively by the Gaussian mutation kernel), this predicts the observed pattern of quasi-neutral diversification within antigenic clusters followed by rarer directional 'jumps' in phenotype. Exact rates of antigenic evolution will depend upon how these emergence processes intersect with population-level epidemic dynamics and competition among variants.

## Epidemic dynamics can alter rates of inoculation selection

We used an epidemic-level model to study the consequences of individual-level inoculation selection for population-level antigenic selection. If inoculation selection is efficient, an intermediate initial fraction of immune hosts maximizes the probability that a new antigenic variant infection is created during an epidemic (*Figure 6*). This is due to a trade-off between the frequency of naive or weakly immune 'generator' hosts who can propagate the epidemic and produce new antigenic variants through de novo mutation, and the frequency of strongly immune 'selector' hosts who, if inoculated, are unlikely to be infected, but can facilitate the survival of these new antigenic variants at the point of transmission. As selector host frequency increases, epidemics become rarer and smaller, eventually decreasing opportunities for evolution, but moderate numbers of efficient selectors can substantially increase the rate at which new antigenic variants reach within-host consensus (*Figure 6B,C*).

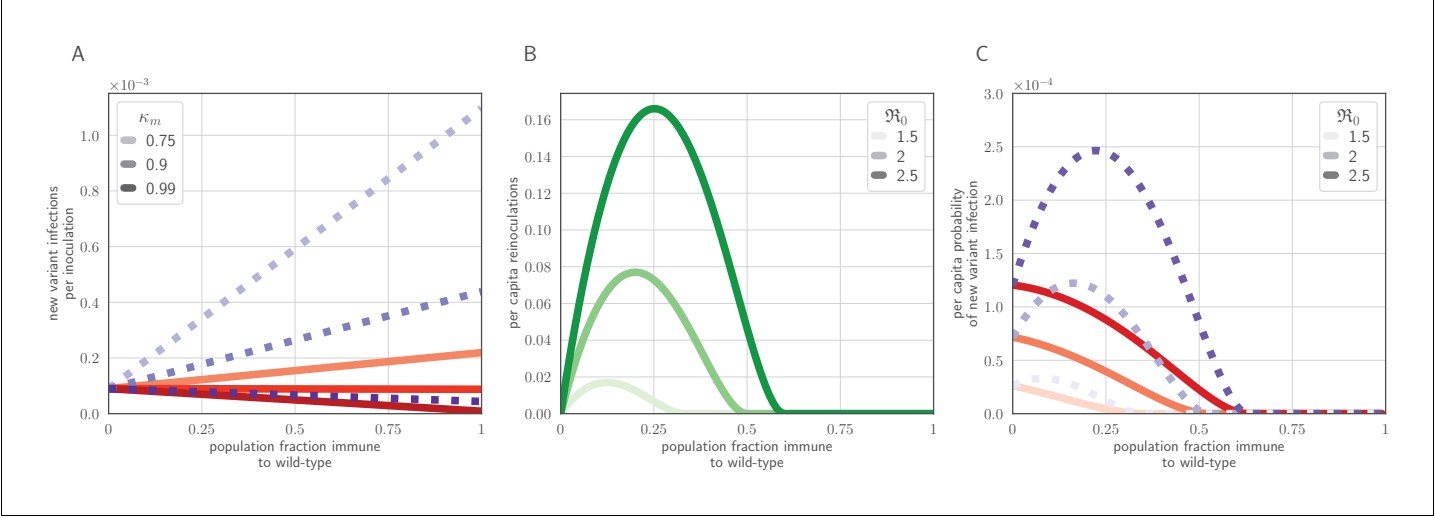

**Figure 6.** Population-level antigenic dynamics resulting from inoculation selection. Analytical model results (see Materials and methods) for population-level inoculation selection, using parameters in **Table 1** and $f_{\mathrm{mt}} = 9 \times 10^{-5}$ unless otherwise stated. (A) Probability per inoculation of a new antigenic variant founding an infection, as a function of fraction of hosts previously exposed to the infecting old antigenic variant virus, mucus bottleneck size $v$ and sIgA cross immunity $\sigma = \kappa_m/\kappa_w$. Red solid lines: $v = 10$. Purple dotted lines: $v = 50$. (B) Expected per-capita reinoculations of previously exposed hosts during an epidemic, given the fraction of previously exposed hosts in the population, if all hosts that were previously exposed to the circulating old antigenic variant virus are fully immune to that variant, for varying population-level basic reproduction number $\mathfrak{R}_0$. (C) Probability per individual host that a new antigenic variant founds an infection in that host during an epidemic, as a function of the fraction of hosts previously exposed to the old antigenic variant. Other hosts naive. $\sigma = 0.75$. Red solid lines: $v = 10$; purple dotted lines: $v = 50$ (as in A).

## Discussion

Any explanation of influenza virus antigenic evolution—and why it is not even faster—must explain why population-level antigenic selection is strong, as evidenced by the typically rapid sequential population-level replacement of old antigenic variants upon the emergence of a major new antigenic variant, but within-host antigenic evolution is rarely observed.

We hypothesized that antibodies present in the respiratory tract mucosa at the time of virus exposure can effectively block transmission, but have only a small effect on viral replication once cells become productively infected. Antigenic selection after successful infection therefore begins with the mounting of a recall response 48–72 hr post infection. In this case, selection pressure can be strong at the point of transmission, but subsequently weak until after the period of virus exponential growth. This mechanistic paradigm reconciles strong but not perfect sterilizing homotypic immunity with rare observations of new antigenic variants in successfully reinfected experienced hosts.

### Alternative explanations for rare new antigenic variants

We consider several possible explanations for the observed phenomenon that new antigenic variants are rare within experienced infected hosts and at the population level prior to cluster transitions. But among these candidate hypotheses, only the mechanism of small inocula, transmission-blocking mucosal antibodies, and a slow-to-mount recall adaptive immune response can explain all the aforementioned empirical observations simultaneously.

Alternative possibilities include strong immune protection through antibody neutralization during early viral replication (an immediate recall response), heterogeneous neutralization rates during early viral replication, new antigenic variants that are deleterious in the absence of antibody selection, and the need for new variants to emerge against a favorable genetic background.

As shown above, protection through antibody neutralization early in replication can result in rare within-host observation of new antigenic variants, but it contradicts understanding of antibody kinetics and makes other empirical predictions that are unrealistic. It implies that homotypic challenge has a binary outcome: either it results in an undetectable infection that is rapidly cleared or it results

in a visible infection dominated by an escape mutant (*Figure 2*). This is how influenza viruses have been hypothesized to behave in previous models of immune escape (*Luo et al., 2012*). But empirical work (*Debbink et al., 2017*; *McCrone et al., 2018*; *Javaid et al., 2020*) and human challenge studies (*Clements et al., 1986*; *Memoli et al., 2020*) have shown that detectable reinfection of experienced hosts can occur without observable immune escape. Another empirical prediction of such a model is that intermediately immune hosts should be efficient selectors, since they neutralize the old variant virus poorly enough to allow it to grow, but strongly enough to impose antigenic selection upon that growing population (*Volkov et al., 2010*). While such intermediately immune hosts should be present from the beginning of a new antigenic cluster's circulation (*Fonville et al., 2014*), new variants are rarely observed until a new variant has circulated for multiple years (*Smith et al., 2004*).

Antibody neutralization during early replication could avoid binary outcomes if individual hosts are heterogeneous in the strength of their neutralizing response, so some individuals clear the infection rapidly while others barely exert antibody selection upon it. But while heterogeneity in immunity exists (*Lee et al., 2019*), this explanation requires extreme, bimodal hetegoneity to avoid the intermediately immunity regime in which replication selection is efficient (*Volkov et al., 2010*) and again requires an unrealistically early antibody response.

New antigenic variants could be replication-competent, but weakly deleterious within-host in the absence of immune selection and/or compensatory substitutions. Two studies (*Gog, 2008*; *Kucharski and Gog, 2012*) have invoked this hypothesis to explain population-level antigenic dynamics. However, a realistically strong antibody response during virus replication could still promote new variants during infections of experienced hosts, even in the absence of compensatory mutations (see Appendix Section A7.4 for a model analysis). So a hypothesis to explain the relative weakness of selection during replication is still required, especially for weakly deleterious mutants that offer substantial immune escape. That said, under our hypothesis of founder effects and possible inoculation selection, weak new variant deleteriousness could further limit the rate of antigenic evolution by reducing the probability that new variants are inoculated into hosts (see Appendix Section A7.4).

Finally, it has been hypothesized that antigenic mutants can only proliferate at the population level if they arise against a favorable genetic background (*Koelle and Rasmussen, 2015*)—in the absence of deleterious substitutions elsewhere in the genome. But antigenic cluster transitions are frequently polyphyletic (see Appendix Section A6): a new variant emerges quasi-simultaneously in multiple virus lineages. Since these lineages should have different genetic backgrounds, this suggests that favorable backgrounds are readily available, and that emergence is limited instead by the presence or absence of selection pressure.

We discuss these alternative explanations in more depth in the Appendix (Section A7.4).

## Relationship to prior influenza virus transmission bottleneck literature

Previous literature on influenza virus transmission mentions a 'selective bottleneck' (*Wilker et al., 2013*; *Moncla et al., 2016*; *Sobel Leonard et al., 2016*), but those studies do not refer to antigenic inoculation selection. Rather, a 'selective bottleneck' typically refers either to a tight neutral bottleneck that leads to stochastic loss of diversity or to non-antigenic factors that lead to preferential transmission of certain variants (*Moncla et al., 2016*). An important exception is (*Lumby et al., 2018*). Those authors studied ferret transmission experiments and partitioned selection for adaptive mutants (not necessarily antigenic) into selection for transmissibility (acting at a potentially tight bottleneck) and selection during exponential growth. Subsequently, several studies have hypothesized that influenza virus antigenic selection might be weak in short-lived infections of individual experienced hosts and might occur at the point of transmission (*Petrova and Russell, 2018*; *Han et al., 2019*; *Lumby et al., 2020*).

To our knowledge, however, ours is the first study to undertake a mechanistic, model-based comparison between the role of antigenic selection at the point of transmission and that of antigenic selection during replication, to show that immunologically plausible mechanisms could make the former more salient than the latter, and to connect that finding to the rarity of observable new antigenic variants in homotypically reinfected human hosts. We discuss the relationship between this paradigm and those put forward in previous modeling studies at further length in Appendix Section A7.

## Limitations and remaining uncertainties

The study presented here nonetheless has important limitations that suggest opportunities for future investigation. This is a modeling study, and a mainly theoretical one. That is out of necessity. Quality experiments of the kind that are necessary to observe selection at the point of transmission directly and measure its strength have not been published, and we were unable to find any experimental measurements of within-host competition between known antigenic variants. For additional discussion of unmodeled biological realities and mechanistic uncertainties, see Appendix Sections A2 and A5.

### Within-host model

Our within-host model is a simple target cell-limited model, but the decrease in infectible cells as the infection proceeds can be qualitatively interpreted as any and all antigenicity-agnostic limiting factors that come into play as the virion and infected cell populations grow. This could include the action of innate immunity, which acts in part by killing infected cells and by rendering healthy cells difficult to infect via inflammation (see immunological review in Appendix Section A2). The key mechanistic role played by the target cells in our model is to introduce a non-antigenic limiting factor on infections. This prevents an infection from repeatedly evolving out from under successive well-matched antibody responses, as occurs in HIV and in influenza patients with compromised immune systems (*Xue et al., 2017*). These factors limit the virus regardless of whether the infection remains confined to the upper respiratory tract or also infect the lower respiratory tract, thereby gaining access to additional target cells (*Koel et al., 2019*). The key is that non-antigenic factors prevent persistent large virus populations.

Similarly, the antibody response we introduce at 48 hours is qualitative—it is modeled simply as an increase in the virion decay rate for antibody-matching virions. This response could represent IgA targeted at HA, but other antigenicity-specific modes of virus control could also be subsumed under the increased virion decay rate, for instance IgG antibodies, antibody dependent cellular cytotoxicity (ADCC), antibodies against the neuraminidase protein, or others. The key point we establish is that none of these mechanisms efficiently replication select because they all emerge once non-antigenic limiting factors have come into play. They speed clearance, but should not substantially alter virus evolution. A corollary to this point is that there are many mechanisms—and interventions—that could reduce the severity of influenza infections without substantially speeding up antigenic evolution, including universal vaccines.

### Point of transmission

A key proposal of our study is that population-level antigenic selection and homotypic protection are mediated by antibody neutralization (likely sIgA) at the point of transmission. Currently, empirical evidence for antibody protection at the point of transmission is mostly indirect. Most of this evidence comes from human and animal challenge studies (*Clements et al., 1986*; *Memoli et al., 2020*; *Le Sage et al., 2020*). In these studies, individuals who are challenged with the same antigenic variant sometimes display apparent sterilizing immunity, but other times develop detectable infections. The study by *Memoli et al., 2020* is notable for having used very large inocula—$10^6$ or $10^7$ TCID$_{50}$. Despite these high doses, two of the challenge subjects had neither detectable virus nor seroconversion. Similarly, ferret experiments (*Le Sage et al., 2020*) found that many experienced ferrets developed sufficiently sterilizing immunity to prevent the virus from ever being detected, while some experienced ferrets showed briefly detectable infections that were then cleared. While we cannot rule out a powerful immediate cellular response that was differentially evaded in the various subjects, we believe that our model, coupled with existing understanding of the timing of cellular responses and the speed of influenza virus replication, provides a more parsimonious explanation.

Another limitation of our study is that, while we put forward mucosal sIgA as a biologically-documented potential mechanism of immune protection at the point of inoculation that would not lead to strong selection during early viral replication, no modeling study can establish such a mechanism without empirical investigation. Our study reveals that such an empirical investigation would be of substantial scientific value.

We model neutralization at the point of transmission as a binomial process. Each virion is independently neutralized with a probability $\kappa_i$ that depends on its antigenic phenotype and the host's

immune history. As we discuss in Appendix Section A5.2, this assumption of independence may be violated in practice. Careful experiments are required to develop a more realistic model of neutralization at the point of transmission.

Moreover, individual variation in immune system properties and complex effects of host immune history (*Fonville et al., 2014*; *Lee et al., 2019*) mean that even a pair of hosts who have both been previously exposed to the currently circulating variant may exert different selection pressures at the point of transmission. Modeling neutralization as a series of independent events that depend only on host history and virus phenotype is a baseline: it allows us to establish that antigenic selection at the point of transmission is possible and to show what its consequences might be. But a more realistic model will be required to predict the selective pressures imposed by real hosts with real immune histories on real virions.

Finally, while we believe neutralization at the point of transmission is a crucial mechanism of protection against detectable reinfection for influenza, this may not be true for all RNA viruses. Some, such as measles and varicella, have long incubation periods even in naive hosts and induce reliable, long-lasting immune protection against detectable reinfection. This may be because they replicate slowly enough that they cannot 'outrun' the adaptive response as influenza can. Neither shows influenza virus-like patterns of clocklike immune escape, suggesting that (1) escape mutants may be less available and (2) the adaptive response acts on a small population and is forceful.

## Parameter uncertainties

We parameterized our models based on estimates from previous studies, but there are not good estimates for several important quantities. There are no high quality estimates of the rate of antibody-mediated neutralization in the presence of a homotypic antibody response or of how much this rate is reduced by particular antigenic substitutions, and there may be substantial inter-individual variation (*Lee et al., 2019*). Within-host timeseries data from antigenically heterogeneous infections are needed to estimate these quantities. Similarly, there are no good empirical data on the size of the bottlenecks that precede or follow the sIgA bottleneck (*Figure 3A*). Better estimates of these bottlenecks, and of the probabilities of neutralization for individual virions encountering mucosal sIgA antibodies, would give more certainty about the strength of inoculation selection relative to neutral founder effects.

We also do not have a clear sense of exactly how neutralization probability in the respiratory tract mucosa (parameter $\kappa$ in our model) and neutralization rate during replication (parameter $k$ in our model) relate. We expect them to be positively related, but the exact strength and shape of this relationship is unknown. Knowing whether major antigenic changes reduce both equally or reduce one more than the other could help us better quantify the potential strength of inoculation selection and replication selection. In short, better mechanistic understanding of mucosal antibody neutralization could be extremely valuable for understanding and potentially predicting influenza virus evolution.

## Scaling up to the population level

How readily a particular individual host or host population helps new antigenic variants reach within-host consensus depends upon several unknown quantities: (1) how host susceptibility changes with extent of antigenic dissimilarity, (2) the ratio of virions that encounter sIgA to virions that found the infection ($v/b$), (3) the probability that a single new antigenic variant virion inoculated alongside old antigenic variant virions evades neutralization $1 - \kappa_m$ (*Figure 3E–H*), (4) the duration $\tau$ from the onset of the antibody response to the time of transmission. Better empirical estimates of these quantities could shed light on how the distribution of host immunity shapes antigenic evolution. However, over a range of biologically plausible parameter values, our model contradicts the existing hypothesis that antigenic novelty appears when moderately immune hosts fail to block transmission and then select upon a growing virus population (*Grenfell et al., 2004*; *Volkov et al., 2010*). For influenza viruses, hosts whose mucosal immunity regularly blocks old antigenic variant transmission may be crucial. Mucosal immunity not only produces a population-level advantage for new variants but may also play a role in their within-host emergence (*Figure 3E,H*).

## Implications

Our study has a number of implications for the study and control of influenza viruses.

### Importance of host heterogeneity

Experienced hosts are undoubtedly heterogeneous in their immunity to a given influenza variant (*Lee et al., 2019*), so the overall population average protection against homotypic reinfection with variant $i$, $z_i$, is in fact an average over experienced hosts. Our model implies that the degree of neutralization difference between ancestral variant virions and new antigenic variant virions at the point of transmission strongly affects the probability of inoculation selection. Hosts with more focused immune responses—highly-specific antibodies that neutralize old antigenic variant virions well and new antigenic variant virions poorly—could be especially good inoculation selectors and important sources of population-level antigenic selection. Hosts who develop less specific memory responses, such as very young children (*Neuzil et al., 2006*), could be less important. Similarly, immune-compromised hosts are excellent replication selectors (*Xue et al., 2017*; *Lumby et al., 2020*), and so their role in the generation of antigenic novelty and their impact on overall population-level diversification rates deserve further study.

### Small-population-like evolution

Prior modeling has suggested that despite repeated tight bottlenecks at the point of transmission, evolution of influenza viruses should resemble evolution in idealized large populations (*Sigal et al., 2018*). In large populations, advantageous variants with small selective advantages should gradually fix and weakly deleterious variants should be purged. In *Sigal et al., 2018*, diversity is rapidly generated and fit variants are selected to frequencies at which they are likely to pass through even a tight bottleneck. This is likely true of the phenotypes modeled, which include receptor-binding avidity and virus budding time (*Sigal et al., 2018*). These phenotypes affect virus fitness throughout the timecourse of infection, so they can be efficiently replication-selected (where selection is manifest in the direct competition to infect cells rather than the indirect competition to escape antibodies). Indeed, next-generation sequencing studies have found observable adaptative evolution of non-antigenic phenotypes in individual humans infected with avian H5N1 viruses (*Welkers et al., 2019*).

Seasonal influenza antigenic evolution does not resemble idealized large population evolution. Within an antigenic cluster, influenza viruses acquire substitutions that change the antigenic phenotype by small amounts. Given the large influenza virus populations within individual hosts, we might expect a quasi-continuous directional pattern of evolution away from prior population immunity. Between clusters, evolution is indeed strongly directional: only 'forward' cluster transitions are observed. These are jumps—large antigenic changes. But incremental within-cluster evolution is *not* directional: the virus often evolves 'backward' or 'sideways' in antigenic space toward previously circulated variants (see Figures 1 and 2 of *Smith et al., 2004*).

This noisy jump pattern is easy to explain in light of the weakness of replication selection and the importance of antigenic founder effects. Selection acts on the small sub-sample of donor-host diversity that passes through the excretion, inter-host, and mucus bottlenecks to encounter the sIgA bottleneck. Evolution via inoculation selection is therefore slower and more affected by stochasticity than evolution via replication selection (*Figure 3E–F*). It resembles evolution in small populations—weakly adaptive and weakly deleterious substitutions become nearly neutral (*Kimura, 1968*; *Ohta, 1992*). If influenza virus evolution were not nearly neutral for small-effect substitutions at the within-host scale, it would be surprising to observe 'backward' antigenic changes and noisy evolution at higher scales (*Figure 5A–D*). In fact, there may be analogous 'neutralizing' population dynamics at higher scales as well, and those may also be needed to explain population-level noisiness. But whatever happens at higher scales, within-host replication selection creates a strong directional bias in population-level antigenic diversification, introducing many small forward antigenic changes (*Figure 5A–D*). Inoculation selection does not necessarily do this.

### Population level neutralizing dynamics

Local influenza virus lineages rarely persist between epidemics (*Russell et al., 2008*; *Bedford et al., 2015*), and so new antigenic variants must establish chains of infections in other geographic locations in order to survive. New antigenic variant chains are most often founded when inoculations are

common—that is, when existing variants are causing epidemics. Epidemics result in high levels of local competition between extant and new antigenic variant viruses for susceptible hosts (*Hartfield and Alizon, 2015*) as well as metapopulation-scale competition to found epidemics in other locations. These dynamics could create tight bottlenecks between epidemics similar to those that occur between hosts, resulting in dramatic epidemic-to-epidemic diversity losses. That said, if immune hosts are present at the start of an epidemic, there will not be asynchrony between diversity and selection pressure, so new variants may pass through between-epidemic bottlenecks more readily than through between-host bottlenecks. Further work is needed to elucidate mechanisms at the population and meta-population scales.

## Population immunity sets the clock of antigenic evolution

Our work suggests a simple mechanism by which accumulating immunity to an antigenic variant could produce punctuated population-level antigenic evolution. Population-level modeling has shown that influenza virus global epidemiological and phylogenetic patterns can be reproduced if new antigenic variants emerge at the population level with increasing frequency the longer an old antigenic variant circulates (*Koelle et al., 2009*).

If immunity to the old variant only gives new variants a population-level transmission advantage (population-level selection), we anticipate a constant rate of population-level antigenic diversification, with selective sweeps once a new variant has a sufficient population-level advantage over the old variant. If population immunity also helps new variants become the dominant within-host variant through inoculation selection, increasing population immunity to an old variant can produce increasing rates of new variant emergence (*Figure 6A*). Whether this occurs in practice depends on the ecology of hosts and the relative strength of inoculation selection versus drift in partially and fully immune hosts (*Figure 4*).

Potential synergy between brief antigenic replication selection late in infection and subsequent inoculation selection (*Figure 4*) could further promote new variant emergence as population immunity accumulates. However, it is difficult to estimate how much this synergy matters in practice without knowing more about the kinetics of homotypic and heterotypic re-inoculation and reinfection.

One suggestive population-level pattern is that new antigenic variants frequently are observed quasi-simultaneously on multiple branches of the influenza virus phylogeny shortly prior to sweeping (see Appendix Section A6). This suggests that emergence rate and population-level selection pressure do increase together. That said, alternative explanations are possible, such as reduced rates of stochastic loss of new variants at higher scales (e.g. the epidemic scale) with increased population immunity.

Regardless of the strength of these effects, however, our prediction is that new antigenic variant infection foundation events should constitute a rare but non-negligible fraction of transmission events: on the order of 1 in $10^5$ or 1 in $10^4$. This parsimoniously explains why new antigenic variants lineages are hard to observe prior to undergoing positive population-level selection but are readily available to be selected upon (see Appendix Section A6) once that population-level selection pressure has become sufficiently strong. Before that point, they could be rendered unobservable by population-level neutralizing dynamics or by clonal interference from non-antigenic weakly adaptive mutants (*Strelkowa and Lässig, 2012*).

The slow rate of antigenic evolution of A/H3N2 viruses in swine lends further support to this argument. A/H3N2 viruses accumulate genetic mutations at a similar pace in swine and humans, but antigenic evolution is much slower in swine (*ESNIP3 consortium et al., 2016*; *de Jong et al., 2007*). Slaughter for meat means that pig population turnover is high. It follows that the frequency of experienced hosts rarely becomes sufficient to facilitate the appearance of observable new antigenic variants.

The population-level emergence of new antigenic variants, in other words, tracks the accumulation of immunity in the population, not the accumulation of genetic diversity. This suggests that A/H3N2 evolution is indeed selection-limited, not diversity-limited. But much generated antigenic diversity is invisible to surveillance: in the absence of positive selection, it is likely to be lost at bottlenecks.

### Implications for other pathogens

The inoculation and replication selection paradigm has implications for the understanding and management of other pathogens. For example, HIV does not readily evolve resistance to contemporary pre-exposure prophylaxis (PrEP) antiviral drugs, but it can do so when these antivirals are taken by an individual who is already infected with HIV (*Partners PrEP Study Team et al., 2016*). Developing resistance at the moment of exposure is a difficult problem of inoculation selection for the virus, but developing resistance during an ongoing infection is an easier problem of replication selection. Selection on small un-diverse introduced populations may also be of interest in invasion biology and island biogeography.

## Conclusion

The asynchrony between within-host virus diversity and antigenic selection pressure provides a simple mechanistic explanation for the phenomenon of weak within-host selection but strong population-level selection in seasonal influenza virus antigenic evolution. Measuring or even observing antibody selection in natural influenza virus infections is likely to be difficult because it is inefficient and consequently rare. Theoretical studies are therefore essential for understanding these phenomena and for determining which measurable quantities will facilitate influenza virus control. Our study highlights a critical need for new insights into sIgA neutralization and IgA responses to natural influenza virus infection and vaccination. Cross-scale dynamics can decouple selection and diversity, introducing randomness into otherwise strongly adaptive evolution.

# Materials and methods

## Model notation

In all model descriptions, $X += y$ and $X -= y$ denote incrementing and decrementing the state variable $X$ by the quantity $y$, respectively, and $X = y$ denotes setting the variable $X$ to the value $y$. $\dot{X}$ denotes the rate of event $X$.

## Within-host model overview

The within-host model is a target cell-limited model of within-host influenza virus infection with three classes of state variables:

- $C$: Target cells available for the virus to infect. Shared across all virus variants.
- $V_i$: Virions of virus antigenic variant $i$
- $E_i$: Binary variable indicating whether the host has previously experienced infection with antigenic variant $i$ ($E_i = 1$ for experienced individuals; $E_i = 0$ for naive individuals).

New virions are produced through infections of target cells by existing virions, at a rate $\beta C V_i$. Infection eventually renders a cell unproductive, so target cells decline at a rate $\ell \beta C \bar{V}$, where $\bar{V}$ is the total number of virions of all variants. The model allows mutation: a virion of antigenic variant $i$ has some probability $\mu_{ij}$ of producing a virion of antigenic variant $j$ when it reproduces.

Virions have a natural per-capita decay rate $d_v$. A fully active specific antibody-mediated immune response to variant $i$ increases the virion per-capita decay rate for variant $i$ by a factor $k$ (assumed equal for all variants). The degree of activation of the antibody response during an infection is given by a function $M(t)$, where $t$ is the time since inoculation.

We use a parameter $c_{ij}$ to denote the protective strength of antibodies raised against a strain $j$ against a different strain $i$. $c_{ii} = 1$ by definition and $c_{ij} = 0$ indicates complete absence of cross-protection. So if host has antibodies against a strain $j$ but not against the infecting strain $i$, a fully active antibody response raises the virion decay rate for strain $i$ by $c_{ij}k$. If there are multiple candidate forms of cross protection $c_{ij}$ and $c_{ik}$, we choose the strongest. We typically assume that $c_{ij} = c_{ji}$.

We assume that an antibody immune response is raised whenever the host has experienced a prior infection with a partially cross-reactive strain. For notational ease, we define the host's strongest cross-reactivity against strain $i$, $c_i$, by:

$$c_i = \max\{c_{ij}E_j\} \tag{1}$$

So a recall antibody response is raised during an infection with strains $i,j,...$ whenever one of $c_i, c_j, ... > 0$.

For this study, we consider a two-antigenic variant model with an ancestral 'old antigenic variant' virions $V_w$ and novel 'new antigenic variant' virions $V_m$, though the model generalizes to more than two variants. The state variables are charged by the following stochastic events:

$$
\begin{aligned}
C^+ &: \quad C \mathrel{+}= 1 \\
C^- &: \quad C \mathrel{-}= 1 \\
V_w^+ &: \quad V_w \mathrel{+}= 1 \\
V_w^- &: \quad V_w \mathrel{-}= 1 \\
V_m^+ &: \quad V_m \mathrel{+}= 1 \\
V_m^- &: \quad V_m \mathrel{-}= 1
\end{aligned}
\tag{2}
$$

These events occur at the following rates:

$$
\begin{aligned}
\dot{C}^- &= \ell \beta C (V_w + V_m) \\
\dot{C}^+ &= p_C C (1 - C/C_{\max}) \\
\dot{V}_w^+ &= \beta C V_w r_w (1 - \mu) \\
\dot{V}_w^- &= V_w (d_v + c_w k M(t)) \\
\dot{V}_m^+ &= \beta C (V_m r_m + V_w r_w \mu) \\
\dot{V}_m^- &= V_m (d_v + c_m k M(t))
\end{aligned}
\tag{3}
$$

where $M(t)$ is a minimal model of a time-varying antibody response given by:

$$
M(t) = \begin{cases} 1, & t > t_M \\ 0, & \text{otherwise} \end{cases}
\tag{4}
$$

For simplicity, the equations are symmetric between old antigenic variant and new antigenic variant viruses, except that we neglect back mutation, which is expected to be rare during a single infection, particularly before mutants achieve large populations. The parameters $r_w$ and $r_m$ allow the two variants optionally to have distinct within-host replication fitnesses; for all results shown, we assumed no replication fitness difference ($r_w = r_m = r$) unless otherwise stated. A de novo antibody response raised to a not-previously-encountered variant $i$ can be modeled by setting $E_i = 1$ at a time $t_N^i \geq t_M$ post-infection. By default, we model such a de novo response only for fully naive hosts, and assume that it is mounted only against the variant that was most common at the start of infection, which is typically $w$.

We characterize a virus variant $i$ by its within-host basic reproduction number $\mathcal{R}_0^i$, the mean number of progeny virions produced by a single virion at the start of infection in a naive host:

$$
\mathcal{R}_0^i \equiv \frac{\beta C_{\max} r_i}{d_v}
\tag{5}
$$

When parametrizing our model, we fixed the within-host basic reproduction number $\mathcal{R}_0^i$, the initial target cell population $C_{\max}$, the virus reproduction rate $r_i$, and the shared virion decay rate $d_v$. We then calculated the implied $\beta$ according to *Equation (5)*.

Another useful quantity is the within-host effective reproduction number $\mathcal{R}^i(t)$ of variant $i$ at time $t$: the mean number of progeny virions produced by a single virion of variant $i$ at a given time $t$ post-infection.

$$
\mathcal{R}^i(t) \equiv \frac{\beta C(t) r_i}{d_v + c_i k M(t)}
\tag{6}
$$

Note that $\mathcal{R}_0^i$ is $\mathcal{R}^i(0)$ in a naive host, and that if $\mathcal{R}^i < 1$, the variant $i$ virus population will usually decline.

We denote the frequency of new variant virions at time $t$ by $f_m(t) = \frac{V_m(t)}{V_w(t) + V_m(t)}$.

The distribution of virions that encounter sIgA antibody neutralization depends on the mean mucosal bottleneck size $v$ (i.e. the mean number of virions that would pass through the respiratory tract mucosa in the absence of antibodies) and on the frequency of new antigenic variant $f_{\mathrm{mt}} = f_m(t_{\mathrm{inoc}})$ in the donor host at the time of inoculation $t_{\mathrm{inoc}}$. $n_w$ old antigenic variant virions and $n_m$ new antigenic variant virions encounter sIgA. The total number of virions $n_{\mathrm{tot}} = n_w + n_m$ is Poisson-distributed with mean $v$ and each virion is independently a new antigenic variant with probability $f_{\mathrm{mt}}$ and otherwise an old antigenic variant. The principle of Poisson thinning then implies:

$$n_m \sim \mathrm{Poisson}(v f_{\mathrm{mt}})$$
$$n_w \sim \mathrm{Poisson}(v(1 - f_{\mathrm{mt}}))$$

(7)

Note that since $f_{\mathrm{mt}}$ is typically small, the results should also hold for a binomial model of $n_w$ and $n_m$ with a fixed total number of virions encountering sIgA antibodies: $v_{\mathrm{tot}} = v$.

We then model the sIgA bottleneck—neutralization of virions by mucosal sIgA antibodies. Each virion of variant $i$ is independently neutralized with a probability $\kappa_i$. This probability depends upon the strength of protection against homotypic reinfection $\kappa$ and the sIgA cross immunity between variants $\sigma$ ($0 \leq \sigma \leq 1$):

$$\kappa_w = \kappa \max\{E_w, \sigma E_m\}$$
$$\kappa_m = \kappa \max\{E_m, \sigma E_w\}$$

(8)

Since each virion of strain $i$ in the inoculum is independently neutralized with probability $\kappa_i$, then given $n_w$ and $n_m$, the populations that compete the pass through the cell infection bottleneck $x_w$ and $x_m$ are binomially distributed:

$$x_w \sim \mathrm{Binomial}(n_w, \kappa_w)$$
$$x_m \sim \mathrm{Binomial}(n_m, \kappa_m)$$

(9)

By Poisson thinning, this is equivalent to:

$$x_w \sim \mathrm{Poisson}(v(1 - f_{\mathrm{mt}})(1 - \kappa_w))$$
$$x_m \sim \mathrm{Poisson}(v f_{\mathrm{mt}}(1 - \kappa_m))$$

(10)

At this point, the remaining virions are sampled without replacement to determine what passes through the cell infection bottleneck, $b$, with all virions passing through if $x_w + x_m \leq b$, so the final founding population is hypergeometrically distributed given $x_w$ and $x_m$.

If $\kappa_w = \kappa_m = 0$, $f_{\mathrm{mt}}$ is small, and $v$ is large, this is approximated by a binomially distributed founding population of size $b$, in which each virion is independently a new antigenic variant with (low) probability $f_{\mathrm{mt}}$ and is otherwise an old antigenic variant. Alternatively, it can be approximated by a Poisson distribution with a small mean: $f_{\mathrm{mt}} b \ll 1$. So the probability that a new variant survives the bottleneck in the absence of mucosal neutralization is:

$$p_{\mathrm{drift}} \approx 1 - (1 - f_{\mathrm{mt}})^b \approx f_{\mathrm{mt}} b$$

(11)

When there is mucosal antibody neutralization, the variant's survival probability can be reduced below this (inoculation pruning) or promoted above it (inoculation promotion), depending upon parameters. There can be inoculation pruning even when the new antigenic variant is more fit (neutralized with lower probability, positive inoculation selection) than the old antigenic variant (see *Figure 3E*).

## Within-host model parameters
Default parameter values for the minimal model and sources for them are given in *Table 1*.

## Selection and drift within hosts
In this section, we derive analytical expressions for the within-host frequency of the new variant over time in an infected host (*Figure 7*), the probability distribution for the time of the first de novo

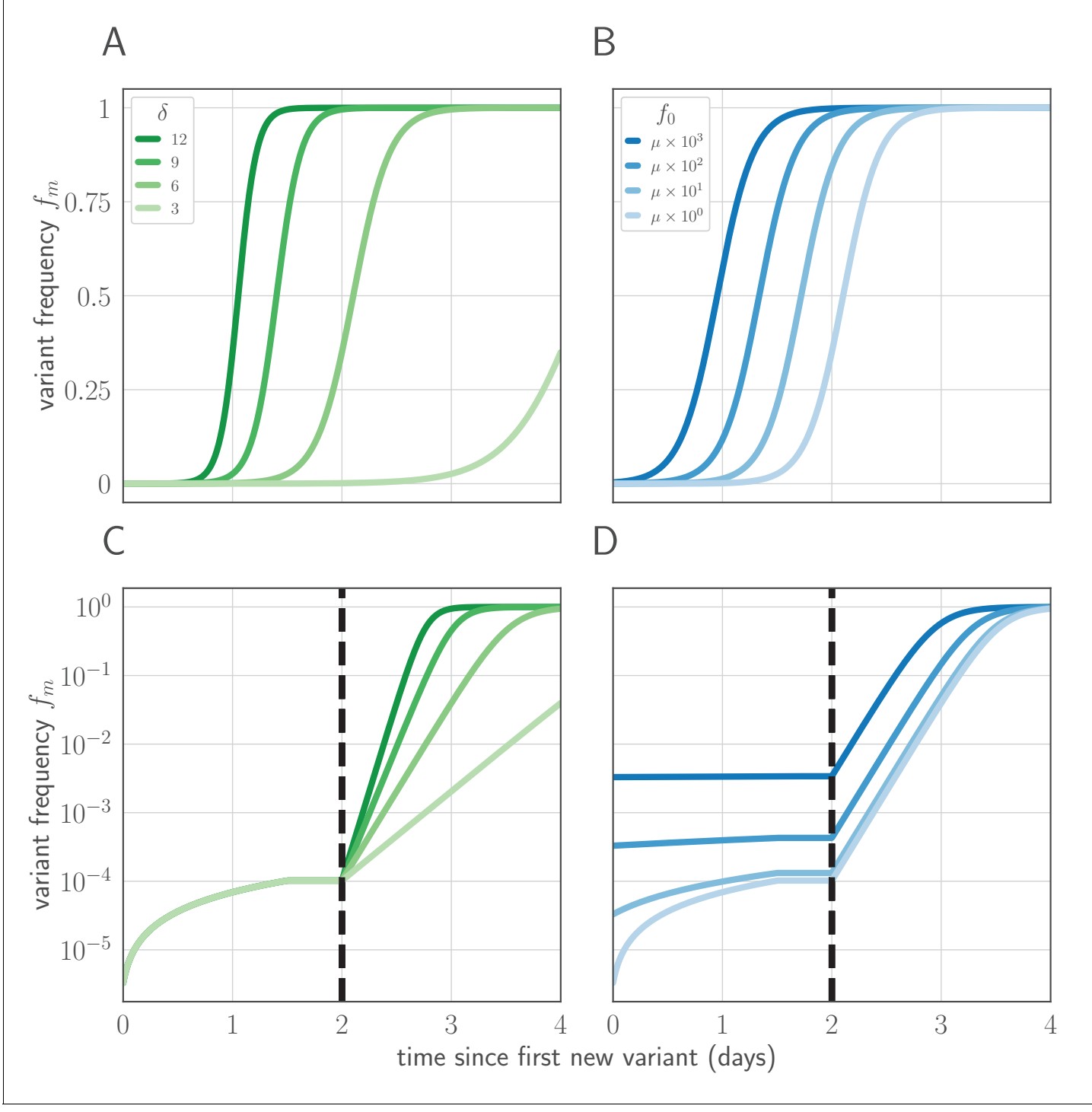

**Figure 7.** Variant within-host frequency as a function of time and initial variant frequency, according to derived replicator equation (*Equations 13, 15*). (A, B) Variant frequency over time for an initially present new variant. (A) Selection strength $\delta$ varied, with initial frequency $f_0$ equal to the mutation rate $\mu = 0.33 \times 10^{-5}$. (B) Initial frequency $f_0$ varied, with $\delta = 6$. (C, D) Variant frequency over time when antigenic selections begins at $t = 2$ days after first variant emergence, with ongoing mutation prior to that point. (C) $\delta$ varied and $f_0$ fixed as in (A); (D) $f_0$ varied and $\delta$ fixed as in (B). Parameters as in *Table 1* unless otherwise noted.

mutation to produce a surviving new variant lineage, and the approximate probability of replication selection to frequency x by time t given our parameters.

## Within-host replicator equation

The within-host frequency of the new variant, $f_m$, obeys a replicator equation of the form:

$$\frac{\mathrm{d}f_m}{\mathrm{d}t} = f_m(1 - f_m)\delta(t) \tag{12}$$

where $\delta(t)$ is the fitness advantage of the new antigenic variant over the old antigenic variant at time $t$ (see Appendix Section A3 for a derivation).

If the variant is neutral in the absence of antibodies, then $\delta(t) = k(c_w - c_m)$ if $t > t_M$ and $\delta(t) = 0$ otherwise. Let $t_e$ denote the first time during the infection that a de novo mutation produces a surviving new antigenic variant lineage. If $t_e \leq t_M$, then at a time $t \geq t_M$:

$$f_m(t) = \frac{e^{\delta(t - t_M)}}{e^{\delta(t - t_M)} + f_M^{-1} - 1} \tag{13}$$

where $\delta = k(c_w - c_m)$ and $f_M = f_m(t_M)$.

When additional mutations after the first cannot be neglected, we add a correction term to $\frac{\mathrm{d}f_m}{\mathrm{d}t}$ for $t_e < t < \min\{t_M, t_{\text{peak}}\}$ (**Figure 7**), where $t_{peak}$ is the time of peak virus population:

$$\frac{\mathrm{d}f_m}{\mathrm{d}t} = f_m(1 - f_m)\delta(t) + \mu\mathcal{R}_0 d_v(1 - 2f_m + f_m^2) \tag{14}$$

which for $\delta \neq 0$ yields:

$$f_m(t) \approx \frac{A\exp(\delta t) + \mu g_0}{A\exp(\delta t) + \mu g_0 - \delta}$$
$$A = \frac{f_0}{1 - f_0}(\mu g_0 - \delta) - \frac{\mu g_0}{f_0} \tag{15}$$

And when $\delta = 0$:

$$f_m(t) \approx \frac{\frac{1}{1 - f_0} + \mu g_0 t - 1}{\frac{1}{1 - f_0} + \mu g_0 t} \tag{16}$$

where $g_0 = \mathcal{R}_0 d_v$ and $f_0 = f_m(t_e)$. See Appendix Section A3.2 for derivations and discussion.

## Distribution of first mutation times

In our stochastic model, new variant lineages that survive stochastic extinction are produced by de novo mutation according to a continuous-time, variable-rate Poisson process. The cumulative distribution function for the time of the first successful mutation, $t_e$, depends on the mutation rate μ and the per-capita rate at which old antigenic variant virions are produced, $g_w(t) = r_w\beta C(t)$. It also depends on $p_{\text{sse}}$, the probability that the generated new antigenic variant survives stochastic extinction. Denoting the new variant per-capita virion production rate $g_m(t) = r_m\beta C(t)$, we calculate $p_{\text{sse}}$ using a branching process approximation (**Ball et al., 2016**).

$$p_{\text{sse}} = \frac{g_m(t)}{g_m(t) + d_v + kM(t)\max\{E_m, cE_w\}} \tag{17}$$

Surviving mutants therefore occur at a rate $\lambda_m(t)$:

$$\lambda_m(t) = \mu g_w(t) V_w(t) p_{\text{sse}} \tag{18}$$

We define the cumulative rate $\Lambda_m(x)$:

$$\Lambda_m(x) = \int_0^x \lambda_m(x)dx \tag{19}$$

It follows that the CDF of the first mutation time $t_e$ is:

$$P(t_e < x) = 1 - e^{-\Lambda_m(x)} \tag{20}$$

This expression is exact for any given realization of the stochastic model if the realized values of the random variables $V_w(t)$, $C(t)$, and $g_w(t)$ are used. In practice, we mainly use it to get a closed form for the CDF of $t_e$ by making the approximation that $C(t) \approx C_{\max}$ early in infection. This yields approximations for $g_w(t)$, $g_m(t)$, and $V_w(t)$:

$$g_w(t), g_m(t) \approx g_0 \equiv \mathcal{R}_0 d_v \tag{21}$$

$$V_w(t) \begin{cases} b \exp((g_0 - d_v)t) & t \leq t_M \\ \\ b \exp((g_0 - d_v)t_M) \exp((g_0 - d_v - c_w k)(t - t_M)) & t > t_M \end{cases} \tag{22}$$

The resultant approximate solution for the CDF of new variant mutation times agrees well with simulations (**Figure 8**).

The slightly earlier simulated mutation times in the immediate recall response case (**Figure 8A**) would only make replication selection more likely in that case than our analytical approximation suggests.

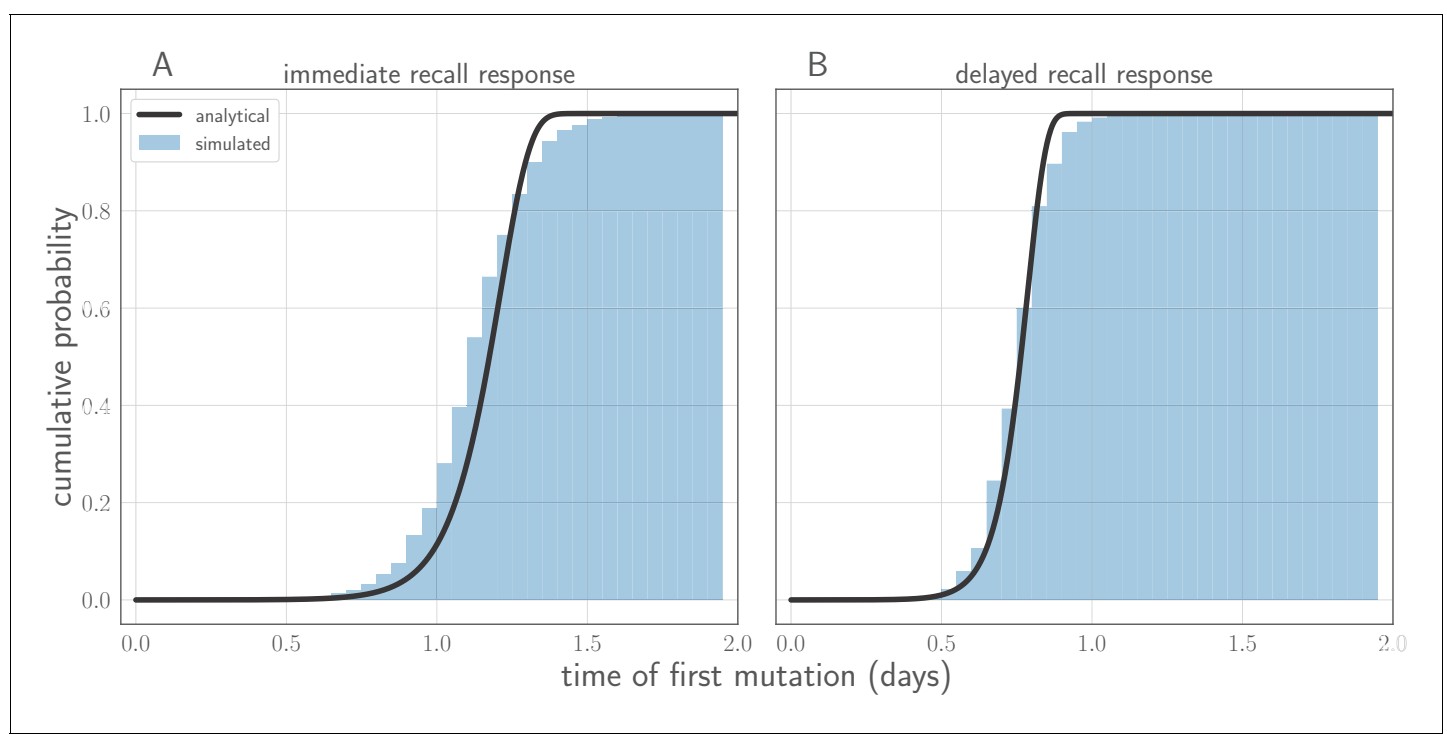

**Figure 8.** Comparison of analytically calculated cumulative distribution function (CDF) for time of first successful de novo mutation with simulations. Black line shows analytically calculated CDF. Blue cumulative histogram shows distribution of new variant mutation times for 250,000 simulations from the stochastic within-host model with (**A**) an immediate recall response ($t_M = 0$) and (**B**) a realistic recall response at 48 hr post-infection ($t_M = 2$). Other model parameters as in **Table 1**. Note that time of first successful mutation tends to be later with an immediate recall response than with a delayed recall response. This occurs because the cumulative number of viral replication events grows more slowly in time at the start of the infection because of the strong, immediate recall response.

## Required mutation time for a variant to reach a given frequency

By inverting the within-host replicator equation, we can also calculate the time $t^*(x,t)$ by which a new variant must emerge if it is to reach at least frequency $x$ by time $t$. We show (see Appendix Section A9.4 for derivation) that there are two candidate values for $t^*$, depending on whether the time that the new variant first emerges ($t_e$) is before or after the onset of the antibody response ($t_M$):

If $t_e \leq t_M$:

$$t_-^*(x,t) = \frac{\ln\left[\frac{1-x}{x}\exp(\delta(t-t_M)) + 1\right] - \ln b}{g_0 - d_v} \tag{23}$$

If $t_e > t_M$:

$$t_+^*(x,t) \approx \frac{\ln(\frac{1-x}{x}) - \ln(b) + \delta t - c_w k t_M}{g_0 - d_v - c_w k + \delta} \tag{24}$$

It may be that $t_+^*(x,t) > t$ and $t_-^*(x,t) > t$. This indicates that the mutant will be at frequency $x$ if it emerges at $t$ itself. In that case, we therefore have $t^*(x,t) = t$. So combining:

$$t^*(x,t) = \begin{cases} t_-^*(x,t) & t_-^*(x,t) < t \text{ and } t_-^*(x,t) \leq t_M \\ t_+^*(x,t) & t_+^*(x,t) < t \text{ and } t_-^*(x,t) > t_M \\ t & \text{otherwise} \end{cases} \tag{25}$$

Finally, it is worth noting that in the case of a complete escape mutant ($c_m = 0$, $\delta = c_w k$), the approximate expression for $t_+^*$ is exactly equal to the equivalent approximate expression for $t_-^*$:

$$t^*(x,t) \approx \frac{\ln(\frac{1-x}{x}) - \ln(b) + c_w k(t - t_M)}{g_0 - d_v} \tag{26}$$

This is a linear function of $t$.

## Probability of replication selection

Given this and the new variant first mutation time CDF calculated in *Equation 20*, it is straightforward to calculate the probability of replication selection to a given frequency $a$ by time $t$, assuming that $\mathcal{R}_0^w > 1$ early in infection:

$$p_{\text{repl}}(a,t) = P(t_e < t^*(a,t)) = 1 - e^{-\Lambda(t^*(a,t))} \tag{27}$$

This analytical model agrees well with simulations (*Figure 9*). We use it used to calculate the heatmaps shown in *Figure 1*, with the $C(t) \approx C_{\max}$ early infection approximations that give us a closed form for $\Lambda_m(x)$.

When $t^* > t_M$, the integral $\int_0^{t^*} \lambda_m(x)dx$ can be evaluated piecewise, first from 0 to $t_M$, and then from $t_M$ to $t^*$. A similar approach can be used to evaluate the probability density function (PDF) of first mutation times, as needed.

Finally, note that for $t_e < t_M$, the expression for $p_{\text{repl}}$ in practice depends only on $\delta = (c_w - c_m)k$, not on $c_w$, $c_m$ and $k$ separately. In the absence of an antibody response, the mutant is generated with near certainty by 1 day post-infection ($P(t_e < 1) \approx 1$, *Figure 8B*). So when $t_M \geq 1$, values for $p_{\text{repl}}$ calculated with $c_w = 1, c_m = 0$, and $\delta = k$—as in (*Figure 1C,D*)—will in fact hold for any $c_w$, $c_m$, and $k$ that produce that fitness difference $\delta$.

When $\mathcal{R}_0^w < 1$ early in infection, the probability of replication selection depends on the probability of generating an escape mutant before the infection is extinguished:

$$p_{\text{repl}} \approx \frac{\mathcal{R}^w(0)}{1 - \mathcal{R}^w(0)} b \mu p_{\text{sse}} \tag{28}$$

See Appendix Section A3.6 for a derivation.

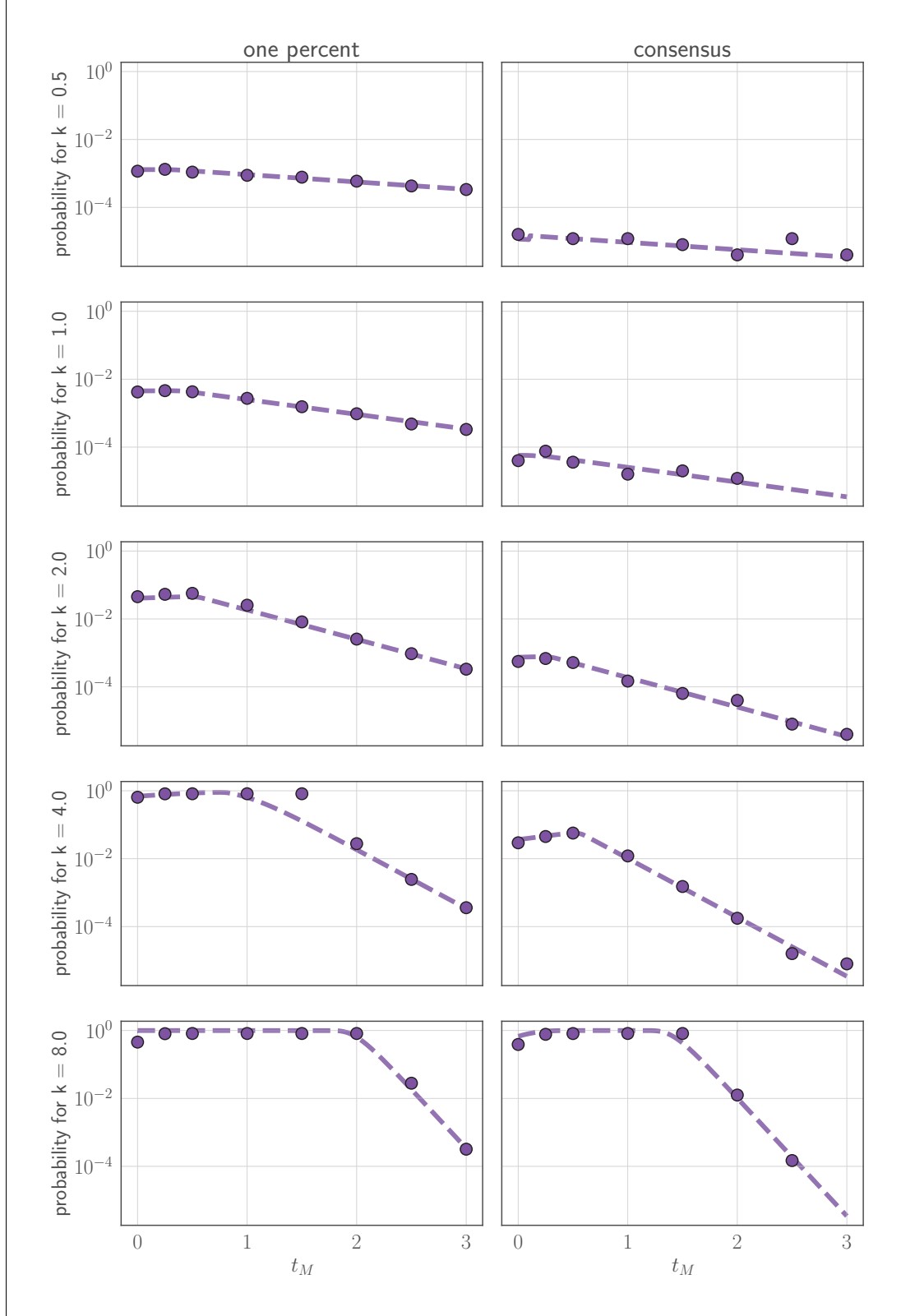

**Figure 9.** Comparison of analytically calculated probability of replication selection with stochastic simulations. Probability of replication selection to one percent (left column) or consensus (right column) by $t = 3$ days post-infection as a function of $k$ and $t_M$ for 250,000 simulations from the stochastic within-host model. $c_w = 1, c_m = 0$ (so the fitness difference $\delta = k$). Other model parameters as parameters in **Table 1**. Dashed lines show analytical prediction and dots show simulation outcomes.

## Point of transmission model

In this section, we describe our model of the point of transmission, including how sIgA neutralization may impose selection pressure.

### Transmission probability

Given contact between an infected host and an uninfected host, we assume that a transmission attempt occurs with probability proportional to the current virus population size $V_{\text{tot}}$ in the infected host:

$$P(\text{transmit}) = 1 - \exp\left(\log(2)\frac{V_{\text{tot}}}{V_{50}}\right) \tag{29}$$

The parameter $V_{50}$ sets the scaling, and reflects virus population size at which there is a 50% chance of successful transmission to a naive host. We used a default value of $V_{50} = 1 \times 10^8$ virions. In addition to this *probabilistic model*, we also consider an alternative *threshold model* in which a transmission attempt occurs with certainty if $V_{\text{tot}}$ is greater than a threshold $\theta$ and does not occur otherwise.

### Bottleneck survival

If there are $x_w$ old antigenic variant virions and $x_m$ new variant virions competing to pass through the cell infection bottleneck, at least one new variant virion passes through with probability:

$$p_{\text{cib}}(x_w, x_m, b) = 1 - \frac{\binom{x_w}{b}}{\binom{x_w + x_m}{b}} \tag{30}$$

Summing $p_{\text{cib}}(x_w, x_m)$ over the possible values of $x_w$ and $x_m$ weighted by their joint probabilities yields the new variant's overall probability of successful onward transmission $p_{\text{surv}}(f_{\text{mt}}, v, b, \kappa, c)$:

$$p_{\text{surv}} = \sum_{x_m, x_w} P(x_m, x_w) p_{\text{cib}}(x_w, x_m, b) \tag{31}$$

$P(x_w, x_m)$ is the product of the probability mass functions for $x_w$ and $x_m$:

$$P(x_m, x_w) = \frac{\bar{w}^{x_w} \bar{m}^{x_m}}{x_w! \, x_m!} e^{-(x_w + x_m)} \tag{32}$$

where $\bar{w} = v(1 - f_{\text{mt}})(1 - \kappa_w)$ and $\bar{m} = v f_{\text{mt}}(1 - \kappa_m)$.

At low donor-host variant frequencies $f_{\text{mt}} \ll 1$, $p_{\text{surv}}$ can be approximated using the fact that almost all probability is given to $x_m = 0$ or $x_m = 1$ (0 or 1 new antigenic variant after the sIgA bottleneck).

At least one new antigenic variant survives the sIgA bottleneck with probability:

$$p_{\text{inoc}}(\kappa_m, v, f_{\text{mt}}) = 1 - e^{-v f_{\text{mt}}(1 - \kappa_m)} \tag{33}$$

If $f_{\text{mt}}$ is small, there will almost always be at most one such virion ($x_m = 1$). That new antigenic variant virion's probability of surviving the cell infection bottleneck depends upon how many old antigenic variant virions are present after neutralization, $x_w$:

$$p_{\text{cib}}(x_w, 1, b) = 1 - \frac{\binom{x_w}{b}}{\binom{x_w + 1}{b}} = 1 - \frac{x_w - b + 1}{x_w + 1} = \frac{b}{x_w + 1} \tag{34}$$

Summing over the possible $x_w$, we obtain a closed form for the unconditional probability $p_{\text{cib}}(\kappa_w, v, b)$ (see Appendix Section A9.5 for a derivation):

$$p_{\text{cib}}(\kappa_w, v, b) = (1 - e^{-\bar{w}})\frac{b}{\bar{w}} + e^{-\bar{w}} \sum_{j=0}^{b-1} \frac{\bar{w}^j}{j!}(1 - \frac{b}{j+1}) \tag{35}$$

where $\bar{w} = v(1 - f_{\text{mt}})(1 - \kappa_w)$ is the mean number of old antigenic variant virions present after sIgA neutralization. If $b = 1$, this reduces the just the first term, and it is approximately equal to just the first term when $\bar{w}$ is large, since $e^{-\bar{w}}$ becomes small.

It follows that there is an approximate closed form for the probability that a new variant survives the final bottleneck:

$$p_{\text{surv}}(\kappa_m, \kappa_w, v, b, f_{\text{mt}}) \approx p_{\text{inoc}}(\kappa_m, v, f_{\text{mt}}) * p_{\text{cib}}(\kappa_w, v, b) \tag{36}$$

We use this expression to calculate the analytical new variant survival probabilities shown in the main text, and to gain conceptual insight into the strength of inoculation selection relative to neutral processes (see Appendix Section A4.5).

## Neutralization probability and probability of no infection

A given per-virion sIgA neutralization probability $\kappa_i$ implies a probability $z_i$ that a transmission event involving only virions of variant $i$ fails to result in an infected cell.

Some transmissions fail even without sIgA neutralization; this occurs with probability $\exp(-v)$. Otherwise, with probability $1 - \exp(-v)$, $n_i$ virions must be neutralized by sIgA to prevent an infection. We define $z_i$ as the probability of no infection given inoculated virions in need of neutralization (i.e. given $n_i > 0$).

The probability that there are no remaining virions of variant $i$ after mucosal neutralization is $\exp(-v(1 - \kappa_i))$. So we have:

$$z_i = \frac{\exp(-v(1 - \kappa_i)) - \exp(-v)}{1 - \exp(-v)} \tag{37}$$

This can be solved algebraically for $\kappa_i$ in terms of $z_i$, but it is more illuminating to express $\kappa_i$ in terms of the overall probability of no infection given inoculation $p_{\text{no}}$ and then find $p_{\text{no}}$ in terms of $z_i$:

$$\begin{aligned} p_{\text{no}} &= \exp(-v(1 - \kappa_i)) \\ \kappa_i &= 1 + \frac{\ln(p_{\text{no}})}{v} \end{aligned} \tag{38}$$

An infection occurs with probability $(1 - \exp(-v))(1 - z_i)$ (the probability of at least one virion needing to be neutralized times the conditional probability that it is not) so $p_{\text{no}}$ in terms of $z_i$ is:

$$p_{\text{no}} = 1 - (1 - \exp(-v))(1 - z_i) \tag{39}$$

This yields the same expression for $\kappa_i$ in terms of $z_i$ as a direct algebraic solution of *Equation 37*.

For moderate to large $v$, $1 - \exp(-v)$ approaches 1, so $p_{\text{no}}$ approaches $z_i$ and $\kappa_i$ approaches $1 + \frac{\ln(z_i)}{v}$. This reflects the fact that for even moderately large $v$, it is almost always the case that $n_i > 0$: a least one virion must be neutralized to prevent infection. In those cases, $z_i$ can be interpreted as the (approximate) probability of no infection given a transmission event (i.e. as $p_{\text{no}}$).

Note also that seeded infections can also go stochastically extinct; this occurs with approximate probability $\frac{1}{\mathcal{R}_0}$. At the start of infection, if there is no antibody response and $\mathcal{R}_0$ is large (5 to 10), stochastic extinction probabilities should be low ($\frac{1}{5}$ to $\frac{1}{10}$), and equal in immune and naive hosts. We have therefore parametrized our model in terms of $z_i$, the probability that no cell is ever infected, as that probability determines to leading order the frequency with which immunity protects against detectable reinfection given challenge.

## Susceptibility models

Translating host immune histories into old antigenic variant and new antigenic variant neutralization probabilities for the analysis in *Figure 3G,H* requires a model of how susceptibility decays with antigenic distance, which we measure in terms of the typical distance between two adjacent 'antigenic clusters' (*Smith et al., 2004*). *Figure 3* shows results for two candidate models: a multiplicative

model used in a prior modeling and empirical studies of influenza evolution (**Boni et al., 2006**; **Asaduzzaman et al., 2018**), and a sigmoid model as parametrized from data on empirical HI titer and protection against infection (**Coudeville et al., 2010**).

In the multiplicative model, the probability $z(i,x)$ of no infection with variant $i$ given that the nearest variant to $i$ in the immune history is variant $x$ is given by:

$$z(i,x) = z_0 \left(\frac{z_1}{z_0}\right)^{d(i,x)} \tag{40}$$

where $z_0$ is the probability of no infection given homotypic reinfection, $d(i,x)$ is the antigenic distance in antigenic clusters between $i$ and $x$, and $z_1$ is the probability of no infection given $d(i,x) = 1$.

In the sigmoid model:

$$z(i,x) = 1 - \frac{1}{1 + e^{b(\ln(T(i,x)) - a)}} \tag{41}$$

where $a = 2.844$ and $b = 1.299$ ($\alpha$ and $\beta$ estimated in **Coudeville et al., 2010**) and $T(i,x)$ is the individual's HI titer against variant $i$ (**Coudeville et al., 2010**). To convert this into a model in terms of $z_0$ and $z_1$ we calculate the typical homotypic titer $T_0$ implied by $z_0$ and the $n$-fold-drop $D$ in titer per unit of antigenic distance implied by $z_1$, since units in antigenic space correspond to a $n$-fold reductions in HI titer for some value $n$ (**Smith et al., 2004**). We calculate $T_0$ by plugging $z_0$ and $T_0$ into **Equation 41** and solving. We calculate $D$ by plugging $z_1$ and $T_1 = T_0 D^{-1}$ into **Equation 41** and solving. We can then calculate $T(i,x)$ as:

$$T(i,x) = T_0 D^{-d(i,x)} \tag{42}$$

## Probability of bottleneck survival

With these analyses in hand, it is possible to combine the within-host and the point of transmission processes to calculate an overall probability that a new variant survives the transmission bottleneck. **Equation 36** gives the probability of bottleneck survival given $f_{\mathrm{mt}}$ and the properties of the recipient host. And given a time of transmission $t_t$, we can calculate the probability distribution of $f_{\mathrm{mt}}$ using our expressions for the CDF of successful mutation times $t_e$ (**Equation 20**) and for $f_m(t)$ (**Equations 13, 15**). To calculate the overall probability $p_{\mathrm{nv}}$, we average over the possible values of $f_{\mathrm{mt}}$, weighted by their probability:

$$p_{\mathrm{nv}} = \int_0^\infty p_{\mathrm{surv}}(f_m(t_t \mid t_e = t)) p(t_e = t)\, dt \tag{43}$$

## Within-host simulations (*Figures 1, 4*)

To evaluate the relative probabilities of replication selection and inoculation selection for antigenic novelty and to check the validity of the analytical results, we simulated $10^6$ transmissions from a naive host to a previously immune host. The transmitting host was simulated for 10 days, which given the selected model parameters is sufficient time for almost all infections to be cleared. Time of transmission was randomly drawn from that period, weighted by transmission probability, and an inoculum was drawn from the within-host virus population at that time. Variant counts in the inoculum were Poisson-distributed with probability equal to the variant frequency within the transmitting host at the time of transmission. We simulated the recipient host until the clearance of infection and found the maximum frequency of transmissible variant: that is, the variant frequencies when the transmission probability was greater than $5 \times 10^{-2}$ (probabilistic model) or when the virus population was above the transmission threshold $\theta$ (threshold model). For **Figure 3D**, we defined an infection with an emerged new antigenic variant as an infection with a maximum transmissible new variant frequency of greater than 50%.

## Transmission chain model (*Figure 5*)

To study evolution along transmission chains with mixed host immune statuses, we modeled the virus phenotype as existing in a 1-dimensional antigenic space. Host susceptibility $s_x$ to a given phenotype $x$ was $s_x = \min\{1, \min\{|x - y_i|\}\}$ for all phenotypes $y_i$ in the host's immune history.

Within-host cross immunity $c_{ij}$ between two phenotypes $y_i$ and $y_j$ was equal to $\max\{0, 1 - |y_i - y_j|\}$. When mucosal antibodies were present, protection against infection $z_x$ was equal to the strength of homotypic protection $z_{max}$ scaled by susceptibility: $z_x = (1 - s_x)z_{max}$. $\kappa_x$ was calculated from $z_x$ by $\exp(-v(1 - \kappa_x)) = z_x$. We used $z_{max} = 0.95$. Note that this puts us in the regime in which intermediately immune hosts are the best inoculation selectors (*Figure 3H*). We set $k = 25$ so that there would be protection against reinfection in the condition with an immediate recall response but without mucosal antibodies.

We then simulated a chain of infections as follows. For each inoculated host, we tracked two virus variants: an initial majority antigenic variant and an initial minority antigenic variant. If there were no minority antigenic variants in the inoculum, a new focal minority variant (representing the first antigenic variant to emerge de novo) was chosen from a Gaussian distribution with mean equal to the majority variant and a given variance, which determined the width of the mutation kernel (for results shown in *Figure 5*, we used a standard deviation of 0.08). We simulated within-host dynamics in the host according to our within-host stochastic model.

We founded each chain with an individual infected with all virions of phenotype 0, representing the current old antigenic variant.

We simulated contacts at a fixed, memoryless contact rate $\rho = 1$ contacts per day. Given contact, a transmission attempt occurred with a probability proportional to donor-host viral load, as described above. If a transmission attempt occurred, we chose a random immune history for our recipient host according to a pre-specified distribution of host immune histories. We then simulated an inoculation and, if applicable, subsequent infection, according to the within-host model described above. If the recipient host developed a transmissible infection, it became a new donor host. If not, we continued to simulate contacts and possible transmissions for the donor host until recovery. If a donor host recovered without transmitting successfully, the chain was declared extinct and a new chain was founded.

We iterated this process until the first phenotypic change event—a generated or transmitted minority phenotype becoming the new majority phenotype. We simulated 1000 such events for each model and examined the observed distribution of phenotypic changes compared to the mutation kernel.

For the results shown in *Figure 5*, we set the population distribution of immune histories as follows: 20% −0.8, 20% −0.5, 20% 0.0, and the remaining 40% of hosts naive. This qualitatively models the directional pressure that is thought to canalize virus evolution (*Bedford et al., 2012*) once a cluster has begun to circulate.

## Analytical mutation kernel shift model (*Figure 5*)

To assess the causes of the observed behavior in our transmission chain model, we also studied analytically how replication and inoculation selection determine the distribution of observed fixed antigenic changes given the mutation kernel when a host with one immune history inoculates another host with a different immune history.

We fixed a transmission time $t = 2$ days, roughly corresponding to peak within-host virus titers. For each possible new variant phenotype, we calculated $p_{nv}$ according to *Equation 43*, with parameters given by the old variant antigenic phenotype, new variant phenotype, and host immune histories. Finally, we multiplied each phenotype's survival probability by the same Gaussian mutation kernel used in the chain simulations (with mean 0 and s.d. 0.08), and normalized the result to determine the predicted distribution of surviving new variants given the mutation kernel and the differential survival probabilities for different phenotypes.

## Population-level model (*Figure 6*)

To evaluate the probability of variants being selected and proliferating during a local influenza virus epidemic, we first noted that the per-inoculation rate of new antigenic variant infections for a population with $n_s$ susceptibility classes (which can range from full susceptibility to full immunity) is:

$$\sum_{i=1}^{n_s} s_i(0)p_{surv}(\kappa_m(i), \kappa_w(i), v, b, f_{mt}) \tag{44}$$

where $\kappa_w(i)$ and $\kappa_m(i)$ are the mucosal antibody neutralization probabilities for the old antigenic

variant and the new antigenic variant associated with susceptibility class $i$, and $s_i(0) = \frac{S_i(0)}{N}$ is the initial fraction of individuals in susceptibility class $i$.

We then considered a well-mixed population with frequency-dependent transmission, where infected individuals from all susceptibility classes are equally infectious if infected. Using an existing result from epidemic theory (*Magal et al., 2018*), we calculated $R_\infty$, the average fraction of individuals who are infected if an epidemic occurs in such a population. During such an epidemic, each individual will on average be inoculated (challenged) $\Re_0 R_\infty$ times, where $\Re_0$ is the population-level basic reproduction number (*Miller, 2012*). We can then calculate the probability that a new variant transmission chain is started in an arbitrary focal individual:

$$\Re_0 R_\infty \sum_{i=1}^{n_s} s_i(0) p_{\mathrm{surv}}(\kappa_m(i), \kappa_w(i), v, b, f_{\mathrm{mt}}) \qquad (45)$$

## Sensitivity analysis (*Appendix 1—figure 3*)

We assessed the sensitivity of our results to parameter choices by re-running our simulation models with randomly generated parameter sets chosen via Latin Hypercube Sampling from across a range of biologically plausible values. *Appendix 1—figure 3* gives a summary of the results.

We simulated 50,000 infections of experienced hosts ($c_w = 1$) according to each of 10 random parameter sets. We selected parameter sets using Latin Hypercube sampling to maximize coverage of the range of interest without needing to study all possible permutations. We did this for the following bottleneck sizes: 1, 3, 10, 50.

We analyzed two cases: one in which the immune response is unrealistically early and one in which it is realistically timed. In the unrealistically early antibody response model, $t_M$ varied between $t_M = 0$ and $t_M = 1$. In the realistically-timed antibody response model, $t_M$ varied between $t_M = 2$ and $t_M = 4.5$. Other parameter ranges were shared between the two models (*Table 2*).

Discussion of sensitivity analysis results can be found in Appendix Section A8.

## Meta-analysis (*Figure 1*)

We downloaded processed variant frequencies and subject metadata from the two NGS studies of immune-competent human subjects naturally infected with A/H3N2 with known vaccination status (*Debbink et al., 2017*; *McCrone et al., 2018*) from the study Github repositories and journal websites. We independently verified that reported antigenic site and antigenic ridge amino acid substitutions were correctly indicated, determined the number of subjects with no NGS-detectable antigenic amino acid substitutions, and produced figures from the aggregated data.

**Table 2.** Sensitivity analysis parameter ranges shared between models.

| Parameter | Minimum value | Maximum value |
| --- | --- | --- |
| $t_N$ | 6 | 9 |
| $C_{\mathrm{max}}$ | $10^8$ | $10^9$ |
| $\mathcal{R}_0$ | 5 | 15 |
| $r$ | 10 | 500 |
| $\mu_{wm}$ | $0.33 \times 10^{-6}$ | $0.33 \times 10^{-4}$ |
| $d_v$ | 2 | 8 |
| $k$ | 3 | 16 |
| $c_m$ | 0.5 | 1 |
| $z_w$ | 0.70 | 0.99 |
| $z_m/z_w$ | 0.5 | 0.9 |
| $V_{50}$ | $10^7$ | $10^9$ |
| $v/b$ | 1 | 50 |

## Computational methods

For within-host model stochastic simulations, we used the Poisson Random Corrections (PRC) tau-leaping algorithm (*Hu and Li, 2009*). We used an adaptive step size; we chose step sizes according to the algorithm of *Hu and Li, 2009* to ensure that estimated next-step mean values were non-negative, with a maximum step size of 0.01 days. Variables were set to zero if events performed during a timestep would have reduced the variable to a negative value. For the sterilizing immunity simulations in *Figure 2*, we used a smaller maximum step size of 0.001 days in recipient hosts to better handle mutation dynamics involving very small numbers of replicating virions.

We obtained numerical solutions of equations, including systems of differential equations and final size equations, in Python using solvers provided with SciPy (*Jones et al., 2001*).

## Data and materials availability

All code, data, and other materials needed to reproduce the analysis in this paper are provided online on the project Github repository: https://github.com/dylanhmorris/asynchrony-influenza (*Morris, 2020*; copy archived at swh:1:rev:5a9796fa3ab7b8a86aeccd7c9353542f9409e215).

They are also available on OSF: https://doi.org/10.17605/OSF.IO/jdsbp.

Output data generated by stochastic simulations, within-host NGS meta-analysis, and phylogenetic analyses are archived on OSF: https://doi.org/10.17605/OSF.IO/jdsbp.

# Acknowledgements

We thank Daniel B Cooney, Andrea L Graham, Joseph Gibson, Katelyn M Gostic, Alvin X Han, Chadi M Saad-Roy, and Edward C Schrom for helpful discussions. We thank Christopher J Illingworth, Daniel B Weissman, and an anonymous reviewer for helpful comments on prior versions of this manuscript.

We thank the GISAID Initiative and the influenza surveillance and research groups who openly shared the genetic sequence data used in this work (a table with accession numbers and associated labs is available online in the project materials and data repositories on Github and OSF, see above for links).

# Additional information

### Competing interests

Richard A Neher: Reviewing editor, *eLife*. The other authors declare that no competing interests exist.

### Funding

| Funder | Grant reference number | Author |
| --- | --- | --- |
| H2020 European Research Council | Naviflu:818353 | Colin A Russell |

The funders had no role in study design, data collection and interpretation, or the decision to submit the work for publication.

### Author contributions

Dylan H Morris, Conceptualization, Data curation, Software, Formal analysis, Validation, Investigation, Visualization, Methodology, Writing - original draft, Writing - review and editing; Velislava N Petrova, Conceptualization, Investigation, Writing - original draft, Writing - review and editing; Fernando W Rossine, Formal analysis, Validation, Investigation, Writing - review and editing; Edyth Parker, Formal analysis, Investigation, Visualization, Methodology, Writing - original draft, Writing - review and editing; Bryan T Grenfell, Investigation, Writing - review and editing; Richard A Neher, Formal analysis, Investigation, Methodology, Writing - original draft, Writing - review and editing; Simon A Levin, Supervision, Investigation, Writing - review and editing; Colin A Russell,

Conceptualization, Supervision, Funding acquisition, Investigation, Visualization, Methodology, Writing - original draft, Project administration, Writing - review and editing

Author ORCIDs
Dylan H Morris ⓘ https://orcid.org/0000-0002-3655-406X
Edyth Parker ⓘ https://orcid.org/0000-0001-8312-1446
Bryan T Grenfell ⓘ http://orcid.org/0000-0003-3227-5909
Richard A Neher ⓘ http://orcid.org/0000-0003-2525-1407
Colin A Russell ⓘ https://orcid.org/0000-0002-2113-162X

Decision letter and Author response
Decision letter https://doi.org/10.7554/eLife.62105.sa1
Author response https://doi.org/10.7554/eLife.62105.sa2

## Additional files

### Supplementary files
• Transparent reporting form

### Data availability
All data used in this study are specifically listed in the appendix. No new primary data was generated in this study.

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

## Appendix 1

### A1 Summary

This appendix contains additional information for 'Asynchrony between virus diversity and antibody selection limits influenza virus evolution'.

We provide a detailed review of virological and immunological evidence supporting our model of the interaction between influenza viruses and the (human) host immune system (Section A2). We give a more detailed mathematical analysis of our within-host model, deriving the approximate and exact analytical results that are used in the main text, and provide intuition to explain model behavior and limitations (Section A3). We do the same for our model of the point of transmission, and assess which hosts are most likely to act as inoculation selectors and which mutants are most likely to be inoculation selected (Section A4).

We next discuss parameter uncertainties and report the results of a sensitivity analysis for our within-host models (Section A5). We report an additional empirical analysis showing that new variants often emerge polyphyletically: they appear simultaneously on multiple branches of the influenza virus phylogeny. We discuss how this is more easily explained in light of our model than in light of previous models (Section A6).

We then review these prior studies and models of within-host influenza virus antigenic evolution and general pathogen immune escape in depth. We argue that they are insufficient to explain within-host influenza virus antigenic evolution and that inoculation selection accompanied by a realistically-timed antibody response is the most likely competing hypothesis (Section A7). We elaborate on the conclusions that can be drawn from our study and its implications for influenza virus control and for future basic research (Section A8). We conclude by providing complete step-by-step mathematical derivations of lemmas and approximations from elsewhere in the text (Section A9).

### A2 Immunological underpinnings of the inoculation selection model

In this section, we review relevant immunology to establish the biological realities that we built our model to capture. We also discuss some unmodeled biological complexities, and why they are unlikely to change our qualitative conclusions.

#### A2.1 Strong selection at the point of inoculation

The establishment of successful influenza virus infection requires access to respiratory epithelial cells covered by a mucosal layer, which acts as a protective barrier for virus entry. Sialic acid residues present in the mucus act as decoys for virions and the enzymatic activity of the influenza virus neuraminidase (NA) protein is essential for penetration of the mucosal layer (*Matrosovich et al., 2004*). The mucosal layer does not act only as a mechanical barrier. Secretory IgA (sIgA) mucosal antibodies specific to influenza virus proteins can apply virus-specific neutralization by immobilizing virions and slowing down the process of virus infection (*Wang et al., 2017*). Depending on their specificity, sIgA can either play a role in reduction of total virus population size or in providing specific competitive advantage to mutant viruses. Antibodies against the influenza virus NA protein will slow down the virus passage through the mucosal barrier irrespective of the HA antigenic phenotype of the newly infecting virus. By contrast, anti-HA antibodies are essential for inoculation selection because they recognise previously encountered antigenic variants. This provides a competitive advantage to new, distinct antigenic variants. The new variants with strongest competitive advantage are those with large antigenic effects; they have the lowest probability to be neutralized by any cross-reactive anti-HA sIgAs.

#### A2.2 Weak selection during exponential virus growth

Virions that successfully cross the mucosal barrier can infect epithelial cells in the respiratory tract. Upon initiation of replication in host cells, newly generated virions can be subjected to replication selection via virus-specific antibodies. Replication selection can take place only if influenza virus-specific antibodies are present in mucosa-associated lymphoid tissue (MALT) at the time of virus replication.

Influenza virus-specific antibodies in the MALT are secreted by local plasmablast populations, generated following activation of B naive cells or B memory cells. The exposure history of the host determines the timing and the immunological trajectory for the generation of influenza virus-specific antibodies. In individuals without previous exposure to influenza virus, newly generated virions are recognised by naive B cells which undergo rounds of affinity maturation and somatic hypermutation to produce highly-specific antibodies to the virus. This process typically occurs in germinal centers, requires T-cell help, and takes around 7 days for production of highly-specific antibodies (*Wrammert et al., 2008*).

In individuals with previous exposure history to influenza viruses, where the newly generated influenza virions are antigenically matched or have some degree of cross-reactivity to previously encountered variants, the generation of influenza virus-specific antibodies mainly results from B memory recall response. The activation of pre-existing memory pools enables the immune system to respond quicker to previously encountered antigens without the need to recruit naive B cell populations. The generation of antigen-specific antibodies from recalled B memory pools does not typically start until 3–5 days post infection (*Coro et al., 2006*; *Lam and Baumgarth, 2019*).

The B naive and B memory cells generating serological responses to influenza viruses are referred to as B-2 cells. The activation of these cell types depends on the recognition of virus protein via the B-cell receptor and sufficient affinity of binding to trigger stimulatory signal. Depending on their specificity to the recognised virus antigen, the antibodies produced by these cell types have the potential to apply replication selection to any generated new antigenic variant viruses and to old variant viruses during the course of infection. Antibodies against influenza viruses can also be generated from the so-called B-1 cells, which do not require specific recognition of virus antigen and are activated by innate mechanisms (*Lam and Baumgarth, 2019*).

B-1 cells produce natural IgM antibodies with low affinity which constitute the earliest humoral response generated in the first 48–72 hr of infection (*Lam and Baumgarth, 2019*). As the activation of B-1 cells is independent of the specific recognition of influenza virus virions, the natural IgM antibodies act as a non-specific limiting factor for virus population sizes and cannot apply replication selection to particular virus antigenic variants.

The peak of influenza virus replication occurs in the first 24–48 hr post infection (*Hadjichrysanthou et al., 2016*). Despite the presence of several immunological routes for generation of influenza virus-specific antibodies, the timing of the serological immune response is always later than the time of peak of virus titer (at least in typical infections of otherwise healthy individuals), thus limiting the opportunity for replication selection during the course of infection.

## A2.3 Selection during exit of an infected host

The design of the inoculation selection model focuses on selection applied by the mucosal layer upon the entry of influenza virus in naïve or previously exposed hosts. We do not take into account the potential bottleneck applied during exit of viruses from the respiratory tract of an infected host. Selection of newly generated virions at the point of exit of an infected host is technically possible as the virions released from infected epithelial cells need to cross the mucosal barrier again to reach the lumen of the respiratory tract. Such selection upon virus exit would have the same directional effect as inoculation selection; it would provide a competitive advantage for mutant viruses with large antigenic effects. The potential strength of such selection depends heavily the how many sIgA antibodies in the transmitting host's mucosa are free to neutralize the virions passing through; this quantity is unknown. Incorporating such mucosal selection upon virus exit thus requires better knowledge of the quantities of sIgA antibodies in given volume of mucus and what proportion of the available sIgA antibodies are engaged with antigen when an infected host transmits. Such data is not currently available and would require well-designed cellular culture models which take into account the role of mucosal layer in the process of virus infection and release of newly generated viruses.

## A2.4 Nature and timing of previous exposure and effects on inoculation selection

According to our model, the strongest inoculation selection can take place in infection of previously exposed individuals and the highest probability of antigenic diversification occurs in populations

with an intermediate proportion of immune hosts. In deriving this result, we have assumed homogeneous and static quantities of sIgA antibodies among previously exposed individuals. This is an oversimplification. There is substantial variation in level of sIgA antibodies across individuals depending on the nature of their previous exposures (through vaccination or natural infection). For example, intranasal inoculation with influenza virus via live attenuated influenza virus vaccines leads to generation of better mucosal immunity (*Hoft et al., 2017*) than intramuscular administration of trivalent inactivated vaccines and thus previously exposed individuals are likely to vary in the degree of inoculation selection applied during homotypic infections depending on the route of administration of previous vaccinations.

The model also does not account for waning of sIgA antibody titers in the months and years following each exposure. The waning of sIgA titers is likely to be substantial in absence of re-exposure but the waning process should only decrease the efficiency of inoculation selection and make the accumulation of population level selection pressure—and thus the appearance of observable new antigenic variants—rarer.

The dynamics of mucosal immunity with age are also likely to have impact on the selector phenotype of previously exposed hosts. In older individuals, multiple previous exposures will eventually lead to preferential activation of recall responses for antibody production to new variants as a result of original antigenic sin (*Davenport and Hennessy, 1956*), immune backboosting (*Fonville et al., 2014*), or antigenic seniority (*Lessler et al., 2012*).

Even if the recall responses result in backboosting and higher levels of sIgA, these antibodies are unlikely to be well-matched to the new variant or succeeding variants, as continuous re-exposure favors responses to conserved epitopes (*Krammer and Palese, 2019*; *Krammer, 2019*) rather than receptor-binding site epitopes suspected to be most important for immune escape.

As a result of the limited ability to generate novel immune responses and the phenomenon of antigenic seniority, an elderly individual who has been exposed multiple times in the past is likely to exert very weak inoculation selection to newly infecting viruses. By contrast, a young person recently exposed to influenza virus is more likely to act as a strong selector, due to the availability of mucosal sIgA antibodies and the more limited impact of original antigenic sin. However, due to the acute nature of influenza virus infection, very young children are likely to require more than one exposure before they begin to develop highly-specific antibody responses to influenza virus infection able to exert inoculation selection (*Neuzil et al., 2006*).

Most current efforts for characterizing humoral immunity to influenza virus infections are focused on antibody responses in the serum. However, there is limited understanding of the relationship between antibodies in the serum measured via HI and the levels of mucosal antibodies able to exert inoculation selection. Better characterization of the dynamics of mucosal immunity with age and across individuals is needed to understand the implications of inoculation selection to influenza virus evolution depending on the host population structure.

## A3 Detailed within-host model analysis

Here we analyze the within-host model in depth, and find expressions for within-host variant frequencies over time, the probability distribution for the time of first successful mutation to a new variant, and an approximate probability of replication selection to a given frequency by a given time.

### A3.1 Expression for the within-host new variant frequency

In our minimal model, competition between new variant and old variant viruses obeys a simple replicator equation.

The within-host frequency of new variant virus is $f_m(t) = \frac{V_m(t)}{V_m(t) + V_w(t)} = \frac{V_m(t)}{V_{\text{tot}}(t)}$. Its rate of change is $\frac{\mathrm{d}f_m}{\mathrm{d}t}$.

We observe that $\frac{\mathrm{d}V_i}{\mathrm{d}t}$ is of the form $\frac{\mathrm{d}V_i}{\mathrm{d}t} = V_i[g_i(t) - d_i(t)]$ for both the new variant and the old variant, where, neglecting mutation, $g_i(t) = r_i \beta C(t)$ and $d_i(t)$ is a (possibly time-varying) neutralization or decay rate. Differentiating $f_m(t)$ with respect to time demonstrates that the frequency of the new variant over time is governed by the following replicator equation (see Section A9 for an explicit derivation):

$$\frac{\mathrm{d}f_m}{\mathrm{d}t} = f_m(1 - f_m)([g_m(t) - g_w(t)] - [d_m(t) - d_w(t)]) \tag{A1}$$

If the new variant is neutral in the absence of antibody-mediated selection, $r_w = r_m = r$, and therefore $g_m(t) = g_w(t)$. It follows that despite changing target cell populations and virion population sizes, the variants compete in a manner independent of $V_{\text{tot}}$, provided each $d_i(t)$ is independent of $V_{\text{tot}}$. We have competition only through the respective death rates of old variant and new variant virions. These may be affected by antibody-mediated neutralization of virions.

If the fitness difference $\delta(t) = [g_w(t) - g_m(t)] - [d_m(t) - d_w(t)]$ is a constant $\delta$, this differential equation has a closed-form solution:

$$f_m(t) = \frac{e^{\delta t}}{e^{\delta t} + f_0^{-1} - 1} \tag{A2}$$

where $f_0 = f_m(0)$

If $\delta = 0$, $f_m(t)$ is constant and equal to $f_0$, as expected. If $\delta(t)$ is piecewise-constant, we can apply the constant-$\delta$ solution iteratively to find the new variant frequency.

It is therefore straightforward to apply this expression to the within-host model with binary immunity by evaluating different immunity regimes piece-wise.

We consider the case of a host who is experienced to the old variant ($E_w = 1$) but naive to the new variant ($E_m = 0$).

If first appears at a time $t_e \geq 0$ post-infection at an initial frequency $f_m(t_e)$, then if $t_e < t_M$:

$$f_m(t) = \begin{cases} 0 & t < t_e \\ f_m(t_e) & t_e \leq t < t_M \\ \dfrac{e^{\delta(t - t_M)}}{e^{\delta(t - t_M)} + f_m(t_e)^{-1} - 1} & t_M \leq t \end{cases} \tag{A3}$$

if $t_e > t_M$:

$$f_m(t) = \begin{cases} 0 & t < t_e \\ \dfrac{e^{\delta(t - t_e)}}{e^{\delta(t - t_e)} + f_m(t_e)^{-1} - 1} & t_e \leq t \end{cases} \tag{A4}$$

where $\delta = k(c_w - c_m)$ is the fitness difference between the variants due to the recall adaptive response.

## A3.2 New variant frequency over time given ongoing de novo mutation

Thus far we have neglected mutation the first mutation of interest (i.e. the first that produces a new variant lineage that evades stochastic extinction). In fact, with symmetric mutation between the two types of interest, we have:

$$\begin{aligned} \frac{dV_w}{dt} &= V_w[(1 - \mu)g_w(t) - d_w(t)] + \mu g_m(t) V_m \\ \frac{dV_m}{dt} &= V_m[(1 - \mu)g_m(t) - d_m(t)] + \mu g_w(t) V_w \end{aligned} \tag{A5}$$

The rate of fitness-irrelevant mutation is assumed to be equal for the two types, and to preserve antigenic type ($w$ or $m$). The rate of lethal or deleterious mutants is assumed equal for the two types and is captured in $g$.

It can be shown (see Section A9) that the equation for $\frac{df_m}{dt}$ in this case is:

$$\frac{df_m}{dt} = f_m(1 - f_m)(\alpha_m - \alpha_w) + \mu(g_w(t) - 2g_w(t)f_m + g_w(t)f_m^2 - g_m(t)f_m^2) \tag{A6}$$

When $g_w(t) = g_m(t) = g(t)$:

$$\frac{df_m}{dt} = f_m(1 - f_m)(d_m(t) - d_w(t)) + \mu g(t)(1 - 2f_m) \tag{A7}$$

When $f_m \ll 0.5$, the net effect of symmetric mutation on $f_m$ is to increase it at rate $\approx \mu$ (note that

the symmetric mutation term becomes zero at $f_m = 1/2$). In other words, it is roughly equivalent to a case with no back mutation at all:

$$\begin{aligned}
\frac{dV_w}{dt} &= V_w[(1-\mu)g_w(t) - d_w(t)] \\
\frac{dV_m}{dt} &= V_m[g_m(t) - d_m(t)] + \mu g_w(t)V_w
\end{aligned} \tag{A8}$$

in that case, we have:

$$\frac{df_m}{dt} = f_m(1-f_m)([g_m(t) - (1-\mu)g_w(t)] - [d_m(t) - d_w(t)]) + \mu g_w(t)(1 - 2f_m + f_m^2) \tag{A9}$$

In either case, we mainly study a situation in which $g_w(t) = g_m(t) = g(t)$ at all times (though note that $g(t)$ varies in time), $d_w(t) = d_m(t) = d(t) = d_v$ prior to the onset of the adaptive immune response, and $d_w(t) = d_v + k$, $d_m(t) = d_v + c_m k$ after the onset of the adaptive immune response.

If $f_m$ is large relative to $\mu$, the mutant growth term $V_m g(t)$ is large relative to the de novo mutation term $\mu g(t)V_w$. We can see this from the fact that $V_m = f_m V_{\text{tot}} \geq f_m V_w$, with equality if and only if $f_m = 0$. It follows that if $f_m > \mu > 0$, then:

$$g(t)V_m > g(t)f_m V_w > g(t)\mu V_w$$

But if the first mutation to produce a new variant is sufficiently late and there is little or no positive selection, we do not necessarily have $f_m \gg \mu$, so ongoing mutation cannot be ignored in the dynamics of $f_m$. In that case, we can consider either the $\mu g(t)(1 - 2f_m + f_m^2)$ term in *Equation A9* or the $\mu g(t)(1 - 2f_m)$ term in *Equation A7*. Denote the new variant frequency at the time of first mutation $t_e$ by $f_0 = f_m(t_e)$. We can approximate $g(t) \approx g_0 = g(0) = \mathcal{R}_0 d_v$ early in infection, since for small $t$, $C(t) \approx C_{\max}$ and therefore $g(t) = r\beta C(t) \approx g_0$.

With this approximation for $g(t)$, the one-way mutation case has an analytical solution for $\delta \neq 0$:

$$\begin{aligned}
f_m(t) &\approx \frac{A\exp(\delta t) + \mu g_0}{A\exp(\delta t) + \mu g_0 - \delta} \\
A &= \frac{f_0}{1 - f_0}(\mu g_0 - \delta) - \frac{\mu g_0}{f_0}
\end{aligned} \tag{A10}$$

And when $\delta = 0$:

$$f_m(t) \approx \frac{\frac{1}{1-f_0} + \mu g_0 t - 1}{\frac{1}{1-f_0} + \mu g_0 t} \tag{A11}$$

## A3.3 When ongoing mutation matters

$f_m$ must be small for ongoing mutation to matter; it ceases to matter when $f_m \gg \mu g(t)$, and $\mu g(t)$ is small by assumption. It follows that in the absence of a fitness difference between the types:

$$\frac{df_m}{dt} \approx \mu g_0 \tag{A12}$$

and therefore:

$$f_m(t) \approx f_0 + \mu g_0 t \tag{A13}$$

We can also see this by inspecting the other two expressions for $f_m(t)$ and seeing that they are quasi-linear for small $\mu$ and small $t$. This further confirms that it is not crucial to decide whether symmetric or one-way mutation is more realistic, as they will be functionally the same from the point of view of practical mutant frequencies in the absence of positive selection.

These approximations break down once $C(t) \ll C_{\max}$ and therefore $g_w(t) \ll g_0$; fewer wild-type replications are occurring, and rate of ongoing mutation therefore falls. So a reasonable approximation for the maximum value of $f_m(t)$ in the absence of positive selection is given by $f_m(t_{\text{peak}})$. For a

mutation rate of $0.33 \times 10^{-5}$, typical values of $f_m(t_{\text{peak}})$ without selection range from $1 \times 10^{-5}$ to $2 \times 10^{-4}$, depending on $\mathcal{R}_0$ and $t_{\text{peak}}$.

We use the expression in *Equation A10* to plot the analytical predicted mutant frequencies shown in *Figure 1* and to calculate the transmission frequencies for the analytical model of observed phenotypes in *Figure 1*.

### A3.4 Consequences of ongoing de novo mutation

Ongoing mutation enhances possibilities for both replication and inoculation selection. It truncates the left tail of the distribution of $f_m(t_M)$ and the distribution of $f_{\text{mt}}$ (the new variant frequency at the time of transmission). Whenever $t_M$ is sufficiently late that we have reached $V_w \ll \frac{1}{\mu}$ before $t_M$, $f_m(t_M)$ should be on the order of the inverse mutation rate, or larger.

The consequence is that probabilities of replication selection and inoculation selection should both be somewhat higher than those estimated based on the time to the first non-extinct de novo mutant lineage, as in *Figure 1*. This further strengthens the case for the importance of an adaptive response at 48 hr or later in explaining the absence of replication selection.

### A3.5 Replication selection is helped when individual infected cells are more productive

For a fixed $\mathcal{R}_0$, replication selection becomes more likely if the virions produced per infected cell $r$ becomes large—that is, if every infected cell produces more individual virions.

For a given $d_v$, and $f$, a virus can achieve a large $\mathcal{R}_0$ either by having a high cell productivity $r$ or by having a high cell infection rate $\beta$.

The expected number of virus replications per lost target cell is given by $\frac{\dot{V}^+}{\dot{C}^-} = \frac{r\beta C\bar{V}}{\ell\beta C V} = \frac{r}{\ell}$. The infection peaks and begins to decline when $\frac{dV}{dt} = 0$, which occurs when $g(t) = d(t)$, or $\frac{C_{\max}}{C} = \mathcal{R}_0$, provided $d(t)$ is fixed. It follows that $C_{\text{peak}} = \frac{C_{\max}}{\mathcal{R}_0}$. So the virus peaks once $C_{\max}\left(1 - \frac{1}{\mathcal{R}_0}\right)$ target cells have been consumed (assuming negligible target cell replenishment during the timecourse of infection), and $\frac{r}{\ell}C_{\max}\left(1 - \frac{1}{\mathcal{R}_0}\right)$ viral replication events have occurred (this is a slight overestimate due to target cell depletion).

Fixing $d_v$ and $\mathcal{R}_0$, as $r \to \infty$, $\beta \to 0$ and the number of viral replication events before peak viral load goes to $\infty$. Since each generated mutant survives stochastic extinction roughly proportional to $1/\mathcal{R}_0$, these earlier mutants also are lost stochastically less frequently. In other words, a mutant that survives stochastic extinction is generated with certainty well before the infection peaks and can spend arbitrarily long under selection. Conversely, as $r \to 0$, $\beta \to \infty$, the number of replication events that occur before the infection peaks goes to 0, and replication selection becomes impossible.

A virus of a given $\mathcal{R}_0$ that takes a shotgun approach of producing many not-especially-infectious virions is more likely to result in the proliferation of a mutant of interest under replication selection than one that produces a small number of highly infectious virions.

In other words, the relatively large infected cell productivity of influenza viruses—perhaps as large as hundreds or thousands of viable virions per cell (*Frensing et al., 2016*)—makes potential replication selection more efficient. This in turn makes the absence of observed replication selection harder to explain unless another mechanism can be invoked, such as inoculation selection with replication selection only occurring 2–3 days post-infection.

### A3.6 Replication selection in the presence of sterilizing immunity

One special case bears mentioning, because it corresponds to multiple existing models (*Luo et al., 2012*; *Volkov et al., 2010*; *Kennedy and Read, 2017*) and may be relevant for other systems. If there is an immediate recall response ($t_M = 0$), a founding population of size $b$, and $k$ is large enough such that $\mathcal{R}^w(t = 0) < 1$ (the virus population is initially declining, in expectation), then the infection will go extinct after generating $q$ new virions for some finite $q$.

Each virion has a probability $\mu$ of being an escape mutant with $\mathcal{R}^m(0) > 1$ and each mutant independently survives stochastic extinction with probability $p_{\text{sse}} = (1 - 1/\mathcal{R}^m(0))$ (per the theory of

supercritical branching processes). The conditional probability of replication selection given $q$ is therefore:

$$p_{\mathrm{repl}}(q) = 1 - (1 - \mu p_{\mathrm{sse}})^q \tag{A14}$$

if $\mu p_{\mathrm{sse}}$ is small, this is approximately equal to:

$$p_{\mathrm{repl}}(q) \approx 1 - e^{-q\mu p_{\mathrm{sse}}} \approx q\mu p_{\mathrm{sse}} \tag{A15}$$

The approximations use the Poisson approximation to the binomial and the approximation $e^x \approx 1 + x$ when $|x| \ll 1$.

The unconditional probability of replication selection $p_{\mathrm{repl}}$ is the expectation in $q$ of $p_{\mathrm{repl}}(q)$ : $E_q(p_{\mathrm{repl}})$. But since $p_{\mathrm{repl}}$ is approximately linear in $q$, the linearity of expectation implies:

$$p_{\mathrm{repl}} \approx E_q(q\mu p_{\mathrm{sse}}) = \bar{q}\mu p_{\mathrm{sse}}$$

where $\bar{q}$ is the expected value of $q$.

Given $\mathcal{R}^w(0) < 1$, branching process theory predicts that the virus will produce on average $\bar{q} = \frac{b}{1-\mathcal{R}^w(0)} - b$ new copies before the infection is cleared (expected total length of $b$ independent subcritical branching processes, each with expected length $\frac{1}{1-\mathcal{R}^w(0)}$, minus the initial virions present [**Ball et al., 2016**]). This can be simplified to $\bar{q} = \frac{b\mathcal{R}^w(0)}{1-\mathcal{R}^w(0)}$.

So we have

$$p_{\mathrm{repl}} \approx \bar{q}\mu p_{\mathrm{sse}} = \frac{\mathcal{R}^w(0)}{1-\mathcal{R}^w(0)} b\mu p_{\mathrm{sse}} \tag{A16}$$

Note that *Iwasa et al., 2004* have previously derived this approximate result for the probability that a subcritical replicator mutates to a supercritical one prior to going extinct in the context of analyzing tumor cell mutations. They use a more rigorous generating function approach.

## A4 Point of transmission analysis

### A4.1 Distinction between selection and drift at the point of transmission

In our model, mucosal antibodies acting at the point of transmission provide a mechanism for population level selection pressure: some protection of experienced hosts against infection with old variant virus, and worse protection against infection with new variant virus. In particular, it provides a mechanism that produces selection pressure for new antigenic variants while predicting that reinfections with old variant viruses—without observable new variant viruses—should still be observed. Prior models (see Section A7) predict that in fully immune hosts, any observable reinfections will have new antigenic variant viruses at consensus.

Antibodies at the point of transmission appear to play a key role in population-level influenza virus evolution: they produce the selection pressure that allows new variants to spread more reliably and thus rapidly from host to host than old variants. But they may play a second role as well: promoting of new variants in frequency from the low frequencies at which they are typically generated to high frequencies at which they can be observed and reliably transmitted onward. This inoculation selection takes on the role of previously held by replication selection in explaining why inoculations of experienced hosts might produce new variant infections at a higher rate (per inoculation) than inoculations of naive hosts.

### A4.2 The inoculation selection paradigm

As noted in the main text, new variants can reach high within-host frequencies through founder effects. These events are both possible and rare due to influenza's tight transmission bottleneck (*McCrone et al., 2018*; *Xue and Bloom, 2019*).

Whether this process is purely neutral or stochastically selective depends on whether the recipient host is naive, and if not how well they neutralize old variant versus new variant virions.

As mentioned in the Methods, in a fully naive host with large $v$ and small transmitting host new variant frequency, the sampling process approximates a low-probability binomial or a low-frequency Poisson:

$$p_{\text{drift}} \approx 1 - e^{-f_{\text{mt}}b} \tag{A17}$$

When $f_{\text{mt}} \ll 1$, this is approximately equal to $bf_{\text{mt}}$

As noted in the main text Materials and methods, mucosal antibody neutralization can reduce the new variant's survival probability relative to this neutral case (inoculation pruning) or increase it (inoculation promotion), depending upon parameters.

There can be inoculation pruning even when the new variant is more fit than the old variant (i.e. neutralized with lower probability). But sufficient subsequent bottlenecking after sIgA neutralization can create an inoculation promotion effect.

Whether inoculation selection produces pruning or promotion depends on the ratio of $v/b$, and on the mutant's probability of avoiding neutralization $1 - \kappa_m$ (main text **Figures 4F,5**, **Appendix 1—figure 1**, and Section A4.5 below).

## A4.3 Expression for and properties of the new variant survival probability

As shown in the Materials and methods, the probability that a new variant survives the cell infection bottleneck of size $b$ is well approximated for small $f_{\text{mt}}$ by:

$$p_{\text{surv}}(\kappa_m, \kappa_w, v, b, f_{\text{mt}}) \approx p_{\text{inoc}}(\kappa_m, v, f_{\text{mt}}) * p_{\text{cib}}(\kappa_w, v, b) \tag{A18}$$

$p_{\text{inoc}}$ is the probability that at least one mutant survives the sIgA bottleneck:

$$p_{\text{inoc}}(\kappa_m, v, f_{\text{mt}}) = 1 - e^{-vf_{\text{mt}}(1-\kappa_m)} \tag{A19}$$

Note that when $f_{\text{mt}} \ll 1, p_{\text{inoc}} \approx vf_{\text{mt}}(1 - \kappa_m)$

$p_{\text{cib}}(\kappa_w, v, b)$ is the probability that a mutant that survives the sIgA bottleneck is not lost at the final bottleneck (as given in **Equation 35** above):

$$p_{\text{cib}}(\kappa_w, v, b) = (1 - e^{-\bar{w}})\frac{b}{\bar{w}} + e^{-\bar{w}} \sum_{j=0}^{b-1} \frac{\bar{w}^j}{j!}\left(1 - \frac{b}{j+1}\right)$$

where $\bar{w} = v(1 - f_{\text{mt}})(1 - \kappa_w)$. If $b = 1$, this expression reduces to the first term.

### 4.3.1 Properties of these probabilities

These probabilities have several intuitive and useful properties:

- $p_{\text{cib}} < 1$ if $\kappa_w < 1$ and $f_{\text{mt}} < 1$, and $p_{\text{cib}} = 1$ if $\kappa_w = 1$ or $f_{\text{mt}} = 1$. That is, surviving the cell infection bottleneck is certain only if there is no competition. See Section A9.6 for derivation.

- The correction term in $p_{\text{cib}}$, $e^{-\bar{w}} \sum_{j=0}^{b-1} \frac{\bar{w}^j}{j!}\left(1 - \frac{b}{j+1}\right)$, is always negative. Each term except the last in the summation is negative, since $b > j + 1$ for $j < b - 1$. The last term is zero (the last term is included for notational reasons, so that the formula is correct with $b = 1$). This implies that the first term is always $\geq p_{\text{cib}}$, with equality if and only f $b = 1$.

- If $\kappa_m = \sigma\kappa_w$, then $p_{\text{inoc}}$ is decreasing in $\kappa_w$ for $0 < \sigma \leq 1$. This is as expected: the more likely the new variant is to be neutralized, the smaller $p_{\text{inoc}}$ becomes. It suffices to show that $\frac{\mathrm{d}p_{\text{inoc}}}{\mathrm{d}\kappa_w} < 0$ for $0 < \sigma \leq 1$. We find:

$$\frac{\mathrm{d}p_{\text{inoc}}}{\mathrm{d}\kappa_w} = -e^{-vf_{\text{mt}}(1-\sigma\kappa_w)}(vf_{\text{mt}}\sigma) \tag{A20}$$

Since $f_{\text{mt}}, v > 0$, this is negative whenever $\sigma > 0$, and 0 if $\sigma$ is 0.

It is sometimes useful to write this as:

$$\frac{\mathrm{d}p_{\mathrm{inoc}}}{\mathrm{d}\kappa_w} = vf_{\mathrm{mt}}\sigma(p_{\mathrm{inoc}} - 1) \tag{A21}$$

- $p_{\mathrm{cib}}$ is decreasing in $\bar{w}$ (see Section A9.7 for derivation). This is intuitive: the less competition in the form of (expected numbers of) un-neutralized old variant virions, the more likely a surviving mutant is to pass through the cell infection bottleneck.
- $p_{\mathrm{cib}}$ is increasing in $\kappa_w$ and $f_{\mathrm{mt}}$ and decreasing in $v$. Since $v$, $1 - \kappa_w$, and $1 - f_{\mathrm{mt}}$ are all $\geq 0$, it is clear that $\bar{w}$ is decreasing in $\kappa_w$ and $f_{\mathrm{mt}}$ and increasing in $v$. It follows that $p_{\mathrm{cib}}$ is increasing in $\kappa_w$ and $f_{\mathrm{mt}}$ and decreasing in $v$. In all cases, this reflects the parameters' effect on the mean amount of competition for the new variant from old variant virions for the cell infection bottleneck.
- $p_{\mathrm{inoc}}$ is increasing in $f_{\mathrm{mt}}$, since a larger $f_{\mathrm{mt}}$ implies an $vf_{\mathrm{mt}}(1 - \kappa_m)$ term that is larger in absolute value, so $e^{-vf_{\mathrm{mt}}(1-\kappa_m)}$ becomes smaller and $p_{\mathrm{inoc}} = 1 - e^{-vf_{\mathrm{mt}}(1-\kappa_m)}$ becomes larger.

## A4.4 Mutant survival probability increases as transmitted mutant frequency increases

For given values of the other parameters, larger $f_{\mathrm{mt}}$ implies larger $p_{\mathrm{surv}}$. Since both $p_{\mathrm{inoc}}$ and $p_{\mathrm{cib}}$ are increasing in $f_{\mathrm{mt}}$ and are both always positive, it follows that $p_{\mathrm{surv}} = p_{\mathrm{inoc}}p_{\mathrm{cib}}$ is also increasing in $f_{\mathrm{mt}}$. This makes intuitive sense: all else equal, higher frequency mutants always have a better chance of surviving at the point of transmission, regardless of the effects of the sIgA bottleneck.

## A4.5 The relative importance of selection and drift at the point of inoculation

Consider $p_{\mathrm{surv}}/p_{\mathrm{drift}}$. This ratio quantifies the degree to which sIgA neutralization at the point of transmission facilitates (or hinders) mutant survival of the final bottleneck relative to a purely neutral founder effect.

For realistically small $b$ and $f_{\mathrm{mt}}$, we can use the approximations $p_{\mathrm{drift}} \approx bf_{\mathrm{mt}}$ and $p_{\mathrm{inoc}} \approx vf_{\mathrm{mt}}(1 - \kappa_m)$ Then we have

$$p_{\mathrm{surv}}/p_{\mathrm{drift}} = \frac{p_{\mathrm{inoc}}p_{\mathrm{cib}}}{p_{\mathrm{drift}}} \approx \frac{v}{b}(1 - \kappa_m)p_{\mathrm{cib}}(\kappa_w, v, b) \tag{A22}$$

Several intuitive results follow:

- If we hold $\kappa_w$ constant, increasing $\kappa_m$ makes inoculation selection less effective relative to drift, because the inoculated new variant is at greater risk of neutralization.
- Conversely, if we hold $\kappa_m$ constant, increasing $\kappa_w$ makes inoculation selection more effective relative to drift (this follows from the fact that $p_{\mathrm{cib}}$ is increasing in $\kappa_w$, see Section A4.3.1 above).
- Since $p_{\mathrm{cib}}$ is increasing in $f_{\mathrm{mt}}$ (Section A4.3.1), $p_{\mathrm{surv}}/p_{\mathrm{drift}}$ is increasing in $f_{\mathrm{mt}}$. That is, inoculation selection is more efficient relative to drift when promoting higher frequency mutants, all else equal.
- $p_{\mathrm{surv}}/p_{\mathrm{drift}}$ is decreasing in $b$ (see Section A9.9 for a derivation). In other words, when the cell infection bottleneck is wider, inoculation selection is less important relative to drift. The large bottleneck gives the mutant a reasonably good chance of surviving even in the absence of any neutralization of competing old variant virions.

For a bottleneck of size $b = 1$, we can also see that inoculation selection becomes more efficient relative to drift as $v$ increases, since:

$$
\begin{aligned}
p_{\mathrm{surv}}/p_{\mathrm{drift}} \quad &\approx \frac{v}{b}(1 - \kappa_m)p_{\mathrm{cib}} \\
&= v(1 - \kappa_m)\frac{1 - \exp(-v(1 - f_{\mathrm{mt}})(1 - \kappa_w))}{v(1 - f_{\mathrm{mt}})(1 - \kappa_w)} \\
&= \frac{1}{1 - f_{\mathrm{mt}}}\frac{1 - \kappa_m}{1 - \kappa_w}[1 - \exp(-v(1 - f_{\mathrm{mt}})(1 - \kappa_w))]
\end{aligned}
$$

This is increasing in $v$.

## A4.6 Strongly immune hosts with small bottlenecks provide intuition for *Figure 4*

In a host strongly immune to the old variant (large $\kappa_w$), we have $-v(1-f_{\mathrm{mt}})(1-\kappa_w) \ll 1$. Applying the linear approximation to the exponential to equation A4.5:

$$
\begin{aligned}
p_{\mathrm{surv}}/p_{\mathrm{drift}} &\approx \frac{1}{1-f_{\mathrm{mt}}}\frac{1-\kappa_m}{1-\kappa_w}(v(1-f_{\mathrm{mt}})(1-\kappa_w)) \\
&= (1-\kappa_m)v
\end{aligned}
$$

This provides intuition for the effects seen in main text *Figure 4*, in which inoculation selection improves survival relative to drift by a factor of $\frac{v}{b}$ for a full escape mutant, and by a factor scaled down by $1-\kappa_m$ for a partial escape mutant.

Moreover, if we let $f_{\mathrm{mt}} = f_d$ for the drift case and $f_{\mathrm{mt}} = f_r$ for the selective case, the approximate ratio becomes

$$
p_{\mathrm{surv}}/p_{\mathrm{drift}} \approx \frac{f_r}{f_d}(1-\kappa_m)v
$$

Our replicator equation implies that this ratio $\frac{f_r}{f_d}$ will increase in expectation when more replication selection occurs prior to transmission (larger $\delta\tau$), explaining the increasing ratio of the selective variant survival relative to drift observed in *Figure 4* when replication selection is permitted prior to transmission.

Note that we did not integrate over the emergence times to obtain this ratio; instead, we derived principles that take the initial frequency as a given.

## A4.7 Effect of sIgA cross immunity $\sigma$ on probability of inoculation selection

We also wish to know which hosts best promote antigenic novelty when inoculated, given a level of cross-immunity $\sigma$, or a degree of escape $1-\sigma$. We will show that whereas replication selection suggests that intermediately immune hosts should be key selectors for antigenic novelty (*Grenfell et al., 2004*), in an inoculation selection regime, new variants may most often reach observable frequencies in fully immune hosts. We do this by analyzing an expression for $p_{\mathrm{surv}}$, the probability that a mutant survives transmission to found an infection in a new host, and showing how it changes with old variant neutralization probability $\kappa_w$ and sIgA cross immunity $\sigma = \kappa_m/\kappa_w$. We find that transmission survival probability can be maximized in strongly immune or fully naive hosts, depending on parameters.

We optimize $p_{\mathrm{surv}}(\kappa_w)$ given some value of $\sigma$ by differentiating.

$$
\begin{aligned}
\frac{\mathrm{d}p_{\mathrm{surv}}}{\mathrm{d}\kappa_w} &= \frac{\mathrm{d}p_{\mathrm{cib}}}{\mathrm{d}\kappa_w}p_{\mathrm{inoc}} + p_{\mathrm{cib}}\frac{\mathrm{d}p_{\mathrm{inoc}}}{\mathrm{d}\kappa_w} \\
&= \frac{\mathrm{d}p_{\mathrm{cib}}}{\mathrm{d}\kappa_w}p_{\mathrm{inoc}} + p_{\mathrm{cib}}(vf_{\mathrm{mt}}\sigma)(p_{\mathrm{inoc}}-1)
\end{aligned}
\tag{A23}
$$

Consider the endpoint at $\kappa_w = 1$. $p_{\mathrm{cib}} = 1$, $p_{\mathrm{inoc}} \approx f_{\mathrm{mt}}v(1-\kappa_m) = f_{\mathrm{mt}}v(1-\sigma)$, and $\bar{w} = v(1-f_{\mathrm{mt}})(1-\kappa_w) = 0$.

So $\frac{\mathrm{d}p_{\mathrm{cib}}}{\mathrm{d}\kappa_w} = \frac{\mathrm{d}p_{\mathrm{cib}}}{\mathrm{d}\bar{w}}\frac{\mathrm{d}\bar{w}}{\mathrm{d}\kappa_w} = \frac{\mathrm{d}p_{\mathrm{cib}}}{\mathrm{d}\bar{w}}(-v)(1-f_{\mathrm{mt}})$ can be evaluated (using the expressions for $\frac{\mathrm{d}p_{\mathrm{cib}}}{\mathrm{d}\bar{w}}$ derived in section A9.7 below), and it evaluates to 0 except if $b = 1$ (when it is equal to $\frac{1}{2}v(1-f_{\mathrm{mt}})$).

**Case 1**: $b>1$. For $b>1$, it follows from the above that

$$
\frac{\mathrm{d}p_{\mathrm{surv}}}{\mathrm{d}\kappa_w} = 0 - p_{\mathrm{cib}}\frac{\mathrm{d}p_{\mathrm{inoc}}}{\mathrm{d}\kappa_w}
\tag{A24}
$$

Since $\frac{\mathrm{d}p_{\mathrm{inoc}}}{\mathrm{d}\kappa_w} \geq 0$, $\frac{\mathrm{d}p_{\mathrm{surv}}}{\mathrm{d}\kappa_w} < 0$ unless $\frac{\mathrm{d}p_{\mathrm{inoc}}}{\mathrm{d}\kappa_w} = 0$, which occurs in cases of interest when $\sigma = 0$ (complete escape mutant).

In other words, if immune escape is incomplete for $b>1$, there is always some less strongly immune host (though possibly very slightly less, see *Appendix 1—figure 1*) that improves new variant survival relative to a host who neutralizes old variant virions with 100% certainty. Note, however, that hosts with $\kappa_w = 1$ are very unlikely to exist in nature. Our study is motivated precisely by the fact

that even an experienced host encountering homotypic reinfection neutralizes virions with probability $\kappa_w < 1$, and therefore can be productively reinfected (*Clements et al., 1986*; *McCrone et al., 2018*). It follows that in practice the most strongly immune hosts (with $\kappa_w$ large but less than 1) could still be the best selectors.

**Case 2**: $b = 1$. When the bottleneck is 1 and $\bar{w} = 0$, $\frac{\mathrm{d}p_{\mathrm{cib}}}{\mathrm{d}\kappa_w} = \frac{1}{2}v(1 - f_{\mathrm{mt}})$. So in this case, the derivative at $\kappa_w = 1$ may be positive or negative, depending on whether:

$$\frac{\mathrm{d}p_{\mathrm{cib}}}{\mathrm{d}\kappa_w}p_{\mathrm{inoc}} + p_{\mathrm{cib}}(vf_{\mathrm{mt}}\sigma)(p_{\mathrm{inoc}} - 1) > 0 \qquad (A25)$$

substituting $\frac{\mathrm{d}p_{\mathrm{cib}}}{\mathrm{d}\kappa_w} = \frac{1}{2}v(1 - f_{\mathrm{mt}})$ and $p_{\mathrm{cib}} = 1$, we have:

$$\frac{1}{2}(1 - f_{\mathrm{mt}})p_{\mathrm{inoc}} + f_{\mathrm{mt}}\sigma p_{\mathrm{inoc}} > f_{\mathrm{mt}}\sigma$$

$$\frac{1}{2f_{\mathrm{mt}}}(1 - f_{\mathrm{mt}})p_{\mathrm{inoc}} + \sigma p_{\mathrm{inoc}} > \sigma$$

$$\frac{1}{2f_{\mathrm{mt}}}(1 - f_{\mathrm{mt}})p_{\mathrm{inoc}} > \sigma(1 - p_{\mathrm{inoc}})$$

$$\frac{1 - f_{\mathrm{mt}}}{2f_{\mathrm{mt}}}\frac{p_{\mathrm{inoc}}}{1 - p_{\mathrm{inoc}}} > \sigma$$

When $p_{\mathrm{inoc}}$ is small, we have approximately $p_{\mathrm{inoc}} \approx f_{\mathrm{mt}}v(1 - \sigma)$ and $1 - p_{\mathrm{inoc}} \approx 1$, so:

$$\frac{1 - f_{\mathrm{mt}}}{2}v(1 - \sigma) > c$$

$$v\frac{1 - f_{\mathrm{mt}}}{2} > \frac{\sigma}{1 - \sigma} \qquad (A26)$$

In other words, with extremely small cell infection bottlenecks like those observed for influenza viruses, fully immune hosts are the best selectors provided that the number of virions $v$ encountering IgA antibodies is sufficiently large given the degree of escape achieved. Less immune escape (larger sIgA cross immunity $\sigma$) necessitates a larger $v$ to make fully immune hosts the best selectors. Since $f_{\mathrm{mt}} \ll 1$, a rule of thumb is that $v > \frac{2\sigma}{1 - \sigma}$.

The intuition is that larger $v$ means more competition for the cell infection bottleneck among virions that reach IgA, and thus a greater opportunity for the mutant's selective advantage to be realized, but that this only works provided that this advantage is large enough so that the mutant is not itself at too large a risk of being neutralized.

## A5 Parameter uncertainties and sensitivity analysis

Here we discuss parameter uncertainties in our models and how they affect our conclusions. We also conduct a simulation-based sensitivity analysis of the central within-host model.

### A5.1 Strength of selection

A crucial parameter in our model is $\delta$: the magnitude of the fitness advantage of the new variant over the old variant during viral replication in an infected individual. One possible objection to our analysis here is that the antibody-mediated virion neutralization rate $k$ is low enough or the antigenic similarity between the variants is great enough to make the fitness difference $\delta = k(c_w - c_m)$ small. In that case, replication selection to consensus will be rare even if $t_M$ is very small, and the adaptive response is mounted immediately (main text *Figure 1C,D*). Despite uncertainty about both $k$ and $c_w - c_m$, $k$ is likely to be high, perhaps extremely so. And given sufficiently high $k$ and a homotypically reinfected host ($c_w = 1$), even moderate immune escape ($0 \ll c_m < 1$) produces a substantial fitness difference. Moreover, assuming small $k$ early in infection grants our basic hypothesis: antibodies

mediated selection is weak early in infection, neutralization during viral replication is not the mechanism of protection against reinfection, and there is asynchrony between antigenic diversity and meaningful antigenic selection.

### A5.1.1 Antibody-mediated virion neutralization rate ($k$) may be very high

Parameter estimates for antibody-mediated virion neutralization in the presence of substantial well-matched antibody correspond to values of $k$ that are extremely high. For instance, by fitting a single-variant within-host model to data, *Cao and McCaw, 2017* estimate antibody neutralization rates between 0.4 and 0.8 per virion-day per pg/mL of antibody, and antibody concentrations of over 100 pg/mL by day 6 in an infection of a naive host. This corresponds to a $k$ of 40 to 80 in our more phenomenological model.

For $t_M = 0$ (constant immunity) if $k$ is sufficiently large that the old variant virus has an initial $\mathcal{R}(0) < 1$ (which occurs if $k > d_v(\mathcal{R}_0 - 1)$), then all visible reinfections will be mutant infections. At $\mathcal{R}_0 < 10$ with $d_v = 4$, this corresponds to a $k$ of 40, the lower end of the Cao and McCaw estimates.

A $k$ of this magnitude has several implications that support our hypotheses of asynchrony between diversity and selection. Such a $k$ would suffice to drive $\mathcal{R}(0)$ below 1. A sufficiently early activation of a recall antibody response would then imply that all observable infections of experienced hosts would be mutant infections (*Figure 2*). In intermediately immune hosts, there could be a substantial fitness advantage $\delta = (c_w - c_m)k$ for new variant viruses over old variant viruses (where $c_m, c_w$ are the cross immunities to the host memory variant for the old variant and the new variant, respectively).

With large $k$ and $t_M = 0$, intermediately immune hosts who cannot block transmission could easily have $\mathcal{R}(0) \approx 1$ for the old variant; this would make them excellent at generating and selecting for mutant before an infection is cleared.

A final reason why a high $k$ supports the hypotheses of this paper is that antigenic evolution is extremely rapid if a sufficiently strong recall antibody responses becomes active after viral replication begins but before the infection peaks. New antigenic variants should be generated with near certainty by virus exponential growth well before the infection's peak—roughly when the number replications that have occurred is at or above the order of magnitude of the inverse mutation rate (*Figure 8*). Suppose a strong ($\mathcal{R}(t_e) < 1$) recall response is mounted at that point $t_e$. The mutant then has target cell resources on which to grow (since $C(t_e) \gg C_{\text{peak}}$) and experiences negligible replication competition from the old variant (since the old variant population is not growing but in fact is shrinking). If not lost stochastically, it should therefore emerge to detectable and transmissible levels. We do not observe this. This suggests that if $k$ is large, not only must the antibody response not be immediate ($t_M > 0$), but it must also be late enough enough to make this effect unlikely: it must happen either just before or after the peak of infection. Evidence suggests that this is indeed the case. Infections peak by 36–48 hours post-infection, and antibody responses only begin 48–72 hours post infection.

### A5.1.2 Small values of $k$ are a sub-hypothesis of the general model of asynchrony between diversity and selection

Small $k$ early in infection does produce weak replication selection, but it means that neutralization during viral replication cannot explain protection against reinfection. In such a scenario, it is still necessary to invoke mucosal sIgA antibodies or another mechanism of protection at the point of transmission, and so inoculation selection again comes into play.

Indeed, it is unlikely that $k$ is truly zero before the adaptive response is mounted. Occasionally, a virion may encounter residual IgA antibodies, for instance (see Section A2). But the effective value of $k$ is likely to be small—too small to curtail the growing infection or produce substantial replication selection. We set it to 0 before $t_M$ for simplicity of model analysis, but our simple binary model approximates the likely scenario in which $k$ is small but non-zero before $t_M$ and then increasingly large afterwards (see A5.1.1).

## A5.2 Relationship between $k$ and $\kappa$

As noted in the Discussion, the relationship between the mucosal antibody neutralization rate $\kappa$ and the antibody neutralization rate during replication $k$ is unknown, though we have every reason to expect it to be positive.

$\kappa$ values are particularly difficult to calculate. In our model, we have generally assumed that all virions of type $i$ inoculated into an experienced host are independently neutralized with probability $\kappa_i$, and therefore $z_i = e^{-v(1-\kappa_i)}$ for an inoculum consisting only of type $i$. But this assumption of independence may be violated in practice.

One reason statistical independence may be violated is that antibody numbers are finite, and an antibody that binds to one virion cannot bind to another. Consider a focal virion. Given that another virion has been neutralized, there are fewer antibodies remaining to neutralize our focal virion. This is particularly important for mixed inocula. Old variant virions, which have higher affinity for the inoculated host's sIgA antibodies, may indirectly protect the new variant by competing with it for antibody-binding. A new variant virion may have a higher individual chance of being neutralized by those same antibodies if it is part of a monomorphic inoculum composed of other, identical new variants. Such an interaction would strengthen inoculation selection relative to the independent neutralization we have modeled here.

## A5.3 Bottleneck sizes

Inoculation selection may improve mutant bottleneck survival relative to neutral drift (see *Figure 4F*, *Appendix 1—figure 1*, and Section A4). For this to be true, a non-antigenic bottlenecking must follow IgA antibody neutralization, with a ratio $v/b \gg 1$

Evidence suggests that a non-antigenic bottleneck does occur after the sIgA bottleneck because bottlenecks measured in vaccinated and non-vaccinated hosts are of comparable size (indistinguishable from one), suggesting that founding virion population sizes are cut down to small numbers even in the absence of IgA antibodies, though this would also be consistent with the case of $v = 1$ (i.e. IgA antibodies must neutralize on average a single virion to prevent infection, and this virion, if not neutralized or otherwise lost, uniquely founds the infection).

An experimental evolution study of avian influenza virus adaptation in ferrets found that transmission bottlenecks in naive hosts became tighter that as the virus adapted (*Moncla et al., 2016*). A reanalysis of that data found that bottlenecks were tight throughout (*Lumby et al., 2018*). The fact that bottlenecks do not appear to loosen (and may tighten) with adaptation for better transmission and replication is further evidence, albeit circumstantial, that when an influenza virus is well-adapted to its host and replicates rapidly, the first virion or first few virions to infect a cell will be the ancestor of the vast majority of progeny viruses.

## A5.4 Double-peaked infections

One modeling study of influenza viruses proposes that infections should have a second peak after initial innate responses are overcome and some target cells are once again susceptible to infection (*Pawelek et al., 2012*). A relaxation of non-antigenic limiting factors later in infection can and should provide additional opportunities for replication selection, as occurs in immune-comprised human patients.

That said, the empirical data suggesting double-peaked kinetics comes from experimental inoculation of naive horses with a large quantity of influenza virus: $10^6$ 50% egg infectious dose (*Quinlivan et al., 2007*). The novel antibody response curbed the second peak of infection, which was much lower than the first.

In human kinetics data (*Hadjichrysanthou et al., 2016*) and animal transmission experiments in which one animal infects another (*Le Sage et al., 2020*; *Canini et al., 2020*) it is common to see clearly single-peaked infections.

Furthermore, for realistic (incomplete) degrees of antibody-binding escape, we expect both old and new antigenic variant population sizes to decline even due to *antigenic* limiting factors, though of course we expect the new variant to decline more slowly.

The upshot is that virus clearance should be rapid following the mounting of the adaptive response in the experienced hosts where we expect selection to take place, and thus in immune-competent hosts we should expect a limited window for antibody-mediated selection on transmissible virus populations.

## A5.5 Threshold versus probabilistic transmission

In the simulation results shown in the main text (*Figures 1,2,4,6*) we use a probabilistic model of transmission: the donor host's probability of inoculating the recipient at a given point during the infection is proportional to donor viral load in virions, $V_{\text{tot}}(t)$.

Since influenza virus population sizes grow and shrink rapidly around the within-host peak, however, this model should be qualitatively similar to a threshold model in which transmission occurs with certainty if the total viral load is above a certain threshold $\theta$ and does not occur if it is below that threshold. To confirm this, we here show corresponding transmission chain simulation results based on a threshold model, with the threshold as in *Table 1*.

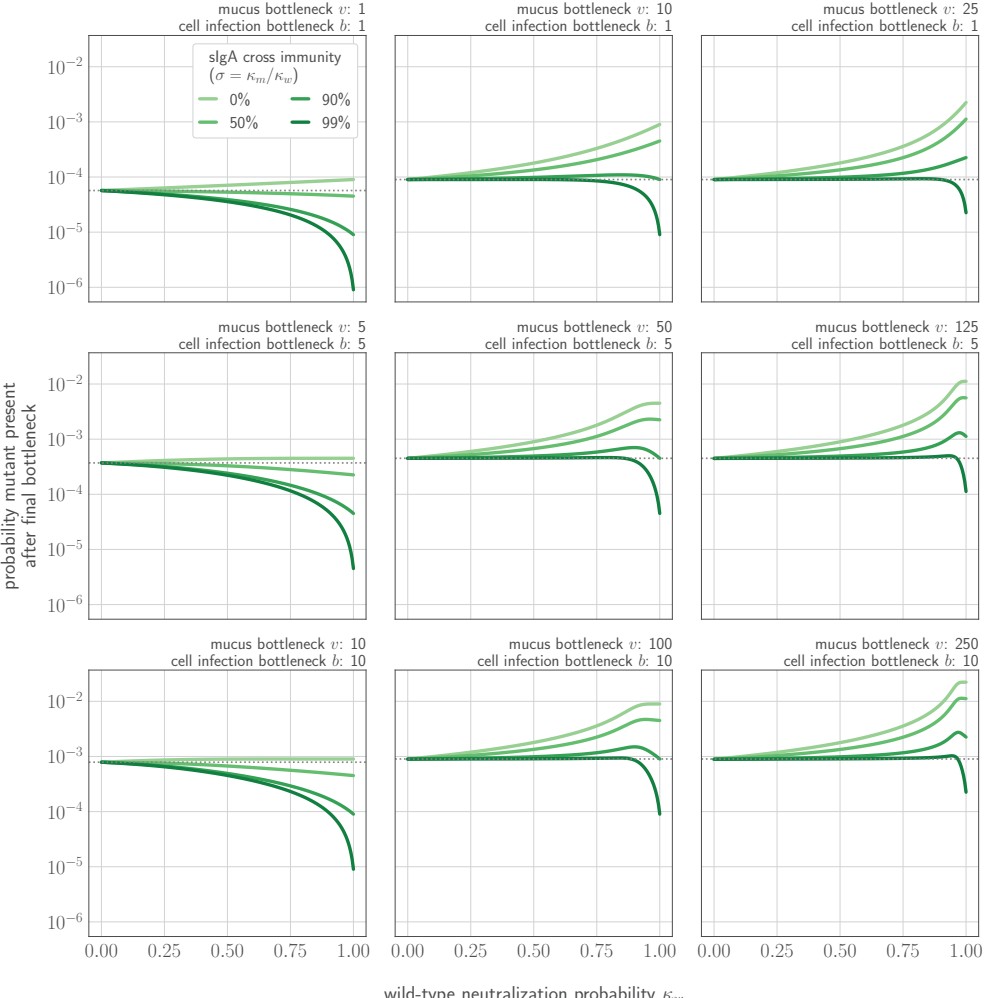

**Appendix 1—figure 1.** Probability that a new variant is present after the cell infection (final) bottleneck as a function of cross immunity and degree of competition for the final bottleneck. Probability shown as a function of probability of no old variant infection ($z_w$), degree of cross immunity between mutant and new variant $\sigma = \kappa_m/\kappa_w$, mucus bottleneck size $v$, and final bottleneck size $b$. Gray dotted line indicates probability that a new variant survives the transmission bottleneck in a host who is naive both to the old variant and to the new variant (i.e. drift). $f_{mt} = 9 \times 10^{-5}$, a typical value for a naive transmitting host in stochastic simulations.

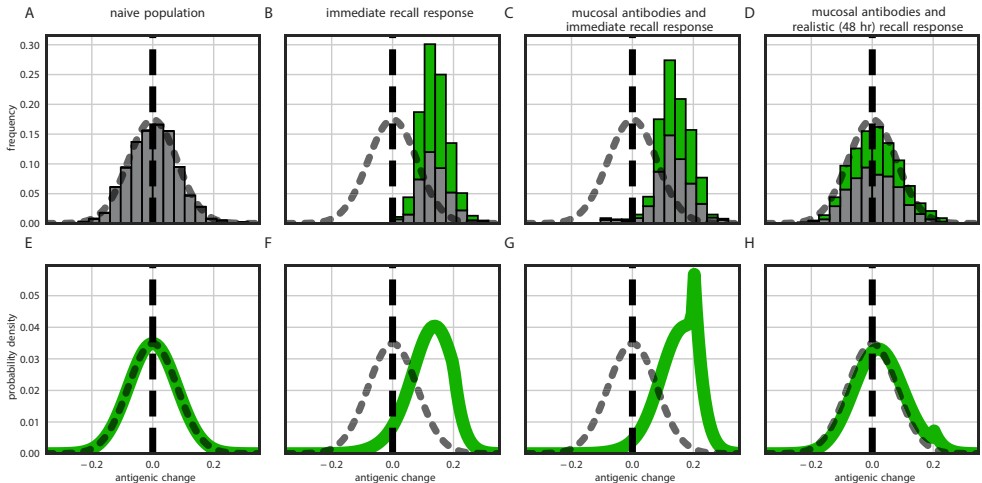

**Appendix 1—figure 2.** Distribution of mutant effects given replication and inoculation selection, with a transmission threshold model. Threshold version of *Figure 4*. Distribution of antigenic changes along 1000 simulated transmission chains (**A–D**) and from an analytical model (**E–H**). In (**A,E**) all naive hosts, in other panels a mix of naive hosts and experienced hosts. Antigenic phenotypes are numbers in a 1-dimensional antigenic space and govern both sIgA cross immunity, $\sigma$, and replication cross immunity, $c$. A distance of $\geq 1$ corresponds to no cross immunity between phenotypes and a distance of 0 to complete cross immunity. Gray line gives the shape of Gaussian within-host mutation kernel. Histograms show frequency distribution of observed antigenic change events and indicate whether the change took place in a naive (gray) or experienced (green) host. In (**B–D**) distribution of host immune histories is 20% of individuals previously exposed to phenotype $-0.8$, 20% to phenotype $-0.5$, 20% to phenotype 0 and the remaining 40% of hosts naive. In (**E**), naive hosts inoculate naive hosts. In (**F–H**) hosts with history $-0.8$ inoculate hosts with history $-0.8$. Initial variant has phenotype 0 in all sub-panels. Model parameters as in *Table 1*, except $k = 25$. Spikes in densities occur at 0.2 as this is the point of full escape in a host previously exposed to phenotype $-0.8$.

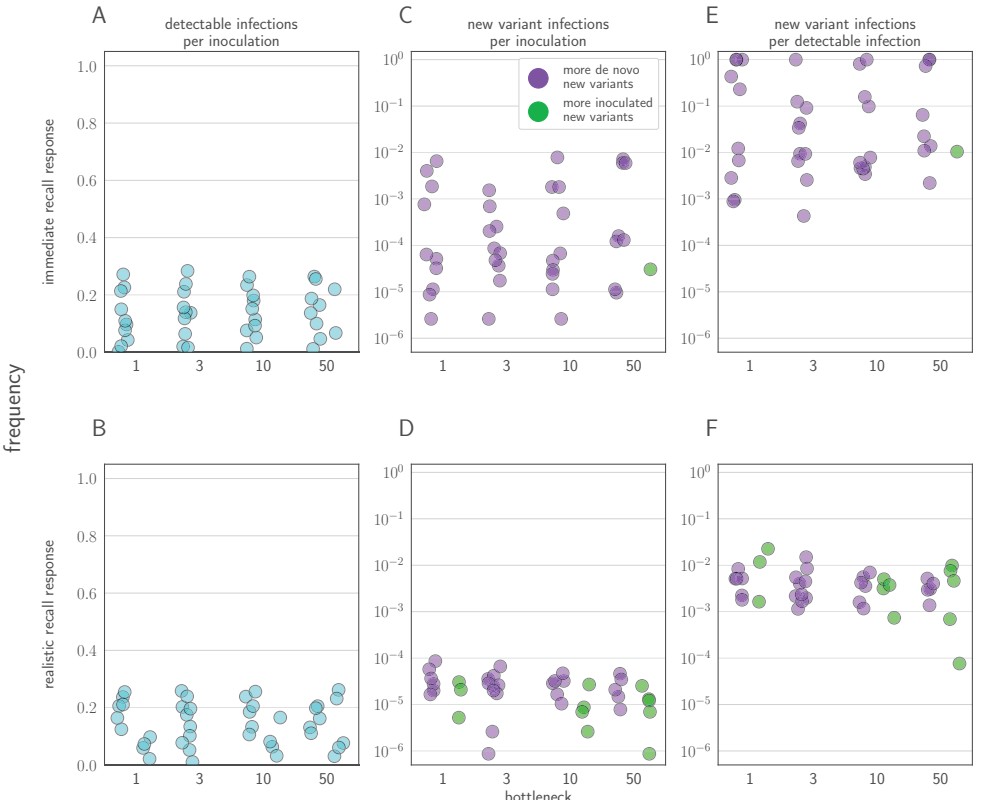

**Appendix 1—figure 3.** Sensitivity analysis varying model parameters across biologically-reasonable parameter ranges. (**A, B**) Probability of a detectable infection per inoculation of an experienced host. (**C, D**) Probability of a detectable new variant infections per inoculation of an experienced host. (**E, F**) Fraction of detectable infections of experienced hosts that are new variant infections. Points colored according to whether new variant infections were more frequently caused by de novo generated (purple) or inoculated (green) new variant viruses. Each point represents a random parameter set; 10 random parameter sets generated for each bottleneck value shown, and 50,000 inoculations of an experienced host simulated for each parameter set. Two regimes of simulated parameter set shown: (**A, C, E**) an immediate recall response regime, in which $t_M$ varied from 0 to 1, and (**B, D, F**) a realistically-timed response regime, in which $t_M$ varied from 2 to 4.5. All other parameters varied across the same ranges in both regimes (see *Table 2* for ranges). Parameters Latin Hypercube sampled from within ranges for each regime and bottleneck size.

## A5.6 Sensitivity analysis

As described in the Materials and methods, we further assessed sensitivity of the within-host model to variation in key parameters by studying random parameter sets chosen from biologically plausible parameter ranges.

We analyzed two cases: one in which the immune response is unrealistically early ($0 \leq t_M \leq 1$, *Appendix 1—figure 4*), and one in which it is realistically-timed ($2 \leq t_M \leq 4.5$) *Appendix 1—figure 5*, see also *Appendix 1—figure 3* and other model parameters in *Table 2*.

We found that with early immunity, regardless of particular parameter values, new variants are frequently seen when experienced hosts are detectably reinfected, and the overall probability of new variant infections is unrealistically high. These observable new variants are most often generated de novo and replication-selected (*Appendix 1—figure 4*, see also *Appendix 1—figure 3*).

When immunity is realistically-timed, the pattern of much rarer replication selection is robust to variation in parameters. New variant infections compose a realistically small fraction of all detectable reinfections (*Appendix 1—figure 5*, *Appendix 1—figure 3*). Moreover, inoculation selection is

sometimes more common than replication selection as a source of new antigenic variants (*Appendix 1—figure 5*, see also *Appendix 1—figure 3*).

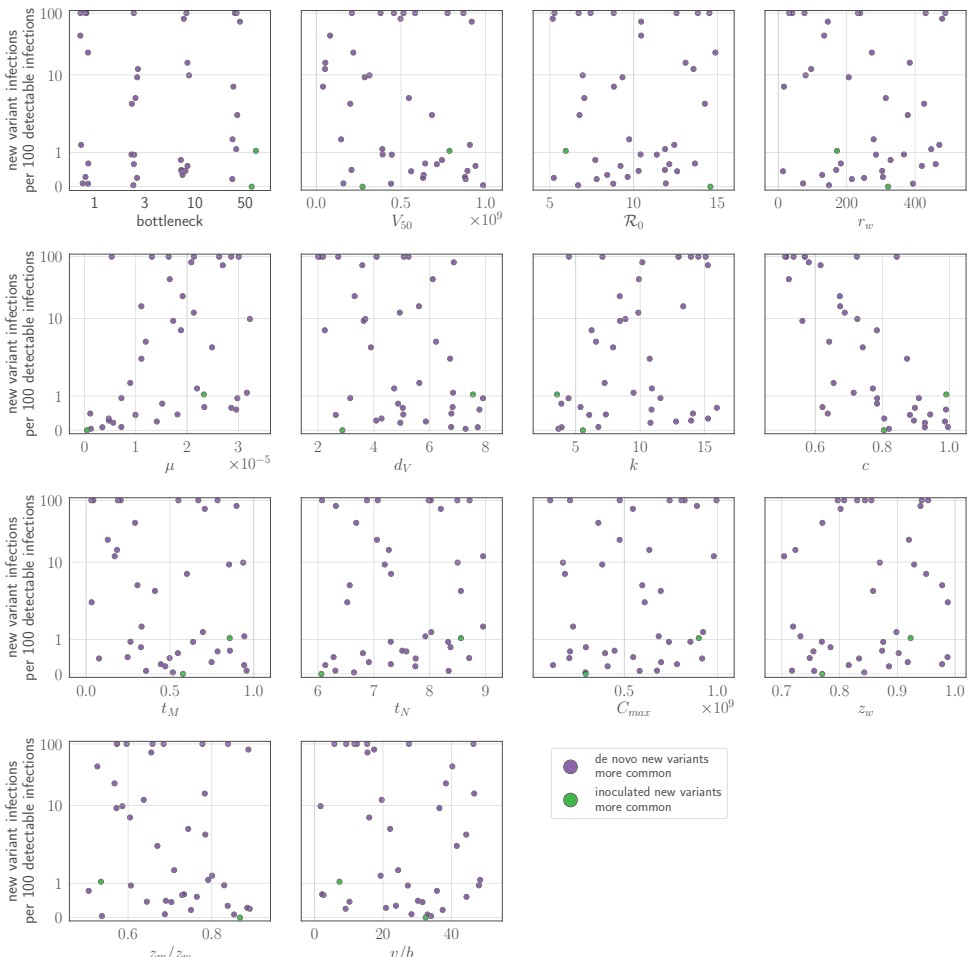

**Appendix 1—figure 4.** Sensitivity analysis: parameter values versus rate of new variant infections per 100 detectable infections of an experienced host, given an unrealistically early recall response. Parameters randomly varied across the ranges given in *Table 2*, with $t_M$ varied between 0 and 1. Each point represents a parameter set; the rate of new variant infections per hundred 100 detectable infections is estimated from 50,000 simulated inoculations of an experienced host. A new variant infection is defined as one in which the new variant reached a transmissible frequency of at least 1% at any point in the infection. Points are colored according to whether new variant infections were more frequently caused by de novo generated (purple) or inoculated (green) new variant viruses.

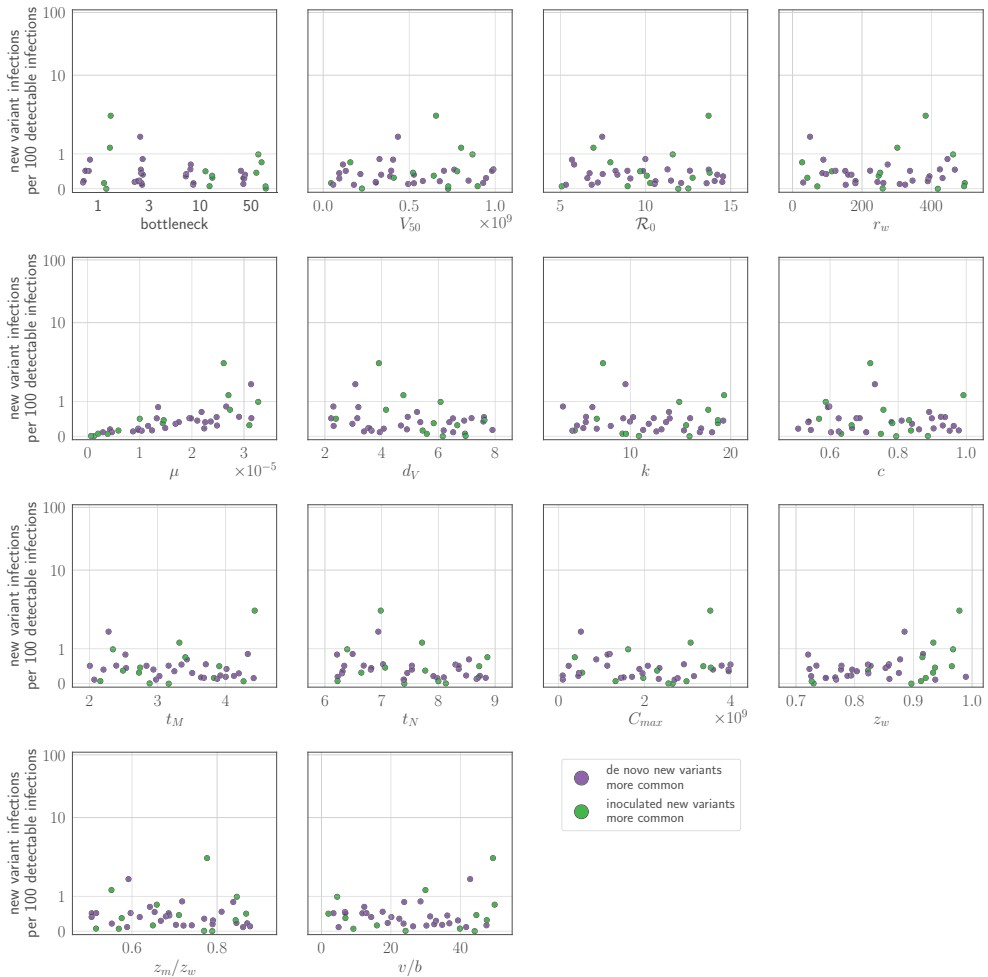

**Appendix 1—figure 5.** Sensitivity analysis: parameter values versus rate of new variant infections per 100 detectable infections of an experienced host given a realistic (48 hours or more post-infection) recall response. Parameters randomly varied across the ranges given in *Table 2*, with $t_M$ varied between 2 and 4.5. Each point represents a parameter set; the rate of new variant infections per hundred 100 detectable infections is estimated from 50,000 simulated inoculations of an experienced host. A new variant infection is defined as one in which the new variant reached a transmissible frequency of at least 1% at any point in the infection. Points are colored according to whether new variant infections were more frequently caused by de novo generated (purple) or inoculated (green) new variant viruses.

## A6 Polyphyletic antigenicity altering substitutions
### A6.1 Background

Amino acid substitutions associated with antigenic "cluster transitions" have reached fixation in parallel—and near-synchronously in time—in genetically distinct co-circulating lineages. This has occurred both in A/H1N1 and in A/H3N2 seasonal influenza viruses.

Existing 'mutation-limited' or 'diversity-limited' model explanations of why observable influenza virus antigenic novelty is rare despite continuous genetic evolution are less convincing in light of this synchronous polyphyly (see Section A7). In particular, a number of models hypothesize that large effect antigenicity-altering substitutions can only emerge in specific genetic contexts (*Koelle et al., 2006*; *Gog, 2008*; *Koelle and Rasmussen, 2015*; *Kucharski and Gog, 2012*), and that this constraint limits the rate of population-level antigenic change.

Synchronous, polyphyletic cluster transitions cast doubt on these claims. If major antigenic changes appeared infrequently due to the rarity of 'jackpot' scenarios of a large-effect mutant in a favorable genetic background (*Koelle and Rasmussen, 2015*), it would be surprising to see two or three distinct and synchronous jackpots after years of none. So while genetic background may play an important role in determining which lineages are fittest, synchronous polyphyly suggests that it is unlikely to set the clock of antigenic evolution. Rather, it suggests that accumulating population immunity may be crucial in setting the clock, and that there may even be a rising generation rate of observable antigenic novelty as a cluster ages, rather than a fixed rate (see main text Section "Population immunity sets the clock of antigenic evolution").

To show that large effect antigenic changes occur near-synchronously in multiple backgrounds, we assessed evolutionary relationships and built phylogenetic trees for known recent polyphyletic antigenicity-altering substitutions that have emerged in A/H1N1 and A/H3N2. Our analysis rules out reassortment as an explanation for the apparent polyphyly, thus verifying that the branches represent distinct, independent, but simultaneous de novo lineages.

## A6.2 Phylogenetic methods

We analyzed polyphyletic antigenic changes using hemagglutinin (HA) gene nucleotide sequences deposited in the GISAID EpiFlu database.

We downloaded all seasonal A/H1N1 HA sequences from the period 1999–2008, to center on the antigenic cluster transition from the New Caledonia/1999-like phenotype to the Solomon Islands/2006-like phenotype, excluding pandemic A/H1N1 viruses. We downloaded A/H3N2 virus HA sequences for two periods: 2008 to 2011, to center on cluster transition from Brisbane/07 to the Perth/2009-like and Victoria/2009-like phenotypic split, and 2012 to 2014, to capture the co-circulation of Clade 3C.2a and 3C.3a (*Appendix 1—table 1*). We discarded all sequences with an incomplete HA1 domain or with more than 1% ambiguous nucleotides.

**Appendix 1—table 1.** Dataset composition.

| A/H1N1 1999–2008 | | A/H3N2 2008–2011 | | A/H3N2 2012–2014 | |
|---|---|---|---|---|---|
| **Pre-filter** | **Final dataset** | **Pre-filter** | **Final dataset** | **Pre-filter** | **Final dataset** |
| 4882 | 3514 | 7050 | 5738 | 11970 | 10107 |

We aligned sequences using MAFFT v7.397 (*Katoh et al., 2002*) and reconstructed phylogenetic trees using RAxML 8.2.12 under the GTRGAMMA model (*Stamatakis, 2014*). We performed global optimization of branch length and topology on the RAxML reconstructed tree using Garli 2.01 with model parameters matching the RAxML reconstruction (500,000 generations) (*Bazinet et al., 2014*). It was computationally intractable to run Garli on the full phylogeny of A/H3N2 2012–2014 (n = 10107). We visualized phylogenetic trees using Figtree (http://tree.bio.ed.ac.uk/software/figtree) and ggtree (*Yu et al., 2017*)

We mapped amino acid substitutions onto branches using custom Python scripts. All numbering complies with H1/H3 numbering scheme (*Burke and Smith, 2014*).

## A6.3 A/H1N1

The HA K140E antigenicity-altering substitution first occurred in 2000 and was detected sporadically prior to its independent fixation in 2006–2007 in three phylogenetically distinct, geographically segregated lineages of A/H1N1 that diverged in 2004 (*Appendix 1—figure 6*; *Bedford et al., 2015*). The K140E substitution resulted in a cluster transition from the New Caledonia/1999-like antigenic phenotype to the Solomon Islands/2006-like phenotype, with lineage A emerging as the dominant lineage globally before the 2009 H1N1 pandemic (*Bedford et al., 2015*). Position 140 is located in the Ca2 antigenic site of the HA1 domain immediately adjacent to the receptor binding site, where amino acid composition mediates receptor binding function (*Koel et al., 2013*).

The three lineages have distinct HA1 mutational trajectories away from their most recent common ancestor, as defined by the set of amino acid substitutions that accumulate along the predominant

trunk lineages both prior to and following the fixation of the K140E substitution (*Appendix 1—table 2*, *Appendix 1—figure 6*). All three lineages acquired amino acid substitutions at previously characterized antigenic sites, as well as substitutions with suggested functional consequences, including glycosylation gain and loss (*Caton et al., 1982*). The three lineages share the Y94H substitution prior to K140E fixation, with limited overlap of other substitutions: lineage A and B share changes at residue 188, whereas A189T occurs in lineage B and A prior to and post K140E emergence respectively.

**Appendix 1—table 2.** Substitutions that characterize the co-circulating K140E-defined lineages.

| Lineage | Geographic composition in first year of co-circulation | Trunk substitutions from MRCA to K140E fixation | Trunk substitutions post-K140E fixation |
|---|---|---|---|
| 1 | South Asia | Y94H, R188K, E273K | D35N, K145R*, A189T, G185V, N183S, G185S |
| 2 | East Asia | Y94H, S36N, A189T, R188M, T193K* | N244S, K82R*, I47K, E68G |
| 3 | South-East Asia | Y94H, K73R, V128A*, A128T* | P270S |

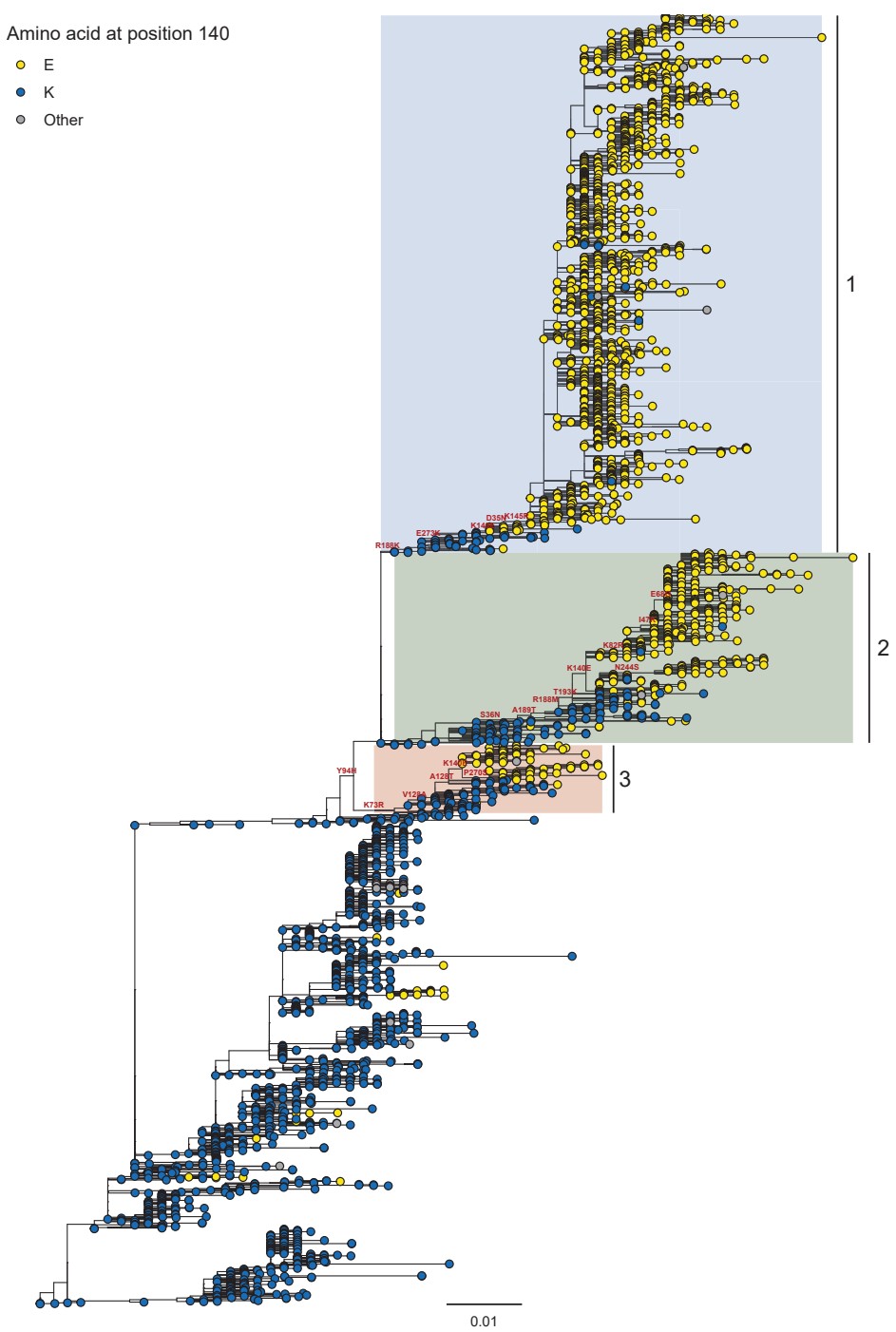

**Appendix 1—figure 6.** Phylogeny of A/H1N1 seasonal viruses for the period 1999 to 2008. Branch tip color indicates the amino acid identity at position 140. Co-circulating lineages defined by the K140E fixation are highlighted. Scale bar indicates the number of nucleotide substitutions per site. Tree rooted to A/New Caledonia/20/1999.

## A6.4 A/H3N2 2008-2011

In the period 2008-2011, the K158N substitution was only detected once in 2009 before fixing in combination with N189K in the same year in two distinct co-circulating A/H3N2 lineages. N189K was

not detected before fixing as a K158N/N189K double substitution. The combination of the substitutions resulted in an antigenic phenotype switch from Wisconsin/2005-like to the Perth/2009-like and Victoria/2009-like phenotypes respectively. Residues 158 and 189 are both located in antigenic site B adjacent to the receptor binding site, with substitutions at both residues characterized as cluster-transition substitutions in multiple historic A/H3N2 cluster transitions (*Koel et al., 2013*). The HA-defined evolutionary trajectories of viruses from the Perth/2009-like lineage and Victoria/2009-like lineages from their most recent common ancestor are characterized by distinct sets of substitutions (*Appendix 1—table 3*, *Appendix 1—figure 7*). Both lineages acquired substitutions at previously characterized antigenic sites (*Koel et al., 2013*), but only share the T212A substitution.

**Appendix 1—table 3.** Substitutions that characterize the co-circulating genetic / antigenic clades defined by K158N and N189K.

| Lineage | Trunk substitutions from MRCA to K158N/N189K fixation | Trunk substitutions post- K158N/N189K fixation |
|---|---|---|
| 1 (Victoria / 2009-like) | | T212A, S45N, T48A, K92R, Q57H, A198S, V223I, N312S, N278K, Q33R, N145S, G5E, E62V, D53N, E280A, I230V, Y94H, I192T, S199A |
| 2 (Perth / 2009-like) | E62K | N144K, R261Q, I260M, P162S, E50K, V213A, N133D, T212A, R142G |

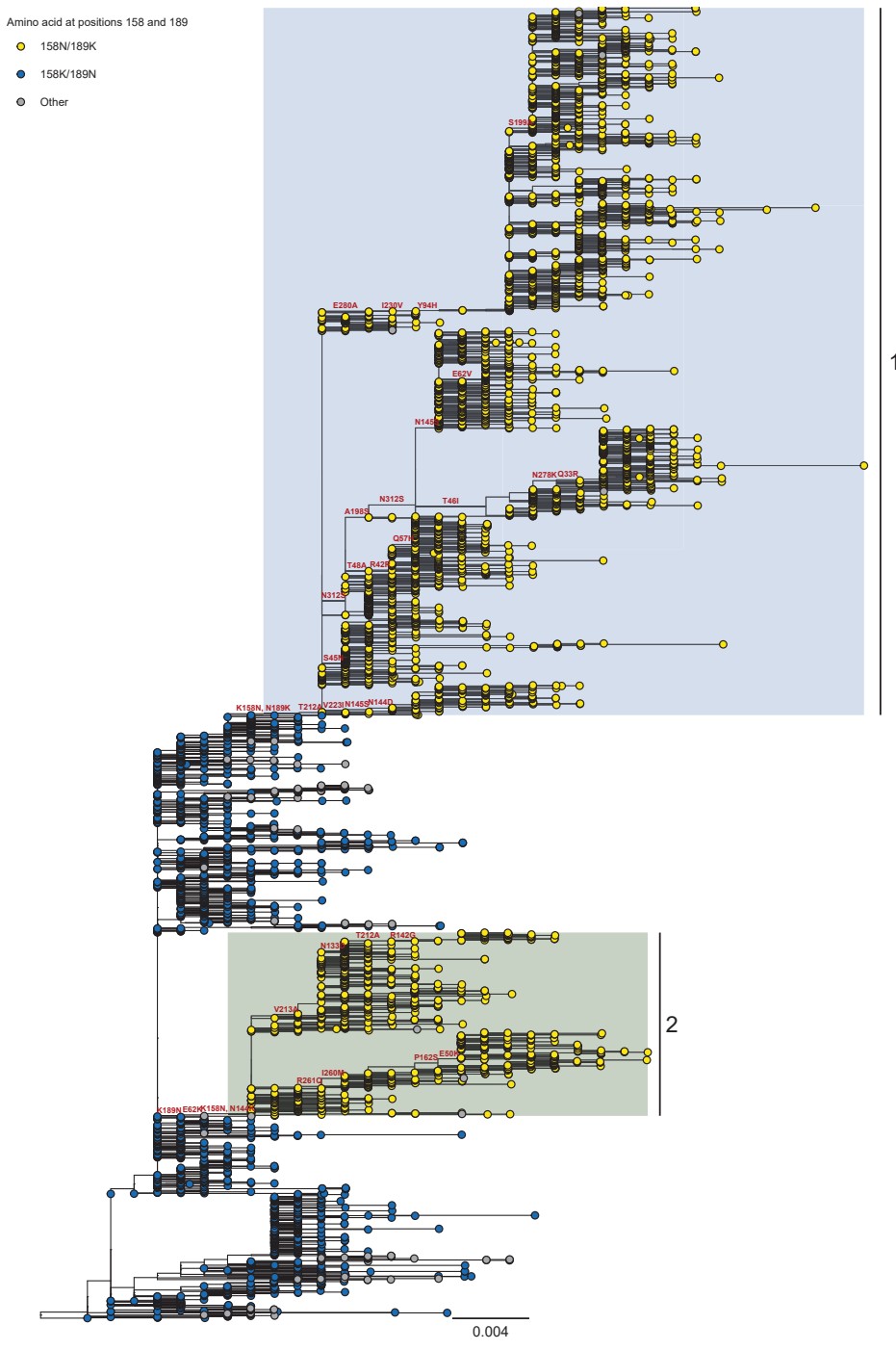

**Appendix 1—figure 7.** Phylogeny of A/H3N2 seasonal viruses for the period 2008 to 2011. Branch tip color indicates the amino acid identity at position 158 and 189. Co-circulating lineages defined by the K158N/N189K fixation are highlighted. Scale bar indicates the number of nucleotide substitutions per site. Tree rooted to A/Brisbane/10/2007.

## A6.5 A/H3N2 2012–2014

In 2014 two independent substitutions at position 159, F159S and F159Y, fixed in distinct A/H3N2 lineages co-circulating globally (*Appendix 1—figure 8*). The S159Y substitution was detected

sporadically in 2012/2013 before fixing in 2014 to define the A/H3N2 clade 3C.2a, which circulated as the dominant clade globally for the next three years. The F159S substitution was not detected in 2012–2013 before reaching fixation in 2014 to define clade 3C.3a, which continued to circulate globally at low frequencies.

Residue 159 is located in antigenic site B. The S159Y substitution in combination with Y155H and K189R substitutions previously resulted in the transition of the A/H3N2 Bangkok/79-like antigenic phenotype to Sichuan/87-like phenotype (*Koel et al., 2013*).

The two lineages have independent mutational trajectories for their HA gene away from their most recent common ancestor, excluding both acquiring the N225D substitution prior to F159X-fixation, with acquired changes at the major antigenic epitopes (*Appendix 1—table 4*, *Appendix 1—figure 8*; *Koel et al., 2013*).

**Appendix 1—table 4.** Substitutions that characterize the co-circulating genetic / antigenic clades defined by F159Y and F159S.

| Lineage | Trunk substitutions from MRCA to F159X fixation | Trunk substitutions post- F159X fixation |
|---|---|---|
| F159Y (lineage 1, Clade 3C.2a) | L3I, N225D, Q311H, N144S, K160T | R142K, R261L |
| F159S (lineage 2, Clade 3C.3a) | R142G, T128A, A138S, N225D | |

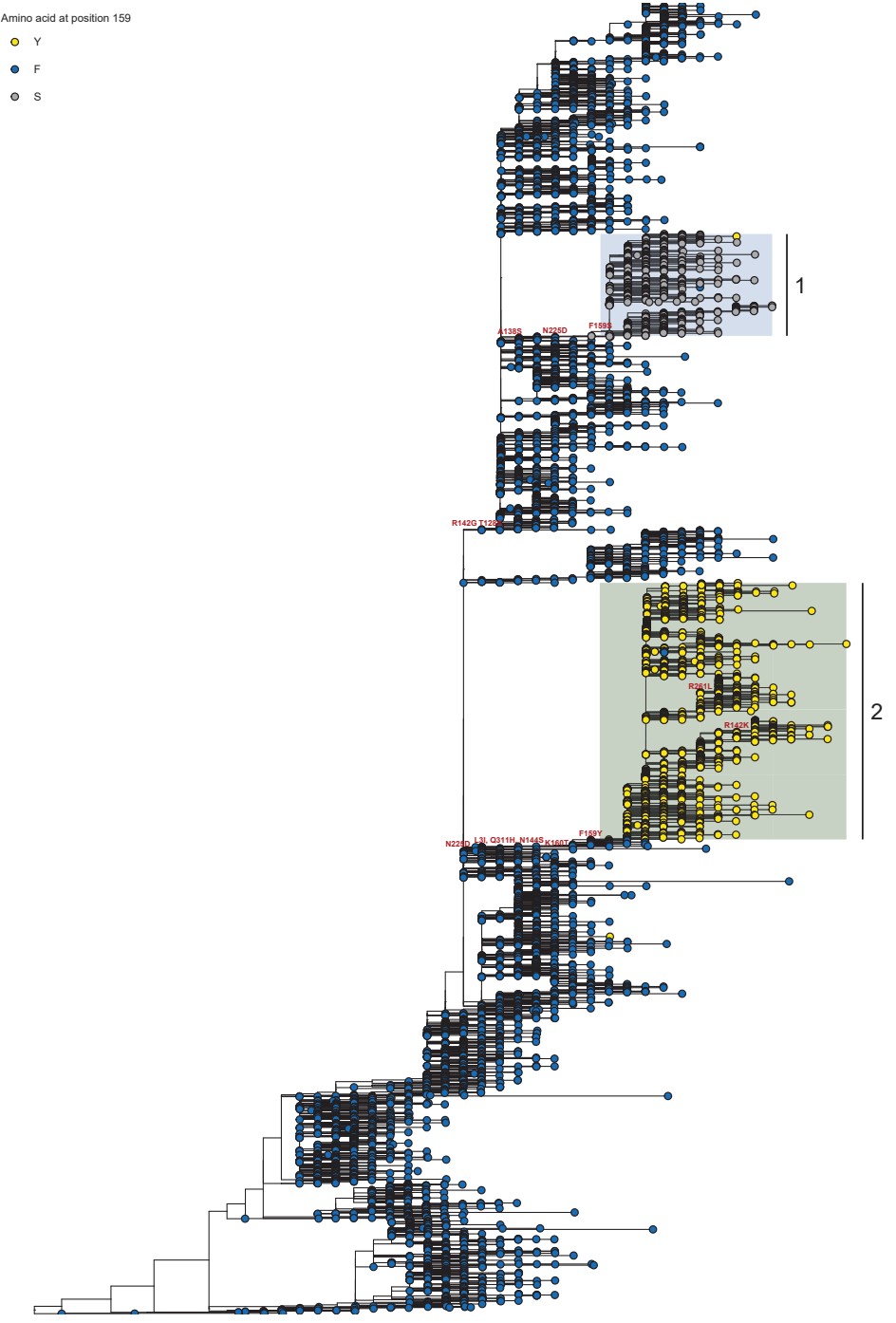

**Appendix 1—figure 8.** Phylogeny of A/H3N2 seasonal viruses for the period 2012 to 2014. Branch tip color indicates the amino acid identity at position 159. Co-circulating lineages defined by the F159X fixation are highlighted. Scale bar indicates the number of nucleotide substitutions per site. Tree is rooted to A/Perth/16/2009.

## A7 Prior theoretical studies

In this section, we discuss how our work fits into the substantial existing theoretical literature on influenza virus evolutionary dynamics.

A number of theoretical studies over the last 20 years have addressed the tempo and structure of seasonal influenza virus antigenic evolution, but only two have addressed within-host influenza virus antigenic evolution (*Luo et al., 2012*; *Volkov et al., 2010*). Most have focused on population level dynamics (*Ferguson et al., 2003*; *Koelle et al., 2006*; *Recker et al., 2007*; *Gog, 2008*; *Koelle et al., 2009*; *Strelkowa and Lässig, 2012*; *Bedford et al., 2012*; *Wikramaratna et al., 2013*; *Zinder et al., 2013*; *Koelle and Rasmussen, 2015*).

Our model suggests revisions to the theoretical understanding of within-host and point-of-transmission evolutionary dynamics. The revised paradigm has implications for the population-level models—most notably, it suggests that the rate of population-level antigenic diversification may not be constant over time.

## A7.1 Prior within-host studies

The two within-host studies both make two key assumptions: (1) immune selection pressure is strong from the moment of inoculation in experienced hosts and (2) influenza virus transmission bottlenecks are wide ($1 \times 10^5$ virions [*Luo et al., 2012*]; the three most frequent within-host variants deterministically transmitted onward in *Volkov et al., 2010*). As discussed in Section A2, antibody-mediated selection pressure is in fact likely to vary substantially over the course of infection. Recent empirical studies have found that transmission bottlenecks are in fact very tight (on the order of a single virion [*McCrone et al., 2018*; *Xue and Bloom, 2019*]).

Both within-host studies find that new antigenic variants are routinely generated de novo during an infection and undergo substantial positive selection during that same infection. NGS studies of natural human infections suggest that this is in fact uncommon (*Debbink et al., 2017*; *McCrone et al., 2018*; *Xue and Bloom, 2020*).

The most crucial assumption in prior work on influenza virus immune escape within-hosts (*Luo et al., 2012*; *Volkov et al., 2010*) or within-host pathogen immune escape generally (*Kennedy and Read, 2017*) is that immunity protects against observable reinfection by neutralizing pathogens during the timecourse of replication. This is immunologically unrealistic for a virus that replicates as rapidly as influenza virus; it 'outruns' the memory B-cell response.

What is more, our models show that it protection via neutralization produces binary outcomes: either no reinfection, or detectable reinfection with exclusively mutant viruses. This binary outcome was previously considered a feature, not a bug, because the frequency of homotypic reinfection was not yet known.

A model of immune and therapeutic escape for a generic pathogen (*Kennedy and Read, 2017*) studied ideas related to replication selection to argue that prophylactic anti-pathogen interventions (such as pre-exposure vaccination) could be less vulnerable to pathogen evolutionary escape than therapeutic interventions (such as post-symptomatic courses of antivirals). Therapeutic interventions occur after a number of pathogen replication events and therefore select upon on a larger, more diverse array of potential pathogen variants. Prophylactic interventions may limit the number of replications before clearance and thereby limit opportunities for diversification. Like the influenza-specific models discussed, this general model does not address the question of how evolution can be rare in symptomatic or otherwise observable infections, where substantial antigenic diversity can be generated.

Similarly, the antimicrobial resistance literature makes a distinction between 'acquired' and 'transmitted' (or 'primary') drug resistance (*Bonhoeffer et al., 1997*), but this should not be confused with replication and inoculation selection. Transmitted resistance refers to the acquisition of a resistant infection from a host infected with resistant microbes, without regard for whether those microbes were a minority or majority variant in the transmitting host. Inoculation selection, in contrast, refers to natural selection acting on inoculated diversity that *favors* transmitted drug resistance variants or transmitted new antigenic variants. One variant present in a mixed infection has a higher probability of surviving the transmission bottleneck or becoming the majority variant in the new host than its frequency in the transmitting host alone would imply, simply because the recipient host is more susceptible to that variant than to any competing inoculated variants.

## A7.2 Limits to immune escape through non-variant-specific limiting factors

Two studies (*Luo et al., 2012*; *Hartfield and Alizon, 2014*) have previously noted that the evolutionary emergence of escape mutants or otherwise fitter pathogens can be made more difficult due to ongoing competition from the old variant for a shared resource, whether susceptible cells within a host, as in *Luo et al., 2012* and in our model, or susceptible hosts within a population, as in *Hartfield and Alizon, 2014*.

In our paradigm, non-variant-specific limiting factors such as target cell depletion play three roles. (1) Denying new variants the opportunity to rise in frequency once an antibody response has been mounted and fitness differences have become substantial. (2) Explaining why immune-compromised hosts, in whom these non-antigenic limiting factors are weaker, can select for antigenic mutants at the scale of a single infection. (3) Clearing infections of naive hosts well before an adaptive immune response can select an escape variant. Point (3) has been posited previously by [60], but to our knowledge (1) and (2) are novel.

*Hartfield and Alizon, 2014* focus on the probability that a fit mutant variant the emerges at the host population scale goes stochastically extinct before causing a large epidemic, and show that competition for hosts with the old variant raises this extinction probability. While this may be relevant to influenza viruses at the population level (see Section A8.6 below), an analogous within-host argument is unlikely to be a sufficient explanation for the rarity of influenza virus antigenic mutants within human hosts. New variants are likely to be generated when an influenza virus infection is still well within its exponential growth phase, competition for susceptible cells is relatively unimportant, and probability of stochastic extinction for all incipient lineages is therefore low (due also to influenza's high $\mathcal{R}_0$). The absence of selection, rather than a high rate of stochastic loss of mutants during within-host replication, is the most important component of our explanation for the rarity of observable influenza virus antigenic evolution within individual human hosts.

Note that *Hartfield and Alizon, 2015* also have a model of within-host immune escape for a general pathogen, but that model is designed to estimate the contribution of an active adaptive immune response to increasing or reducing the probability that a small escape mutant population goes stochastically extinct.

For influenza viruses, the de novo mutation that creates a stable new variant lineage of interest typically precedes time of adaptive immune proliferation, so a distinct model is required.

## A7.3 Fitness costs for antigenic mutants

Previous population-level studies have argued that non-antigenic fitness costs associated with antigenic substitutions (*Gog, 2008*; *Kucharski and Gog, 2012*) or deleterious mutational load on the influenza virus genome (*Koelle and Rasmussen, 2015*) are necessary to explain the pace of influenza virus antigenic evolution in the human population. These models introduce population-level antigenic novelty at rates 5–10 times higher than predicted by inoculation selection, and at equal rates early and late during the circulation of a particular antigenic variant.

There are empirical reasons to doubt that influenza viruses are in fact antigenic diversity-limited due to fitness costs. We find multiple instances of polyphyletic cluster transitions: a cluster- transition amino acid substitution arises and proliferates on two or more branches of the virus phylogeny at this same time (see Section A6). This is inconsistent with the hypothesis that antigenic variants are either intrinsically unfit (deleterious substitutions) or incidentally unfit (poor background) prior to the moment that they proliferate at the population level (*Gog, 2008*; *Kucharski and Gog, 2012*; *Koelle and Rasmussen, 2015*), since the cluster-transition mutant can reach surveillance-detectable levels even before substantial population immunity has accumulated (suggesting that strong antigenic selection is not required to overcome intrinsic deleteriousness), and it can fully emerge when the time is ripe against multiple independent genetic backgrounds, suggesting background may not suffice to constrain diversification earlier in a cluster's circulation.

In the absence of meaningful replication selection, the population level rate of antigenic diversification is the average rate at which new variants survive all transmission bottlenecks to found or co-found infections. In an inoculation selection regime, that rate is unlikely to surpass 2 in $10^4$ inoculations (*Figure 4A*). We obtain that estimate assuming no within-host replication cost for antigenic mutants, weak cross immunity between mutant and old variant viruses, and many experienced hosts

who reliably neutralize old variant virions at the point of transmission. Actual rates of new variant survival are almost certainly lower. First, we have assumed that a new antigenic variant can be produced by a single amino acid substitution (accessible via a single nucleotide substitution), however a recent detailed study of the antigenic evolution of A/H3N2 viruses shows that some new variants require more than one substitution (*Koel et al., 2013*). Second, some (possibly many) previously infected hosts will not possess well-matched antibodies due to original antigenic sin (*Davenport and Hennessy, 1956*), antigenic seniority (*Lessler et al., 2012*), immune backboosting (*Fonville et al., 2014*), or other sources individual-specific variation in antibody production (*Lee et al., 2019*). These factors individually, and particularly in combination, which have not been modeled in this study mean that the above 2 in $10^4$ transmission estimate is likely to be unrealistically high. By comparison, existing population level models that invoke fitness costs introduce population-level antigenic diversity at much higher rates. One study (*Koelle and Rasmussen, 2015*) introduces new antigenic variants at a rate of 7.5 per $10^4$ transmissions; another (*Kucharski and Gog, 2012*) introduces new population-level mutations at a continuous rate of $6.8 \times 10^{-4}$ mutations per infected individual per day, which should produce new variant infections by 2 days post infection in at least 1 of every $10^3$ infected hosts:

$$1 - \exp(-6.8 \times 10^{-4} * 2) \approx 0.0013$$

Prior to the accumulation of population immunity, infections dominated by new variants should be rare, and new variants should be nearly neutral relative to the old variant at the population level. This may be sufficient to explain the absence of diversification early in the circulation of a cluster without invoking non-antigenic fitness costs. That said, additional constraints on the virus, particularly those that reduce the within-host fitness of new antigenic variants, should further slow virus antigenic diversification by reducing variant frequencies in donor hosts and thereby reducing the probability that the variants survive transmission bottlenecks. Furthermore, in the absence of substantial population immunity, adaptive substitutions may be lost at the population level due to competition from lineages with non-antigenic adaptive substitutions. Our work is thus consistent with a role for clonal interference in population level influenza virus antigenic evolution (*Strelkowa and Lässig, 2012*).

## A7.4 Further discussion of alternative explanations for rare within-host new antigenic variants

In this section, we expand on the discussion of alternative hypotheses from the main text (see Alternative explanations for rare new antigenic variants).

### A7.4.1 Protection through neutralization during early viral replication, that is, an immediate recall response

Adaptive immune pressure could be strong enough that $\mathcal{R}^w(0) < 1$. In that case, an old variant infection dies out if it does not produce a mutant virion, and it only has, on average, $\frac{b}{1 - \mathcal{R}^w(0)} - b$ replication events in order to do so before the infection is cleared (average replication events in $b$ independent subcritical branching processes with branching parameter $\mathcal{R}^w(0)$, see Section A3.6), so antigenic mutants can be rare in homotypically challenged hosts, depending upon $b$ and the mutation rate μ (*Luo et al., 2012*).

As we note in the main text, this makes two predictions that do not match empirical reality: binary outcomes to homotypic challenge—no infection or infection with an antigenic mutant—and frequent evolution of antigenic mutants in infections of intermediately immune hosts.

Recent empirical work (*Debbink et al., 2017*; *McCrone et al., 2018*; *Javaid et al., 2020*) and human challenge studies (*Clements et al., 1986*; *Memoli et al., 2020*) both suggest that detectable reinfection of experienced hosts frequently occurs without observable immune escape. This contradicts the binary outcome.

A model of influenza virus evolution *Volkov et al., 2010* found that even a small number of intermediately immune hosts along a transmission chain should reliably promote the evolution of new antigenic variants. The model predicted this because it features an immediate recall response, which implies efficient replication selection in intermediately immune hosts. In that model, intermediately

immune hosts can be productively infected with the old variant ($\mathcal{R}^w(0)>1$), so they reliably generate antigenic mutants. Those mutants then immediately undergo positive selection and are often transmitted onward. We see the same phenomenon in our own transmission chain model when there is an immediate recall response (see main text *Figure 4*).

Intermediate-to-strong immunity to old variant viruses is likely to be common in the human population even at the beginning of a new antigenic cluster's circulation (*Fonville et al., 2014*), and yet new variants are rarely observed until a new variant has circulated for multiple years (*Smith et al., 2004*). This suggests that intermediately immune hosts do not generate and select for new antigenic variants as readily as the model in *Volkov et al., 2010* implies.

Our proposed model of immune selection can explain why intermediately immune hosts may not reliably select for new antigenic variants. A realistically-timed recall response makes replication selection unlikely. Rates of inoculation selection are limited in all hosts due to low transmitting host mutant frequencies $f_{\mathrm{mt}}$. Intermediately immune hosts who possesses sIgA that are not well-matched to the old variant may additionally have a relatively low value of $\kappa_w$ and therefore be little better than a naive host in promoting new variant survival at the point of transmission.

## Heterogeneous neutralization rates during early viral replication

One possibility is that adaptive immune pressure could be present from the start of replication, but vary substantially among individuals, even those with the same immune history. This heterogeneity could explain why some individuals ($k$ small, $\mathcal{R}^w(0)>1$) are productively reinfected with old variant viruses while others ($k$ large, $\mathcal{R}^w(0)<1$) are protected. In this way one could have protection via neutralization during the timecourse of infection but still have frequent observable reinfection without immune escape.

While individual variation in immunity does exist (*Lee et al., 2019*), there are two reasons why this hypothesis is implausible. First, it is unrealistic given our current understanding of the adaptive immune response to influenza viruses, as discussed in Section A2, since it requires a substantial antibody response before 48 hours post-infection. Second, such heterogeneous protection would be need to be extremely bimodal to completely miss the regime of $k$ values in which replication selection is efficient.

Even if this unlikely hypothesis of bimodal early adaptive responses were true, there would remain a role for sampling effects at the point of transmission. If individuals either possess sterilizing-strength immunity that acts early in the timecourse of viral replication or possess sufficiently weak immunity that replication selection is unlikely, antigenic evolution would be dominated by cases in which mutants are inoculated into strongly immune hosts, as in simpler models with $\mathcal{R}^w(0)<1$. Inoculated mutants remain crucial in this scenario.

### A7.4.2 Deleterious antigenic mutants

Antigenic mutants could be replication-competent, but weakly deleterious within-host in the absence of immune selection and/or compensatory substitutions. Two studies (*Gog, 2008*; *Kucharski and Gog, 2012*) have invoked this hypothesis to explain population-level antigenic dynamics. If neutralization is sufficiently strong during virus replication, however, replication selection can still promote mutants during infections of experienced hosts, even in the absence of compensatory mutations. Within-host replication costs act to decrease the value of the fitness difference $\delta$. Recall that:

$$\delta(t) = [g_m(t) - g_w(t)] - [d_m(t) - d_w(t)] \tag{A27}$$

Typically $g_m(t) = g_w(t)$ and so $\delta = k(c_w - c_m)$. Here, $g_m < g_w$. For instance, we could have $r_m = q r_w$, $q<1$ and therefore $g_m = q g_w$. In that case:

$$\delta = k(c_w - c_m) - g(t)(1-q) \tag{A28}$$

We estimate $g$ to be order 20 early in infection and declining subsequently. $k$ may well be order 10. So it is very plausible that a weakly deleterious mutant ($q = 0.95$, for instance) could still undergo substantial replication selection early in infection if it offered enough immune escape ($c = 0.70$) and antibodies were present ($t_M$ small).

The effect could be particularly extreme in intermediately immune hosts if immunity drops superlinearly with antigenic distance. If the old variant is neutralized at a rate $kc_w = 4$ but the mutant is barely neutralized at all $c_m \approx 0$, it could easily overcome even substantial within-host deleteriousness.

In the absence of replication selection, even weak within-host deleteriousness should lead mutants to be rapidly purged (purifying selection should be efficient for phenotypes subject to replication selection) (*Sigal et al., 2018*), reducing $f_{\mathrm{mt}}$. Within-host deleteriousness would therefore reduce the rate at which new variants survive bottlenecks, since $p_{\mathrm{surv}}$ and $p_{\mathrm{drift}}$ both decrease as $f_{\mathrm{mt}}$ decreases (see Section A4.4).

A final possibility is that antigenic mutants could be extremely deleterious in the absence of compensatory mutations: $q$ small enough that $\delta < 0$ even when $t > t_M$ (i.e. the old variant is more fit within-host than the new variant even in the presence of a well-matched adaptive immune response). This scenario is unlikely given the strength of recall responses, but it would make inoculation selection and founder effects even more important for population-level antigenic diversification.

## A8 Further implications

### A8.1 The importance of inoculated diversity

*Kennedy and Read, 2017* and *Luo et al., 2012* study immune escape with $\mathcal{R}^w(0) < 1$ but consider only diversity generated after inoculation. Our study complicates their results: we find that when $\mathcal{R}^w(0) < 1$ the dominant mode of antigenic evolution will be selection on inoculated diversity, not selection on generated diversity (*Figure 2*). The pace of evolution will depend on how likely an escape mutant is to be transmitted to an experienced host. If bottlenecks in experienced hosts are wide, evolution may be rapid. If these bottlenecks are tight or escape mutants are deleterious within untreated naive hosts (so $f_{\mathrm{mt}}$ is low), then $\mathcal{R}^w(0) < 1$ at the start of infection can result in slow evolution, since both replication and inoculation selection will be rare.

In the absence of mucosal neutralization with small $f_{\mathrm{mt}}$, $b \ll \frac{1}{f_{\mathrm{mt}}}$, and small $\mu$, the threshold for where inoculation selection becomes more common than replication selection with $\mathcal{R}^w(0) < 1$ is approximately

$$f_{\mathrm{mt}} b p_{\mathrm{sse}} > \mu \frac{b\mathcal{R}^w(0)}{1 - \mathcal{R}^w(0)} p_{\mathrm{sse}}$$

or simply:

$$f_{\mathrm{mt}} > \mu \frac{\mathcal{R}^w(0)}{1 - \mathcal{R}^w(0)} \tag{A29}$$

the left-hand side comes from *Equation A17*, which gives an probability that a mutant with $\mathcal{R}^m(0) > 1$ is inoculated. The right-hand side comes from *Equation 28*, which gives the probability that an escape mutant is generated in a declining but replicating virus population before that population goes extinct. $p_{\mathrm{sse}}$, which occurs on both sizes and cancels, is the probability that a mutant survives stochastic extinction. We have also ignored the fact the right-hand side should have a term representing the probability that inoculation selection does not occur, but as we are considering cases when both inoculation selection and replication selection are rare, that term is approximately one.

On average, $f_{\mathrm{mt}} > \mu$ due the asymmetry between forward and back-mutation rates when the mutant is rare. It follows that $f_{\mathrm{mt}} > \mu \frac{\mathcal{R}^w(0)}{1 - \mathcal{R}^w(0)}$ should hold when (A) the mutant not is too deleterious in the absence of positive selection (since this reduces $f_{\mathrm{mt}}$) or (B) $\mathcal{R}^w(0)$ is not close to 1 (since then many copies are made on average before the virus goes extinct). Alternatively, if $b$ is sufficiently large—on the order of $\frac{1}{f_{\mathrm{mt}}}$—selection on inoculated diversity will be the dominant mode of evolution simply because most inocula will include mutants.

Selection on inoculated diversity may be important in many host-pathogen systems, not just in influenza virus antigenic evolution. Some anti-microbial resistance in bacteria, for instance, is acquired through the uptake of preexisting plasmids, rather than through de novo mutation (*Perron et al., 2015*). Whether drug resistance emerges in an individual treated host will depend on whether any of the bacteria inoculated bear the needed plasmid.

## A8.2 Consequences for modeling influenza virus dynamics

Population size, population level antigenic diversification ('mutation') rate, and degree of population structure all substantially impact the proliferation of new influenza virus antigenic variants at the population level. It has been hard for population-level models to distinguish plausible hypotheses regarding evolutionary constraints on the virus, because achieving simultaneous realism across in all these effects while retaining model tractability has proven extremely difficult.

For epidemiological models of influenza virus evolution overall population size matters: epidemics in larger populations involve more total inoculations, and therefore more opportunities for new variant survival at the point of transmission to occur. Large host populations with high transmission rates, strong population connectivities, and recurrent epidemics should promote antigenic evolution. This helps explain why influenza virus antigenic evolution appears to occur disproportionately in east and southeast Asia (*Russell et al., 2008*).

It also reveals an important consideration for interpreting results from individual-based simulation models of global influenza virus evolution. Due to computational constraints, many such models must use host population sizes that are orders of magnitude smaller than the true global human population (e.g. $N = 40$ million [*Koelle and Rasmussen, 2015*], $N = 90$ million [*Bedford et al., 2012*]). Such models will have smaller numbers of inoculations than real-world influenza virus dynamics. They therefore run the risk of overestimating rates of strain extinction or, to avoid this, of overestimating per-infection virus diversification rates.

## A8.3 Egg and mouse passage experiments

The observation that immune escape can be reliably be selected for in egg (*Davis et al., 2018*) and mouse (*Hensley et al., 2009*) passage experiments and yet appears rare in reinfected experienced humans is unsurprising in light of our view that influenza virus evolution is limited more by the failure of antibody-mediated selection pressure and antigenic diversity to coincide than by the absence of either.

The egg experiments (*Davis et al., 2018*) amount to strong replication selection: the viruses were passaged in the presence of sera with highly-specific neutralizing antibodies, with these sera present at all times, including during the exponential growth phase within each egg.

The mouse passage experiments (*Hensley et al., 2009*) showed that serial passage in immune mice led to the appearance and fixation of antigenic escape mutants, but serial passage in naive mice did not. Several details of the experiment suggest that both replication selection and inoculation selection should have been more possible than in typical infected experienced humans.

First, the mice were intranasally inoculated with 50 $\mu l$ of homogenized lung isolate from the previous mouse in the passage chain at two days post-infection. This is likely a substantially more concentrated dose of virions than is inhaled through aerosol or even contact transmission (an inter-host bottleneck much wider than occurs in humans, to use the terminology of main text *Figure 4A*). This could produce a very large value of $v$, the number of virions encountering IgA, and potentially a large value of $b$, the final (cell infection) bottleneck as well—in the presence of sufficiently many virions, early cell infections might be sufficiently simultaneous as to produce have observably diverse within-host populations. All of this should tend to facilitate new variant survival and promotion to observable levels, but also to improve chances, relative to humans, that at least some old variant virions could survive alongside the new variant.

Second, inoculations were also carried out until a successful inoculation could be achieved. This further improves the chances of observing an escape mutant selected at the point of transmission. Finally, vaccinated mice in the passage chain were inoculated 10–21 days post-vaccination, meaning antibody levels could be higher than the memory baseline. This would be expected to strengthen inoculation selection and perhaps permit replication selection.

One detail in supplementary table 1 of the mouse passage study (*Hensley et al., 2009*) is particularly relevant. In the passaging of virus stock #3, two mutants were observed for the first time in the second vaccinated mouse in the chain, but neither at fixation. Both remained present for 5 further passages in experienced mice without either being lost or fixing. This suggests a much wider final bottleneck than occurs in naturally-infected humans (*McCrone et al., 2018*). It also suggests that replication selection is unlikely to have been at all strong if present at all: over repeated exponential

growth phases of two days with wide bottlenecks, even very small fitness differences can be easily amplified, so the more fit escape mutants should have fixed if they were subject to replication selection. It also suggests, as expected, that inoculation selection is weak if final bottlenecks are wide and neither antigenic variant is reliably neutralized. In that case, a long intermittent period of coexistence is possible.

An example calculation illustrates this: if $v = 1000$, $b = 10$, $f_{mt} = 0.5$, $\kappa_m/\kappa_w > 0.9$, and $\kappa_w < 0.8$ the mutant has an inoculation-selective advantage, but the expected mutant frequency at after the next transmission event approximately 0.58 or less. Letting $\bar{w} = E(x_w)$ and $\bar{m} = E(x_m)$:

$$\begin{aligned} \bar{w} &> v(1-f_{mt})(1-\kappa_w) &= (1000)(0.5)(0.20) = 100 \\ \bar{m} &> v(f_{mt})(1-\kappa_m) &= (1000)(0.5)(0.28) = 140 \end{aligned} \tag{A30}$$

And since $\bar{w}, \bar{m} \gg b$, the hypergeometric sampling is approximately binomial, and therefore the expected frequency of mutant is just $\frac{\bar{m}}{\bar{m}+\bar{w}} \approx 0.58$

While it is difficult to compare fitness differences during replication to fitness differences in mucosal neutralization, a very small fitness difference of $\delta = 0.5$ should raise the fitter type from frequency 0.5 to frequency 0.73 over the course of two days.

## A8.4 Magnitude of antigenic change and population level patterns

When antigenic effect size of mutations approaches zero, the effects of stochastic loss and mistimed selection pressure dominate, and inoculation and replication selection become sufficiently weak that true drift dominates as a force for introducing new antigenic variants. In such a regime, we expect influenza viruses to evolve gradually, fail to cause large epidemics, and possibly diversify, not unlike influenza B viruses (*Bedford et al., 2015*). In particular, slow spread enables a kind of immunological niche partitioning: if population immune histories are not spatially uniform because epidemiological spread is slow relative to antigenic diversification, multiple variants that are each locally favored in distinct areas could co-circulate and form separate lineages, as has happened with B/Victoria and B/Yamagata.

When antigenic jump size approaches the maximum size possible, such that all population immunity disappears each time there is an antigenic cluster transition (*Smith et al., 2004*), we expect influenza viruses to follow a strongly clock-like pattern with high-amplitude oscillations in case numbers. They would cause massive punctuated epidemics during jump years and then in the next year either select for a new variant or go extinct. There would be huge booms followed by one or more years of bust. In between, at intermediate degrees of escape, a punctuated but somewhat less clocklike pattern—such as the pattern observed in nature for A/H3N2 viruses (*Smith et al., 2004*)—becomes possible.

## A8.5 Preview substitutions and mutation limitation

'Preview' substitutions and polyphyletic cluster transitions also imply that the influenza virus is not typically mutation-limited at the population level—a substantial-effect escape mutant is usually accessible in sequence-space—but that the virus may nonetheless be highly constrained in its evolutionary trajectory. The virus repeatedly finds the same substitution as a solution to its antigenic evolutionary problem. This could occur either because only one escape mutant is available or because one of the available escape mutants provides substantially more immune escape (on average) than the others.

## A8.6 Selection on bottlenecked diversity at higher scales

Without an explanation for why within-host dynamics do not more frequently promote antigenic mutants, it is difficult to explain the slow pace and noisy trajectory of influenza virus evolution. But such a within-host explanation, though likely necessary, may not be sufficient. It is conceivable that in a sufficiently large and well-connected global host population, population-level antigenic selection favoring mutants with higher $\mathfrak{R}_e$ would lead even small-effect antigenic mutants rapidly to emerge and fix.

One important mechanism by which evolution might be also slowed at higher scales is analogous to the within-host dynamic of founder effects and inoculation selection: mutant lineages may be frequently lost (though perhaps selectively favored) at the bottleneck that occurs between distinct influenza virus epidemics.

Prior modeling work has found that population-level host competition between pathogen variants reduces the probability that a fit mutant variant causes a large epidemic in the population in which it first emerges (*Hartfield and Alizon, 2014*). This is likely to be relevant in influenza virus epidemics, since the virus is thought to provoke short-term strain-transcending (i.e. variant-transcending) immunity (*Ferguson et al., 2003*).

We propose a previously unexplored consequence of this argument: successful establishment of a mutant lineage at the population level requires that the mutant lineage be exported to a new host subpopulation where susceptible hosts are common. This is rare but potentially selective sampling event.

When a new variant lineage emerges at the population level, it is likely to be surrounded by many propagating old variant lineages due to the generator-selector dynamic discussed in the main text (see main text *Figure 4B,C*). Moreover, most mutant lineage generation events will occur as an epidemic is peaking, since that is when the most inoculations occur. The consequence is that most generated population-level mutant lineages will encounter severe competition for susceptible hosts (especially if we consider realistic, spatial host contact networks rather than well-mixed epidemic models). So a generated mutant lineage is unlikely to account for many cases—or even necessarily more than one case—in the epidemic in which it emerges.

Many local influenza virus epidemics are likely to be evolutionary dead ends with no cases exported that establish chains of transmission in other locations. Because only a minority of the human population is likely to travel to or from the site of any particular epidemic, the majority of virus diversity generated within each epidemic is likely to be lost in between-epidemic bottlenecks. Conditional on being exported, new variant lineages could have competitive advantages over old variant lineages because they spread should spread more rapidly and go stochastically extinct less easily due to their higher $\Re_e$. It follows that there is analogous dynamic to inoculation selection at the population level: mutant lineages are rarely exported from one sub-population (host, epidemic) to another, but conditional on being exported, they have an advantage in any potential competition with exported old variant lineages.

The rate of proliferation of mutants at the population level, then, may be limited by the rarity of early generation of mutant lineages during an epidemic (analogous to the rarity of very early within-host de novo generation of mutants) and by the rarity of successful mutant lineage exportation when generation is not early (analogous to the rarity of mutant virions surviving the transmission bottleneck between hosts).

We aim to explore this argument with a formal mathematical model in future work.

## A9 Mathematical derivations in full

### A9.1 Full derivation of the within-host replicator equation

The derivation in this section establishes *Equation 12*:

$$\frac{\mathrm{d}f_m}{\mathrm{d}t} = f_m(1-f_m)([g_m(t)-g_w(t)]-[d_m(t)-d_w(t)])$$

### Derivation

We note that:

$$f_m(t) := \frac{V_m(t)}{V_{\mathrm{tot}}(t)}$$

where $V_{\mathrm{tot}}(t) = V_w(t) + V_m(t)$. Let $\dot{V}_i$ denote $\frac{\mathrm{d}V_i}{\mathrm{d}t}$. Note that $\dot{V}_i = V_i\alpha_i(t)$ where $\alpha_i(t) = g_i(t) - d_i(t)$, and that $\frac{V_w(t)}{V_{\mathrm{tot}}(t)} = 1 - f_m(t)$. By the quotient rule:

$$\begin{aligned}
\frac{\mathrm{d}f_m}{\mathrm{d}t} &= \frac{V_{\mathrm{tot}}\dot{V}_m - V_m(\dot{V}_m + \dot{V}_w)}{V_{\mathrm{tot}}^2} \\
&= \frac{V_{\mathrm{tot}}(\alpha_m V_m) - V_m(\alpha_m V_m + \alpha_w V_w)}{V_{\mathrm{tot}}^2}
\end{aligned}$$

Dividing through by $V_{\mathrm{tot}}^2$ yields:

$$\alpha_m f_m - f_m(\alpha_m f + \alpha_w(1 - f_m))$$

$$= \alpha_m f_m - \alpha_m f_m^2 - \alpha_w f_m(1 - f_m)$$

$$= \alpha_m f_m(1 - f_m) - \alpha_w f_m(1 - f_m)$$

$$= f_m(1 - f_m)(\alpha_w - \alpha_m)$$

$$= f_m(1 - f_m)([g_m(t) - g_w(t)] - [d_m(t) - d_w(t)])$$

## A9.2 Replicator equation with symmetric mutation

With symmetric mutation at a rate μ, the replicator equation remains calculable.

### Derivation

Given symmetric mutation, $\dot{V}_i = V_i \alpha_i(t) + \mu g_j(t) V_j$ where $\alpha_i(t) = (1 - \mu)g_i(t) - d_i(t)$

Substituting in to the previous derivation yields:

$$\begin{aligned}
\frac{\mathrm{d}f_m}{\mathrm{d}t} &= \frac{V_{\mathrm{tot}}(\alpha_m V_m + \mu g_w(t)V_w) - V_m(\alpha_m V_m + \mu g_w(t)V_w + \alpha_w V_w + \mu g_m(t)V_m)}{V_{\mathrm{tot}}^2} \\
&= \alpha_m f_m + \mu g_w(t)(1 - f_m) - f_m(\alpha_m f_m + \mu g_w(t)(1 - f_m) + \alpha_w(1 - f_m) + \mu g_m(t)f_m) \\[6pt]
&= \alpha_m f_m + \mu g_w(t)(1 - f_m) - \alpha_m f_m^2 - \mu g_w(t)f_m(1 - f_m) - \alpha_w f_m(1 - f_m) - \mu g_m(t)f_m^2 \\
&= \alpha_m f_m(1 - f_m) - \alpha_w f_m(1 - f_m) + \mu\left(g_w(t)(1 - f_m) - g_w(t)f_m(1 - f_m) - g_m(t)f_m^2\right) \\
&= f_m(1 - f_m)(\alpha_m - \alpha_w) + \mu(g_w(t) - 2g_w(t)f_m + g_w(t)f_m^2 - g_m(t)f_m^2)
\end{aligned}$$

Note that if $g_w(t) = g_m(t) = g(t)$, this simplifies to:

$$\frac{\mathrm{d}f_m}{\mathrm{d}t} = f_m(1 - f_m)[d_m(t) - d_w(t)] + \mu g(t)(1 - 2f_m)$$

## A9.3 Replicator equation with one-way mutation
### Derivation

To find the case with one-way mutation, we simply let all $\mu g_m(t)V_m$ terms from the symmetric mutation replicator equation be zero. This yields:

$$\frac{\mathrm{d}f_m}{\mathrm{d}t} = f_m(1 - f_m)(\alpha_m - \alpha_w) + \mu g_w(t)(1 - 2f_m + f_m^2) \tag{A31}$$

## A9.4 Derivation of $t^*(x, t)$

This establishes the expression for the time $t^*$ by which a mutant must emerge to reach at least frequency $x$ by time $t$ given in **Equation 25** of the Materials and methods.

$$t^*(x, t) = \begin{cases} t^*_-(x, t) & t^*_-(x, t) < t \,\mathrm{and}\, t^*_-(x, t) \le t_M \\ t^*_+(x, t) & t^*_+(x, t) < t \,\mathrm{and}\, t^*_-(x, t) > t_M \\ t & \mathrm{otherwise} \end{cases}$$

where:

$$t^*_-(x, t) = \frac{\ln\left[\frac{1-x}{x}\exp(\delta(t - t_M)) + 1\right] - \ln b}{g_0 - d_v}$$

$$t_+^*(x,t) \approx \frac{\ln(\frac{1-x}{x}) - \ln(b) + \delta t - c_w k t_M}{g_0 - d_v - c_w k + \delta}$$

## Derivation

The frequency of the mutant at time $t$ depends on the quantity

$$\psi(t) = \exp(\delta(t - \max\{t_M, t_e\}))$$

Neglecting ongoing forward mutation, the mutant must emerge at some frequency $f_e$ in order to reach frequency $x$ by time $t$:

$$\frac{\psi(t)}{\psi(t) + f_e^{-1} - 1} \geq x$$

Solving for $f_e^{-1}$ yields:

$$f_e^{-1} \leq \left(\psi(t)\frac{1-x}{x}\right) + 1 \tag{A32}$$

$f_e^{-1}$ is determined by the number of old variant virions at time $t_e$. Early in infection, the old variant population grows near-exponentially at a rate $G_0 = g_0 - d_v$. There are two cases: either time of new variant emergence (first mutation to produce a new variant lineage that evades stochastic extinction) occurs before the antibody response is present ($t_e \leq t_M$) or it emerges once the antibody response is already present ($t_e > t_M$).

### A9.4.1 Case 1: $t_e \leq t_M$

It follows that if $t_e \leq t_M$, $f_e^{-1} \approx b\exp((g_0 - d_v)t_e) = b\exp(G_0 t_e)$.

$$be^{G_0 t_e} \leq \left(\psi(t)\frac{1-x}{x}\right) + 1$$

Taking the natural log of both sides and subtracting off $\ln b$ from both sides:

$$G_0 t_e \leq \ln\left(\left(\psi(t)\frac{1-x}{x}\right) + 1\right) - \ln b$$

$$t_e \leq \frac{\ln\left(\psi(t)\frac{1-x}{x} + 1\right) - \ln b}{G_0}$$

This establishes a value for $t^*$ when $t_e < t_M$. In that case, $\psi(t) = \exp(\delta(t - t_M))$, and:

$$t_-^*(x,t) = \frac{\ln\left[\frac{1-x}{x}\exp(\delta(t - t_M)) + 1\right] - \ln b}{g_0 - d_v} \tag{A33}$$

### 9.4.2 Case 2: $t > t_e > t_M$

If $t > t_e > t_M$, we must change our estimate of $f_e^{-1}$, because the old variant population grows more slowly in the presence of an antibody response. The approximate growth rate is $G_1 = G_0 - c_w k$. For $t_e > t_M$:

$$f_e^{-1} = b\exp[G_0 t_M]\exp[G_1(t_e - t_M)] = b\exp[c_w k t_M + G_1 t_e] \tag{A34}$$

The expression from **Equation A32** no longer yields a closed form solution; it is now a transcendental equation, because $\psi(t)$ now contains $t_e$ terms: $\psi(t) = \exp(\delta(t - t_e))$.

To deal with this, we make the approximation:

$$f_m(t) = \frac{\psi(t)}{\psi(t) + f_e^{-1}}$$

This approximation is excellent unless the initial frequency $f_e$ is large, and it always yields an underestimate of $f_m(t)$, . This is desirable; we would like to be pessimistic about the probability of replication selection given early $t_M$ and early $t_e > t_M$ to avoid biasing ourselves in favor of our hypothesis (namely, that replication selection is unrealistically common if $t_M$ is early).

Letting $f_m(t) = x$, the desired frequency at $t$, we wish to solve:

$$x \leq \frac{\psi(t)}{\psi(t) + f_e^{-1}}$$

We find:

$$f_e^{-1} x + \psi(t) x \leq \psi(t)$$

$$f_e^{-1} \leq \psi(t) \frac{1 - x}{x}$$

$$b \exp(c_w k t_M + G_1 t_e) \leq \exp(\delta(t - t_e)) \frac{1 - x}{x}$$

$$\ln(b) + c_w k t_M + G_1 t_e \leq \delta t - \delta t_e + \ln(\frac{1 - x}{x})$$

$$t_e(G_1 + \delta) \leq \ln(\frac{1 - x}{x}) - \ln(b) + \delta t - c_w k t_M$$

$$t_e \leq \frac{\ln(\frac{1-x}{x}) - \ln(b) + \delta t - c_w k t_M}{G_1 + \delta}$$

So when $t_e > t_M$:

$$t_+^*(x, t) \approx \frac{\ln(\frac{1-x}{x}) - \ln(b) + \delta t - c_w k t_M}{g_0 - d_v - c_w k + \delta} \tag{A35}$$

Note that since we used an underestimate of $f_m(t)$, this approximate $t^*$ is a lower bound for the true $t^*$; it may be that the new variant can actually emerge later and still successfully be replication-selected to the desired frequency $x$.

Finally, it may be that $t_+^*(x, t) > t$ and $t_-^*(x, t) > t$. This indicates that the mutant will be at at least frequency $x$ if it emerges at $t$ itself. In that case, we therefore have $t^*(x, t) = t$.

Combining:

$$t^*(x, t) = \begin{cases} t_-^*(x, t) & t_-^*(x, t) < t \text{ and } t_-^*(x, t) \leq t_M \\ t_+^*(x, t) & t_+^*(x, t) < t \text{ and } t_-^*(x, t) > t_M \\ t & \text{otherwise} \end{cases} \tag{A36}$$

## A9.5 Derivation of the closed form for $p_{\text{cib}}$

The derivation in this section establishes *Equation 35*:

$$p_{\text{cib}}(\kappa_w, f_{\text{mt}}, wb) = E(p_{\text{cib}}(x_w, 1, b)) = (1 - e^{-\bar{w}}) \frac{b}{\bar{w}} + e^{-\bar{w}} \sum_{j=0}^{b-1} \frac{\bar{w}^j}{j!} (1 - \frac{b}{j+1})$$

where $\bar{w} = v(1 - f_{\text{mt}})(1 - \kappa_w)$ is the mean number of old variant virions present after IgA neutralization.

### Derivation

$$p_{\text{cib}}(x_w, 1, b) = \frac{b}{x_w + 1}$$

$$
\begin{aligned}
E(p_{\mathrm{cib}}(x_w,1,b)) &= b\sum_{k=0}^{\infty}\frac{\bar{w}^k}{k!}e^{-\bar{w}}\frac{1}{k+1}\\
&= \frac{b}{\bar{w}}\sum_{k=0}^{\infty}\frac{\bar{w}^{k+1}}{(k+1)!}e^{-\bar{w}}\\
&= \frac{b}{\bar{w}}\sum_{j=1}^{\infty}\frac{\bar{w}^{j}}{j!}e^{-\bar{w}}
\end{aligned}
$$

We know that:

$$
e^{-\lambda}+\sum_{j=1}^{\infty}\frac{\lambda^j}{j!}e^{-\lambda}=1
$$

since the Poisson is a proper probability distribution and therefore:

$$
\sum_{j=1}^{\infty}\frac{\lambda^j}{j!}e^{-\lambda}=1-e^{-\lambda}
$$

So substituting in:

$$
\frac{b}{\bar{w}}\sum_{j=1}^{\infty}\frac{\bar{w}^j}{j!}e^{-\bar{w}}=\frac{b}{\bar{w}}(1-e^{-\bar{w}})
$$

This is exact when $b=1$, but in other cases $p_{\mathrm{cib}}(x_w,1,b)$ is more properly given by $\max\{\frac{b}{x_w+1},1\}$, since whenever we have $x_w+1<b$, a new variant survives with certainty. So for each such $x_w<b-1$ we need to add on correction term equal to $1-\frac{b}{x_w+1}$, weighted by the probability of that $x_w$ takes on that value: $e^{-\bar{w}}\frac{\bar{w}^{x_w}}{x_w!}$.

Summing from $x_w=0$ to $x_w=b-1$ and factoring out $e^{-\bar{w}}$ gives us the complete correction term:

$$
e^{-\bar{w}}\sum_{j=0}^{b-1}\frac{\bar{w}^j}{j!}(1-\frac{b}{j+1})
$$

Adding that to the expression derived above completes the derivation.

## A9.6 $p_{\mathrm{cib}}=1$ only if there is no competition

$p_{\mathrm{cib}}<1$ if $\kappa_w<1$ and $f_{\mathrm{mt}}<1$, and $p_{\mathrm{cib}}=1$ if $\kappa_w=1$ or $f_{\mathrm{mt}}=1$.

### Derivation

This is most easily seen by rewriting $p_{\mathrm{cib}}$ as an infinite sum:

$$
p_{\mathrm{cib}}(\kappa_w,v,b)=e^{-\bar{w}}[\sum_{j=0}^{b-1}\frac{\bar{w}^j}{j!}1+\sum_{j=b}^{\infty}\frac{\bar{w}^j}{j!}\frac{b}{j+1}] \tag{A37}
$$

If $\bar{w}=v(1-f_{\mathrm{mt}})(1-\kappa_w)=0$, we have $p_{\mathrm{cib}}=1$, as expected. Since $v>0$, this occurs either when there are no old variant virions ($f_{\mathrm{mt}}=1$) or when the old variant is neutralized with probability 1 ($\kappa_w=1$). Otherwise, $\bar{w}>0$ and we therefore have:

$$
p_{\mathrm{cib}}=e^{-\bar{w}}\left[\sum_{j=0}^{b-1}\frac{\bar{w}^j}{j!}1+\sum_{j=b}^{\infty}\frac{\bar{w}^j}{j!}\frac{b}{j+1}\right]<e^{-\bar{w}}\left[\sum_{j=0}^{b-1}\frac{\bar{w}^j}{j!}1+\sum_{j=b}^{\infty}\frac{\bar{w}^j}{j!}1\right]=1 \tag{A38}
$$

The last step uses the fact that the Poisson distribution is a proper probability distribution, and therefore sums to 1.

## A9.7 $p_{\mathrm{cib}}$ is decreasing in $\bar{w}$

$p_{\mathrm{cib}}$ is decreasing in $\bar{w}$: the more competing old variant virions, the less likely the new variant is to pass through the cell infection bottleneck.

## Derivation

For $b = 1$,

$$\frac{\mathrm{d}p_{\mathrm{cib}}}{\mathrm{d}\bar{w}} = \frac{e^{-\bar{w}}(\bar{w} - e^{\bar{w}} + 1)}{\bar{w}^2}$$

This is negative for $\bar{w} > 0$, as $e^x > x + 1$ for $x > 0$.

That $p_{\mathrm{cib}}$ is decreasing in $\bar{w}$ for $b > 1$ can be seen by writing:

$$
\begin{aligned}
p_{\mathrm{cib}} \quad &= e^{-\bar{w}} S(\bar{w}) \\
S(\bar{w}) \quad &:= \sum_{j=0}^{b-1} \frac{\bar{w}^j}{j!} + \sum_{j=b}^{\infty} \frac{\bar{w}^j}{j!} \frac{b}{j+1}
\end{aligned}
\tag{A39}
$$

This can be rewritten in two useful ways:

$$S(\bar{w}) = 1 + \sum_{j=1}^{b-1} \frac{\bar{w}^j}{j!} + \sum_{j=b}^{\infty} \frac{\bar{w}^j}{j!} \frac{b}{j+1} \tag{A40}$$

$$S(\bar{w}) = \sum_{j=0}^{b-2} \frac{\bar{w}^j}{j!} + \sum_{j=b-1}^{\infty} \frac{\bar{w}^j}{j!} \frac{b}{j+1} \tag{A41}$$

This uses the fact that $\frac{b}{j+1} = 1$ when $j = b - 1$.

We begin by showing that $S'(\bar{w}) < S(\bar{w})$ when $\bar{w} > 0$. We differentiate **Equation A40** with respect to $\bar{w}$:

$$
\begin{aligned}
S'(\bar{w}) \quad &= 0 + \sum_{j=1}^{b-1} \frac{\bar{w}^{j-1}}{(j-1)!} + \sum_{j=b}^{\infty} \frac{\bar{w}^{j-1}}{(j-1)!} \frac{b}{j+1} \\
&= \sum_{j=0}^{b-2} \frac{\bar{w}^j}{j!} + \sum_{j=b-1}^{\infty} \frac{\bar{w}^j}{j!} \frac{b}{j+2}
\end{aligned}
\tag{A42}
$$

When $\bar{w} > 0$, the first $b - 1$ terms in $S'(\bar{w})$ are equal to the corresponding terms in **Equation A41** for $S(\bar{w})$. Each subsequent term is smaller in $S'(\bar{w})$ than in $S(\bar{w})$, since $\frac{b}{j+2} < \frac{b}{j+1}$ for $b, j > 0$. So we have established that $S'(\bar{w}) < S(\bar{w})$ for $b > 1$.

Now we find $\frac{\mathrm{d}p_{\mathrm{cib}}}{\mathrm{d}\bar{w}}$:

$$\frac{\mathrm{d}p_{\mathrm{cib}}}{\mathrm{d}\bar{w}} = -e^{-\bar{w}} S(\bar{w}) + e^{-\bar{w}} S'(\bar{w}) \quad = e^{-\bar{w}}[S'(\bar{w}) - S(\bar{w})] < 0 \tag{A43}$$

since $e^{-\bar{w}} > 0$ and $S'(\bar{w}) - S(\bar{w}) < 0$.

It follows that $p_{\mathrm{cib}}$ is decreasing in $\bar{w}$.

## A9.8 $p_{\mathrm{cib}}$ is increasing in $b$

### Derivation

This follows from the infinite sum expression for $p_{\mathrm{cib}}$ (**Equation A37**). We will show that $p_{\mathrm{cib}}(b + 1) > p_{\mathrm{cib}}(b)$ for integers $b \geq 1$ by showing that $p_{\mathrm{cib}}(b + 1) - p_{\mathrm{cib}}(b) > 0$

$$
\begin{aligned}
p_{\mathrm{cib}}(\kappa_w, v, b) \quad &= e^{-\bar{w}} \Big[ \sum_{j=0}^{b-1} \frac{\bar{w}^j}{j!} 1 + \sum_{j=b}^{\infty} \frac{\bar{w}^j}{j!} \frac{b}{j+1} \Big] \\
p_{\mathrm{cib}}(\kappa_w, v, b+1) \quad &= e^{-\bar{w}} \Big[ \sum_{j=0}^{b} \frac{\bar{w}^j}{j!} 1 + \sum_{j=b+1}^{\infty} \frac{\bar{w}^j}{j!} \frac{b+1}{j+1} \Big] \\
&= e^{-\bar{w}} \Big[ \sum_{j=0}^{b-1} \frac{\bar{w}^j}{j!} 1 + \sum_{j=b}^{\infty} \frac{\bar{w}^j}{j!} \frac{b+1}{j+1} \Big]
\end{aligned}
\tag{A44}
$$

This uses the fact that $\frac{b+1}{j+1} = 1$ when $j = b$.

For any $0 \leq \kappa_w \leq 1$, any $v \geq 1$, and $b \geq 1$, consider $p_{\mathrm{cib}}(b+1) - p_{\mathrm{cib}}(b)$:

$$
\begin{aligned}
p_{\mathrm{cib}}(b+1) - p_{\mathrm{cib}}(b) &= e^{-\bar{w}} \Big[ \sum_{j=b}^{\infty} \frac{\bar{w}^j}{j!} \frac{(b+1)-b}{j+1} \Big] \\
&= e^{-\bar{w}} \Big[ \sum_{j=b}^{\infty} \frac{\bar{w}^j}{j!} \frac{1}{j+1} \Big] \\
&> 0
\end{aligned}
\tag{A45}
$$

since this is a sum of positive terms, multiplied by a positive number.

## A9.9 $p_{\mathrm{surv}}/p_{\mathrm{drift}}$ is decreasing in $b$
### Derivation

We follow an approach similar to A9.8, showing that $\frac{p_{\mathrm{surv}}(b+1)}{p_{\mathrm{drift}}(b+1)} - \frac{p_{\mathrm{surv}}(b)}{p_{\mathrm{drift}}(b)} < 0$ for integer $b \geq 1$.

$$
\begin{aligned}
\frac{p_{\mathrm{surv}}(b)}{p_{\mathrm{drift}}(b)} &\approx \frac{v}{b}(1-\kappa_m) p_{\mathrm{cib}} \\
&= \frac{v}{b}(1-\kappa_m) e^{-\bar{w}} \left[ \sum_{j=0}^{b-1} \frac{\bar{w}^j}{j!} 1 + \sum_{j=b}^{\infty} \frac{\bar{w}^j}{j!} \frac{b}{j+1} \right] \\
&= v(1-\kappa_m) e^{-\bar{w}} \left[ \sum_{j=0}^{b-1} \frac{\bar{w}^j}{j!} \frac{1}{b} + \sum_{j=b}^{\infty} \frac{\bar{w}^j}{j!} \frac{1}{j+1} \right]
\end{aligned}
$$

Similarly:

$$
\begin{aligned}
\frac{p_{\mathrm{surv}}(b+1)}{p_{\mathrm{drift}}(b+1)} &\approx \frac{v}{b+1}(1-\kappa_m) e^{-\bar{w}} \left[ \sum_{j=0}^{b} \frac{\bar{w}^j}{j!} 1 + \sum_{j=b+1}^{\infty} \frac{\bar{w}^j}{j!} \frac{b+1}{j+1} \right] \\
&= \frac{v}{b+1}(1-\kappa_m) e^{-\bar{w}} \left[ \sum_{j=0}^{b-1} \frac{\bar{w}^j}{j!} 1 + \sum_{j=b}^{\infty} \frac{\bar{w}^j}{j!} \frac{b+1}{j+1} \right] \\
&= v(1-\kappa_m) e^{-\bar{w}} \sum_{j=0}^{b-1} \frac{\bar{w}^j}{j!} \frac{1}{b+1} + \sum_{j=b}^{\infty} \frac{\bar{w}^j}{j!} \frac{1}{j+1}
\end{aligned}
$$

We have again used the fact that $\frac{b+1}{j+1} = 1$ when $j = b$.
Taking the difference of the approximate ratios:

$$
\frac{p_{\mathrm{surv}}(b+1)}{p_{\mathrm{drift}}(b+1)} - \frac{p_{\mathrm{surv}}(b)}{p_{\mathrm{drift}}(b)} \approx v(1-\kappa_m) e^{-\bar{w}} \sum_{j=0}^{b-1} \frac{\bar{w}^j}{j!} \left( \frac{1}{b+1} - \frac{1}{b} \right) < 0
$$

That the expression is negative follows from the fact that all terms in the sum are negative, since $\frac{1}{b+1} < \frac{1}{b}$ for $b \geq 1$ while $\frac{\bar{w}^j}{j!}$ is always positive. The terms multiplying the sum are all positive.
It follows that $p_{\mathrm{surv}}/p_{\mathrm{drift}}$ is decreasing in $b$.

## A9.10 In the absence of selection, the probability of surviving the bottleneck is approximately linear for large $v$ and small $f_{\mathrm{mt}}$
### Derivation

If $\kappa_w = \kappa_m = 0$, we have:

$$
p_{\mathrm{inoc}} = (1 - e^{-v f_{\mathrm{mt}}})
$$

$$
\bar{w} = v(1 - f_{\mathrm{mt}}) \approx v
$$

$$
e^{-\bar{w}} \approx 0
$$

$$p_{\mathrm{cib}} \approx \frac{b}{v}$$

$$p_{\mathrm{surv}} \approx \left(1 - e^{-vf_{\mathrm{mt}}}\right)\left(\frac{b}{v}\right)$$

And since $-vf_{\mathrm{mt}}$ is small, we can use the linear approximation to the exponential:

$$p_{\mathrm{surv}} \approx vf_{\mathrm{mt}}\frac{b}{v} = bf_{\mathrm{mt}}$$

