## [Decision Letter]

**Acceptance summary:**

The manuscript presents a theoretical study that bridges the global and the within-host evolution of the influenza virus. The key finding is that the asynchrony between within-host viral growth and the delayed antibody response following an infection can explain the pattern of antigenic evolution for influenza, where selection for new variants is strong at the population level but not within individuals. This work is conceptually important and provides a theory-driven hypothesis to explain the observed antigenic evolution of influenza across scales.

**Decision letter after peer review:**

[Editors’ note: the authors submitted for reconsideration following the decision after peer review. What follows is the decision letter after the first round of review.]

Thank you for submitting your work entitled "Asynchrony between virus diversity and antibody selection limits influenza virus evolution" for consideration by *eLife*. Your article has been reviewed by a Senior Editor, a Reviewing Editor, and three reviewers. The following individuals involved in review of your submission have agreed to reveal their identity: Christopher J R Illingworth (Reviewer #1); Daniel B Weissman (Reviewer #3).

Our decision has been reached after consultation between the reviewers. Based on these discussions and the individual reviews below, we regret to inform you that your work will not be considered further for publication in *eLife*.

This manuscript brings together a lot of good ideas and makes useful points about selection on influenza at transmission and a delayed within-host response. The reviewers agree that the proposed model is insightful and in principle could be used to understand the observed patterns of within-host viral diversity and the patterns of global influenza evolution. However, the reviewers also raised substantial concerns regarding the limited connection between the model and data and many of the speculations in the manuscript that are not fully supported by data. In addition, reviewers also suggest that paper should be rewritten in a sufficiently clear manner for a general audience to understand, and for this work to be of broad interest. Overall, the reviews propose a substantial re-writing and further analysis and inclusion of data, in order for the manuscript to be considered for publication in *eLife*. This, however, may seem beyond the scope of the current manuscript. Therefore, we agreed to reject the paper in its current form but we encourage the authors to resubmit (as an entirely new submission but reference this submission) if they can (and want to) address the concerns detailed below.

Reviewer #1:

This paper addresses the observation that, while global influenza populations are strongly influenced by selection for antigenic novelty, within-host populations. It describes a model in which within-host selection is active, but temporally delayed, while selection based on the immune history of an individual acts at the time of transmission, during the establishment of an infection.

1) I have two preliminary comments on the style in which this is written, as opposed to the content of the paper. Firstly, my experience reading this was that I found the main text difficult to follow until the point at which I had gained familiarity with the details of the model. For example, once I knew what \kappa_w, p_C, c, and k referred to I could understand the legend of Figure 3, though it took some time for me to get there. I suggest, if possible, rewriting the main text so as to be accessible in its own right to a casual reader. Secondly, I thought that the choice of data to be presented in figures was odd. The paper presents a model of viral evolution. As such, the interest (for me) was the extent to which the model replicates the observed behaviour of influenza virus in the real world. However, only Figure 1A and 1B describe any 'real world' data. Figure 4A is conceptual and useful, but the remainder of the figures exist in 'model space', showing how this or that parameter affects the behaviour of which part of the model. I'm not against model-to-model comparison (Figure 4E-H are useful though I'd personally keep only the top parts), but suggest removing anything from the main text which does not directly and clearly illustrate some key part of the model, or which does not show some direct comparison of the behaviours of different models as they relate directly to observations of the real-world system.

2) The claim is made that 'selection for new antigenic variants acts principally at the point of initial virus inoculation'. I have some questions and concerns around this claim.

- To what extent has antigenic selection during the establishment of infection been experimentally documented? The primary reference given here is that of Wang, 2017. However, while Wang is clear that antibodies in the mucus can be influenza-specific (as opposed to being specific to another virus) it did not seem so clear that these antibodies are strain-specific in the sense of identifying viruses that differed by a single mutation.

- With this in the background I was not clear about the first of the claimed key results, that 'sIgA antibody neutralisation protects experienced hosts against reinfection'. It is certainly true is that a model incorporating selection during the transmission process (plus delayed within-host selection) is consistent with a range of observed data describing influenza infection and evolution. However, this is not the same as what is claimed. The hypothesis of the importance of sIgA antibody neutralisation is shown to be more consistent with the data than other models, but that does not imply that the hypothesis is true.

- Given the modelling that has been done, it would seem straightforward to evaluate this claim further. Within-host selection is delayed, but not entirely absent. As such, the expected frequency of a novel antigenic variant at the time of transmission to a new host will in the mean be higher than it would otherwise be under pure neutrality. It should be possible compare this change in the expected variant frequency to that produced by the transmission process (illustrated in Figure 4E). Averaging over stochastic effects, what proportion of the selection is due to within-host processes and what is due to selection during transmission? In the manuscript, it is claimed without further details that within-host evolution is 'dominated' by inoculation selection. What does this really mean?

3) On the model of within-host viral load, this paper follows a commonly applied model of (roughly speaking) exponential growth followed by exponential decline. However, Pawelek et al. (2012) argue that a more bimodal pattern of behaviour is more realistic, a second peak arising from the re-emergence of susceptible cells that have been made refractory to infection by interferon. Would this increase the effect of within-host selection? Further, the work of Tsang et al. (2015) notes a proportion of transmission events occurring four or more days post the onset of symptoms; if within-host selection kicks in two days after infection it seems likely that within-host selection has a non-negligible effect even if events at transmission are the primary driver of selection.

4) I don't believe that the model presented is a good one for infection in immunocompromised individuals. To the best of my knowledge the oscillations shown in Figure 3B and C have not been observed in clinical cases. My suspicion is that the model ignores some of the complexities of within-host infection in a way that gives a reasonable picture of a typical infection, but that falls short when it comes to longer-term infections. The point being made here is that selection for antigenic variants can be effective during longer infections; I would accept this point fairly readily, but the data of Figure 3 did not make me particularly more convinced.

5) It is claimed that Figure 5 shows that proposed model of global influenza evolution better mimics empirical observations of the global population with regard to changes in the antigenic type of the virus. However, while it is clear from the figure that different models give different results, no indication of the observed behaviour of the global population is given in the Figure. The claim may be true. However, what is the observed distribution of changes in the global population to which Figure 5D provides a better match?

6) I was confused by Figure 7. Based on the methods I think that the term emergence describes the appearance of a variant in the population in such a way that it will not subsequently die out through genetic drift. Figure 7 then shows the distribution of emergence times under the case of an immediate recall response and a delayed recall response. My expectation would be that under an immediate recall response, positive selection would begin to apply to the variant earlier, making its emergence more likely to happen sooner, but the figure shows the opposite effect. I may have missed something in the explanation.

7) Appendix 6: To the best of my understanding, the idea that large effect antigenic changes can occur in multiple backgrounds is not contradicted by the work of Koelle and Rasmussen. In that work a distinction is made between 'good' and 'bad' genetic backgrounds, characterised by the extent of mutational load in a sequence, however, there can be multiple 'good' backgrounds or at least, multiple backgrounds good enough for a variant to appear multiple times in the global population before fixation. As another thought here, is the work of Strelkowa and Lassig, (2011) relevant as a proposed explanation for the pattern of global viral evolution, competition?

8) Appendix 7: The concept of selection acting both in the transmission process and during within-host replication is not entirely novel; Lumby et al. (2018) make this distinction in the context of influenza transmission and within-host growth (and further discuss the idea of a selective bottleneck). The idea that selection during the transmission process could play a role in the adaptation of global influenza populations has been raised very recently (Lumby et al., 2020) if not beforehand. This work therefore builds upon thinking about selection during viral transmission in the published literature.

Reviewer #2:

Summary:

In this paper, the authors develop a model that reconciles the observed lack of antigenic evolution in influenza on the intra-patient scale with the strong evolutionary pressure observed on the global scale. Using this model, the authors infer a biological mechanism for this disparity. They claim that the asynchrony between peak viral loads and the adaptive immune response prevents selection during viral replication in a host. Instead, they propose that under a realistic set of infection and immune parameters, selection happening at the point of inoculation by matched sIgA antibodies would better explain intra-host and global evolutionary patterns.

Although others have similarly proposed that influenza evolution ordinarily takes place between hosts as opposed to within hosts, this paper models this idea in a much more systematic way than has been done before.

Essential revisions:

The paper is virtually all modeling except for a bit of deep sequencing data in Figure 1. Yet the authors strongly posit specific molecular mechanisms (such as secretory IgA) as driving the selection. While these mechanisms certainly seem plausible, the model is really just distinguishing between selection at transmission versus selection within-host after transmission. It would therefore be better to phrase more of the paper in those terms, and then perhaps in Introduction and Discussion section speculate about specific mechanisms that would drive different forms of selection.

Overall, I find these models informative and plausible. However, the comparison of the model predictions to real data is fairly limited and mostly qualitative.

Can the authors comment on the dynamics of infection in children? There is some evidence that infection times are longer in children than adults (Ng et al., 2016). Does this suggest that 'replication selection' could occur in children? If so, children make up a substantial proportion of the world population and would likely have a large impact on global transmission dynamics.

How would the model change in cases where influenza infects the lower respiratory tract? This would change the size of the target cell population substantially.

The authors should be clearer that when speaking about 'wild-type' and 'mutant' viruses, they refer to antigenicity and not the genotype. For example, in subsection “Model overview”, when the authors talk about 'within-host virus diversity' are they talking about the random accumulation of mutations or the random accumulation of antigenic variants? Also in the Results section, there would be a difference in the probability of a new variant infection and a new antigenic variant infection, so they should be clear about which they are referring to.

How does a cellular immune response affect viral titer? Do T-Cells help to mount an immune response? I think the authors are using adaptive immunity and antibody-mediated immunity interchangeably when they aren't fully interchangeable. In particular, vaccination and infection might elicit different levels of T-cell immunity.

How 'target-cell' limited is influenza really in practice? If it is, how would infections in immunocompromised individuals occur? I assume the rate of cellular replenishment is the same, and that rate of target cell depletion would also be the same.

Reviewer #3:

This is an interesting manuscript that uses mathematical modeling to argue that the timing of within-host selection is crucial to understanding antigenic evolution in influenza. In particular, it argues that selection by antibodies on mucosal surfaces during initial infection is the key driver of adaptation. This seems important, although I had a hard time following the argument in places.

Essential revisions:

1) I would appreciate a clear, concise statement of which aspects of observed flu dynamics (e.g., reinfected hosts without escape mutants, global patterns of evolution, etc.) the model hits and which it doesn't, and which really distinguish it from what I understand to be the alternative model, that adaptation is driven by selection in partially immune hosts.

2) Related: I'm not sure I've understood what the point of the paper is. I thought from the Abstract that it was going to show that using a more realistic model of within-host evolution lets you get the right rate of global antigenic evolution, but after reading the paper I think that maybe that's not the case, and the authors are saying that there are also some interesting complications at the population level that would need to be taken into account. Putting it another way: I couldn't tell at times whether the point was that this was the first model of within-host evolution that included the known facts about immune dynamics, allowing it to say X about flu, or whether it was showing using influenza dynamics that a particular model of the immune system is correct.

3) As can be seen from Table 1, there is a lot going on in the model. The authors spend a lot of time giving intuition about the role of key features, but it would be nice if they could do even more, or even present a more stripped-down version of the model in the main text and save more details for the Appendix.

[Editors’ note: further revisions were suggested prior to acceptance, as described below.]

Thank you for submitting your article "Asynchrony between virus diversity and antibody selection limits influenza virus evolution" for consideration by *eLife*. Your article has been reviewed by Aleksandra Walczak as the Senior Editor, a Reviewing Editor, and three reviewers. The following individuals involved in review of your submission have agreed to reveal their identity: Daniel B Weissman (Reviewer #1); Christopher J R Illingworth (Reviewer #3).

The reviewers have discussed the reviews with one another and the Reviewing Editor has drafted this decision to help you prepare a revised submission.

The manuscript presents a theoretical study that bridges the global and the within-host evolution of the influenza virus. The key finding of the manuscript is that the asynchrony between within-host viral growth and the delayed antibody response following an infection can explain the pattern of antigenic evolution for influenza, where selection for new variants is strong at the population level but not within individuals. The reviewers agree that this work is conceptually important and provides a theory-driven hypothesis, which could in principle be tested with data. However, there are still a few points that we would like to see addressed in the manuscript.

Essential revisions:

1) The manuscript is presenting a thorough analysis of a model with qualitative comparison to data that establishes the *plausibility* of the model, rather than a quantitative comparison with alternative models by fitting data. The authors should frame the Abstract, Introduction and Discussion section to make this point more clear.

2) The reviewers agreed that the comparison to data is limited, and they have a suggestion that could strengthen the paper. However, the authors could chose to ignore this suggestion and only discuss it as a potential test of the model:

Figure 5 shows that there is a clear and large difference between the outcomes of the models involving or not involving an immediate recall response, which would seem to give scope for a quantitative comparison of antigenic evolution rates. As you know, the maps of antigenic change, first shown in Smith et al., 2004, are displayed within a quantitative metric space, where one square on the map corresponds to a specific change in titre. Is it possible to convert the antigenic change of Figure 5 to an approximate change in HI titre (e.g. per year) that might be compared with the antigenic maps of Smith et al. or an alternative measurement of the rate of antigenic change?

3) Your original submission made a distinction between two paths:

i)"Transmission selection": This just means that a host with a variant-dominated infection will tend to transmit more, because they can transmit to hosts who are immune to an old variant.

ii) "Inoculation selection": This is when there is competition within a diverse inoculum and immunity favors the new variant by clearing out the old type.

One can expects that under inoculation selection, the absolute rate at which new antigenic-mutant infections arise increases as immunity to the old type increases, whereas under transmission selection they arise at a constant absolute rate. It would be good if the authors could add a discussion about the consequences of these selection paths to the manuscript and also discuss whether global patterns are consistent with one or the other.

---

## [Author Response]

[Editors’ note: the authors resubmitted a revised version of the paper for consideration. What follows is the authors’ response to the first round of review.]

Reviewer #1:This paper addresses the observation that, while global influenza populations are strongly influenced by selection for antigenic novelty, within-host populations. It describes a model in which within-host selection is active, but temporally delayed, while selection based on the immune history of an individual acts at the time of transmission, during the establishment of an infection.1) I have two preliminary comments on the style in which this is written, as opposed to the content of the paper. Firstly, my experience reading this was that I found the main text difficult to follow until the point at which I had gained familiarity with the details of the model. For example, once I knew what \kappa_w, p_C, c, and k referred to I could understand the legend of Figure 3, though it took some time for me to get there. I suggest, if possible, rewriting the main text so as to be accessible in its own right to a casual reader.

We agree with the reviewer that the model was insufficiently introduced prior to the presentation of modeling results. We have now expanded the subsection “Model overview” to give a more complete summary, including definitions, with motivating examples, of relevant parameters. We have also made other modifications throughout the text to improve readability.

Secondly, I thought that the choice of data to be presented in figures was odd. The paper presents a model of viral evolution. As such, the interest (for me) was the extent to which the model replicates the observed behaviour of influenza virus in the real world. However, only Figure 1A and 1B describe any 'real world' data. Figure 4A is conceptual and useful, but the remainder of the figures exist in 'model space', showing how this or that parameter affects the behaviour of which part of the model. I'm not against model-to-model comparison (Figure 4E-H are useful though I'd personally keep only the top parts), but suggest removing anything from the main text which does not directly and clearly illustrate some key part of the model, or which does not show some direct comparison of the behaviours of different models as they relate directly to observations of the real-world system.

A limitation of our study is that many of the needed measurements to compare our model to data do not yet exist. No one has attempted to observe antigenic selection at the point of transmission because the assumption has generally been that antigenic selection happens during replication. As we now say explicitly in the Discussion section, one of our key aims with this study is to provide theoretical motivation for relevant experiments.

As such, our figures are necessarily model exploration for the most part. That said, we have revised Figure 1A-B to provide additional information, and revised the text to highlight the need for appropriate animal model experiments.

2) The claim is made that 'selection for new antigenic variants acts principally at the point of initial virus inoculation'. I have some questions and concerns around this claim.- To what extent has antigenic selection during the establishment of infection been experimentally documented? The primary reference given here is that of Wang, 2017. However, while Wang is clear that antibodies in the mucus can be influenza-specific (as opposed to being specific to another virus) it did not seem so clear that these antibodies are strain-specific in the sense of identifying viruses that differed by a single mutation.

Antigenic selection at the point of transmission is difficult to observe directly and difficult to distinguish from antigenic replication selection in the donor host, as Lumby et al., 2018 point out. We believe that much of the value in a theoretical study like this one lies in identifying potentially fruitful avenues for experiment that would not otherwise be obvious.

We have added the following text to the Discussion section:

“A key proposal of our study is that population-level antigenic selection and homotypic protection are mediated by antibody neutralization (most likely sIgA) at the point of transmission. […] While we cannot rule out a powerful immediate cellular response that was differentially evaded in the various subjects, we believe that our model, coupled with existing understanding of the timing of cellular responses and the speed of influenza replication, provides a more parsimonious explanation.”

- With this in the background I was not clear about the first of the claimed key results, that 'sIgA antibody neutralisation protects experienced hosts against reinfection'. It is certainly true is that a model incorporating selection during the transmission process (plus delayed within-host selection) is consistent with a range of observed data describing influenza infection and evolution. However, this is not the same as what is claimed. The hypothesis of the importance of sIgA antibody neutralisation is shown to be more consistent with the data than other models, but that does not imply that the hypothesis is true.

We agree with the reviewer’s comment. Our central claim is that one needs neutralization prior to virus exponential growth in order to simultaneously explain (a) homotypic reinfection, (b) the absence of observable antigenic mutants in reinfected hosts, and yet also (c) strong population level selection for antigenic novelty and apparent individual protection against reinfection.

We have revised the text to emphasize that mucosal sIgA neutralization is a biologically plausible mechanism that could produce neutralization prior to growth, but that our work is could be consistent with other mechanistic accounts (yet to be described) that produce a “strong-weak-(strong)” pattern of antigenic selection. That is, strong selection at the point of transmission, weak selection during virus exponential growth, and (potentially) strong selection again only late in infection.

We have modified the text in multiple places, including those noted in above, to make these points clear.

- Given the modelling that has been done, it would seem straightforward to evaluate this claim further. Within-host selection is delayed, but not entirely absent. As such, the expected frequency of a novel antigenic variant at the time of transmission to a new host will in the mean be higher than it would otherwise be under pure neutrality. It should be possible compare this change in the expected variant frequency to that produced by the transmission process (illustrated in Figure 4E). Averaging over stochastic effects, what proportion of the selection is due to within-host processes and what is due to selection during transmission? In the manuscript, it is claimed without further details that within-host evolution is 'dominated' by inoculation selection. What does this really mean?

We thank the reviewer for this excellent suggestion, which we have implemented using our analytical model. It forms the basis of a new figure, Figure 4.

We have revised the text to avoid the language of one form of selection “dominating” and now discuss the relative contributions of drift, replication selective effects, and inoculation selective effects to the survival of new variants at the point of transmission. In addition to adding Figure 4, we have added the following two passages to the Results section and Discussion section, respectively, in the main text:

“Onward transmission of new variants can be facilitated by natural selection—replication selection, inoculation selection or both. […] But again, it is difficult to estimate how much this synergy matters in practice without knowing more about that factors that govern homotypic reinfection and heterotypic reinfection.”

3) On the model of within-host viral load, this paper follows a commonly applied model of (roughly speaking) exponential growth followed by exponential decline. However, Pawelek et al. (2012) argue that a more bimodal pattern of behaviour is more realistic, a second peak arising from the re-emergence of susceptible cells that have been made refractory to infection by interferon. Would this increase the effect of within-host selection?

The double peak discussed in Pawelek et al., 2012 was observed in experimental infections of naive horses, and the observed second peak is much lower than the first. In human kinetics data (Hadjichrysanthou et al., 2016) and animal transmission experiments (Canini et al., 2020; Le Sage et al., 2020) it is common to see clearly single-peaked infections.

Furthermore, for realistic (incomplete) degrees of antibody binding escape, we expect both old and new antigenic variant population sizes to decline even due to antigenic limiting factors, though we naturally expect the new variant to decline more slowly.

The upshot is that clearance should be rapid following the mounting of the adaptive response in the experienced hosts where we expect selection to take place.

We have added an Appendix section (5.4) that discusses these points to our overview of parameter and model uncertainty (5).

Further, the work of Tsang et al. (2015) notes a proportion of transmission events occurring four or more days post the onset of symptoms; if within-host selection kicks in two days after infection it seems likely that within-host selection has a non-negligible effect even if events at transmission are the primary driver of selection.

It is first important to clarify that we chose to model an antibody response that is fully “on” at 48 hours. The aim was to be as harsh as possible to our hypothesis that asynchrony between diversity and selection could limit adaptation. The true timing and the rate of ramp-up is not well understood, but is likely to be slower, as we discuss in Appendix section 2.

Variation in observed transmission times might also reflect the timing of viral and resultant immune kinetics in those individuals. Empirical data suggests that inter-individual variation in such quantities is possible, and if individuals have slower initial growth and a later immune response, they may transmit later without having a longer period of antibody-mediated selection *_τ_*.

That is, Tsang et al., 2015 show that transmission might sometimes occur relatively late in infection but this is unlikely to be the case in an individual with an immune response that is capable of exerting efficient selection pressure.

In addition, the added discussion of synergy between replication and inoculation selection referenced in 1.5 provides a clearer quantification of this effect (see Figure 4): later transmissions will aid new variant survival roughly proportional to *δτ*, the product of the selection strength and the duration selection.

4) I don't believe that the model presented is a good one for infection in immunocompromised individuals. To the best of my knowledge the oscillations shown in Figure 3B and C have not been observed in clinical cases. My suspicion is that the model ignores some of the complexities of within-host infection in a way that gives a reasonable picture of a typical infection, but that falls short when it comes to longer-term infections. The point being made here is that selection for antigenic variants can be effective during longer infections; I would accept this point fairly readily, but the data of Figure 3 did not make me particularly more convinced.

We agree with the reviewer that the oscillations in the numbers of target cells and virions could be un-biological. The target cell replenishment was intended as a means of prolonging infection and modeling compromised innate immunity within our minimal model, since the minimal model treats the innate response implicitly, rolling it into target cell depletion.

In the revised manuscript, we have removed the original Figure 3 and replaced it with a verbal argument that references Figure 1G and our new general replicator equation figure (Figure 7). These make the conceptual/modeling point without the need for additional assumptions, and without potentially distracting and un-biological model output.

5) It is claimed that Figure 5 shows that proposed model of global influenza evolution better mimics empirical observations of the global population with regard to changes in the antigenic type of the virus. However, while it is clear from the figure that different models give different results, no indication of the observed behaviour of the global population is given in the Figure. The claim may be true. However, what is the observed distribution of changes in the global population to which Figure 5D provides a better match?

The key behavior we are attempting to explain with Figure 5 is that observed in Figure 1 and Figure 2 of Smith et al., 2004. We of course cannot reproduce these images in our main text, but we now explicitly reference those figures by number both in the caption to Figure 5 and in subsection “Inoculation selection produces realistically noisy between-host evolution” and subsection “Small-population-like evolution”.

While antigenic cluster transitions over time generally move directionally away from the root, within a cluster a more neutral pattern is observed, where substitutions can move in any direction in antigenic space. Bedford et al., 2012 attributed this pattern of seeming within-cluster neutrality to titration and estimation noise, but this is unlikely to be true for the Smith et al. map as each virus was assayed in duplicate or triplicate with high repeatability.

6) I was confused by Figure 7. Based on the methods I think that the term emergence describes the appearance of a variant in the population in such a way that it will not subsequently die out through genetic drift.

We see that this was a terminological issue, and precision is important when describing a cross-scale evolutionary process. We have edited the text to be clearer about mutation, observability, and stochastic extinction, both within-host and at the population level. We have substituted other phrases such as “first successful mutation” and “antigenic diversification”, where most appropriate.

Figure 7 then shows the distribution of emergence times under the case of an immediate recall response and a delayed recall response. My expectation would be that under an immediate recall response, positive selection would begin to apply to the variant earlier, making its emergence more likely to happen sooner, but the figure shows the opposite effect. I may have missed something in the explanation.

We have added a note to the figure caption to clarify this:

“Note that time of first successful mutation tends to be later with an immediate recall response than with a delayed recall response. This occurs because the cumulative number of viral replication events grows more slowly in time at the start of the infection because of the strong, immediate recall response.”

For our parameters, this total event effect swamps the stochastic extinction effect noted by the reviewer. But even the stochastic extinction effect may not necessarily favor new variant emergence in the presence of an early (versus delayed) antibody response. If the new variant achieves only partial immune escape escape, its stochastic extinction probability may be higher in the presence of antibodies than in their absence.

Antibodies controlling the old variant population may also keep the target cell population from being depleted, but this effect is marginal for our parameters, as new variants that survive stochastic extinction tend to be produced well before substantial target cell depletion occurs.

Interestingly, as we note in Appendix section 5.1.1, this means that the antibodies that maximally promote replication selection are the ones that begin when V(t)≈1µ : the mutant has been generated with near certainty, but we remain sufficiently early in the infection to give the new variant time to be selected before non-strain-specific limiting factors come into play.

7) Appendix 6: To the best of my understanding, the idea that large effect antigenic changes can occur in multiple backgrounds is not contradicted by the work of Koelle and Rasmussen. In that work a distinction is made between 'good' and 'bad' genetic backgrounds, characterised by the extent of mutational load in a sequence, however, there can be multiple 'good' backgrounds or at least, multiple backgrounds good enough for a variant to appear multiple times in the global population before fixation.

Koelle and Rasmussen do not argue that cluster transitions are only viable in a single genetic background. But they do argue that the rarity of good backgrounds is limiting for the virus, and is crucial to setting the pace of antigenic evolution and producing a spindly phylogeny with punctuated cluster transitions of intermediate size:

“Successful establishment can only occur when a large antigenic mutation arises in a good genetic background (Figure 2C), a jackpot combination”; “These analyses indicate that deleterious mutation loads should lower the rate of antigenic evolution (Figure 1E).”; “ Our analyses also indicate that deleterious mutation loads increase the average size of antigenic variants that establish in the long run (Figure 1D); observed patterns of punctuated antigenic evolution (Smith et al., 2004) may thus be better reproduced with a model that integrates sublethal deleterious mutations than one that ignores these mutation.”

That there are multiple cases of synchronous, polyphyletic antigenic cluster transitions casts doubt on these claims. If major antigenic changes appeared infrequently due to the rarity of jackpot cases of a good mutant in a good background, it would be very surprising to see two or three distinct and synchronous jackpots after years of none. So, while mutational load may play an important role in determining which lineages are fittest, synchronous polyphyly suggests that it is unlikely to set the clock of antigenic evolution. Rather, it suggests that accumulating population immunity may be crucial in setting the clock. Selection at the point of transmission begins to provide an explanation for how this could be the case.

We have now added similar text to the above to our discussion of deleterious load and other non-antigenic fitness to clarify that our point is about what limits the rate at which observable changes arise in the global population.

As another thought here, is the work of Strelkowa and Lassig, (2011) relevant as a proposed explanation for the pattern of global viral evolution, competition?

We thank the reviewer for pointing this out. Our work is consistent with the possibility of population-level clonal interference. As we note, neutralization of within-host adaptation is most likely necessary, but perhaps not sufficient, to explain why global scale influenza antigenic adaptation is not faster.

In particular, the polyphyletic emergence of antigenicity-altering substitutions suggests that when the time is right for an antigenic change, multiple lineages may compete to supply that change. Conversely, when population immunity has not risen to sufficiently high levels, the mutants may arise but lose in competition with adaptive non-antigenic mutants. That said, it is striking that typically when these polyphyletic emergences occur, they involve the same antigenicity altering substitutions across lineages.

We have added text to this effect to relevant parts of the Discussion section and Appendices, in particular subsection “Population immunity sets the clock of antigenic evolution” and Appendix section 7.3.

8) Appendix 7: The concept of selection acting both in the transmission process and during within-host replication is not entirely novel; Lumby et al. (2018) make this distinction in the context of influenza transmission and within-host growth (and further discuss the idea of a selective bottleneck). The idea that selection during the transmission process could play a role in the adaptation of global influenza populations has been raised very recently (Lumby et al., 2020) if not beforehand. This work therefore builds upon thinking about selection during viral transmission in the published literature.

We thank the reviewer for the Lumby et al., 2018 reference, which we have added in several places, most notably to help discuss the question of why it is so difficult to document antigenic selection at the point of transmission (see 1.3).

We have also added subsection “Relationship to influenza virus transmission prior bottleneck literature”, where we discuss prior literature on influenza bottlenecks, including whether selection might act there and if so how. In this section, we cite key prior work that hypothesized such effects, including Han et al., 2019; Lumby et al., 2018; Petrova and Russell, 2018.

Reviewer #2:Essential revisions:The paper is virtually all modeling except for a bit of deep sequencing data in Figure 1. Yet the authors strongly posit specific molecular mechanisms (such as secretory IgA) as driving the selection. While these mechanisms certainly seem plausible, the model is really just distinguishing between selection at transmission versus selection within-host after transmission. It would therefore be better to phrase more of the paper in those terms, and then perhaps in Introduction and Discussion section speculate about specific mechanisms that would drive different forms of selection.

We agree with the reviewer that it is not possible to establish conclusively the mechanism of selection at the point of transmission from our modeling work.

“We focus on sIgA neutralization at the point of transmission because it is a biologically plausible mechanism by which the temporal pattern of antigenic selection during infection could be strong (at the point of transmission), subsequently weak (during exponential growth), and then possibly strong (toward the end of infection). Other biological mechanisms that produce such a pattern would be consistent with our results. We have weakened the mechanistic language in the Abstract and Introduction (see quoted example below) and added more text of mechanisms to the Discussion to address this, and to clarify how this relates to existing literature on “selective bottlenecks” Antibody immunity at the point of transmission in previously infected or vaccinated individuals should reduce the initial probability of reinfection (Le Sage et al., 2020); secretory IgA antibodies on mucosal surfaces (sIgA) are likely to play a large role (Wang et al., 2017, see Appendix 2).”

Overall, I find these models informative and plausible. However, the comparison of the model predictions to real data is fairly limited and mostly qualitative.

This is true, but unfortunately unavoidable. As we now say in the Discussion section:

“This is a modeling study, and a mainly theoretical one. This is out of necessity. Quality experiments of the kind that are necessary to observe selection at the point of transmission directly and measure its strength have not been published, and we were unable to find any experimental measurements of within-host competition between known antigenic variants.”

Can the authors comment on the dynamics of infection in children? There is some evidence that infection times are longer in children than adults (Ng et al., 2016). Does this suggest that 'replication selection' could occur in children? If so, children make up a substantial proportion of the world population and would likely have a large impact on global transmission dynamics.

The reviewer raises an interesting point. Longer durations of replication in children could be a function of a weak specific antibody response. This would make it unlikely for them to exert selection pressure.

Indeed, there is some evidence that children require multiple infections before they begin to develop specific antibody immunity to influenza, which may make them less efficient inoculation selectors up to that point. We discuss this in detail in Appendix section 2, and have added a brief point to subsection “Importance of host heterogeneity”:

“Hosts with more focused immune responses—highly specific antibodies that neutralize old antigenic variant virions well and new antigenic variant virions poorly—could be especially good inoculation selectors and important sources of population-level antigenic diversity. Hosts who develop less specific memory responses, such as very young children (Neuzil et al., 2006), could be less important.”

How would the model change in cases where influenza infects the lower respiratory tract? This would change the size of the target cell population substantially.

While this does change the target cell population, it should still produce a period of exponential growth in the absence of antibody selection, followed by the imposition of non-antigenic limiting factors. As we state in in the main text, target cell depletion should most properly be considered as a proxy for a range of non-antigenic limiting factors, particularly the action of the innate immune system – many of which do indeed act by decreasing the virus’s cellular resources (e.g. through inflammation).

That said, truly chronic, prolonged influenza infections are excellent opportunities for antigenic diversification Xue et al., 2017, and we discuss this in light of our model.

We have added the following to subsection “Limitations and remaining uncertainties”:

“These factors limit the virus regardless of whether the infection remains confined to the upper respiratory tract or also invades the lower respiratory tract, thereby gaining access to additional target cells Koel et al., 2019. The key is that non-antigenic factors prevent persistent large virus populations.”

The authors should be clearer that when speaking about 'wild-type' and 'mutant' viruses, they refer to antigenicity and not the genotype. For example, in subsection “Model overview”, when the authors talk about 'within-host virus diversity' are they talking about the random accumulation of mutations or the random accumulation of antigenic variants? Also in the Results section, there would be a difference in the probability of a new variant infection and a new antigenic variant infection, so they should be clear about which they are referring to.

We thank the reviewer for this suggestion, which will substantially clarify the manuscript. We have revised language throughout the paper to refer to “old antigenic variants” and “new antigenic variants” and been explicit throughout that we are discussing antigenic phenotypes, particularly in HA.

How does a cellular immune response affect viral titer? Do T-Cells help to mount an immune response? I think the authors are using adaptive immunity and antibody-mediated immunity interchangeably when they aren't fully interchangeable. In particular, vaccination and infection might elicit different levels of T-cell immunity.

We agree with the reviewer that our manuscript most directly addresses antibody immunity and memory B-cell responses, as we are interested in the escape from antibody-binding observed at the population level in influenza virus HA proteins. We have reworded the text throughout to refer specifically to the antibody response rather than the adaptive response in general.

In the interest of tractability and encoding the mechanistic hypothesis, we made our model as simple as possible. But we agree that there are interesting implications for other forms of adaptive immunity. For instance, many forms of adaptive immunity may reduce severity without necessarily promoting antigenic evolution. As we say in the main text:

“The antibody response we introduce at 48 hours is qualitative—it is modeled simply as an increase in the virion decay rate for matching virions. This response could represent IgA, but other antigenicity-specific mode of virus control could also be subsumed under the increased virion decay rate, for instance IgG antibodies, or antibody dependent cellular cytotoxicity (ADCC). The key point we establish is that none of these mechanisms efficiently replication select because they all emerge once non-antigenic limiting factors have come into play. They speed clearance but should not substantially alter viral evolution. A corollary to this point is that there are many mechanisms—and interventions—that could reduce the severity of influenza infections without substantially speeding up antigenic evolution, including universal vaccines.”

How 'target-cell' limited is influenza really in practice? If it is, how would infections in immunocompromised individuals occur? I assume the rate of cellular replenishment is thesame, and that rate of target cell depletion would also be the same.

Following existing within-host influenza kinetics models (e.g. Baccam et al., 2006, Pawelek et al., 2012, Luo et al., 2012, Hadjichrysanthou et al., 2016), we interpret target cell depletion as a combination of virus-mediated cell death and innate immune responses that act to limit cell availability through inflammation, cell death, and so on. We describe this explicitly in subsection “Limitations and remaining uncertainties”. In immune-compromised individuals the waning of these effects (which tend to be short-lived) allows for a consistent supply of infectible cells. That said, we do not believe the cycling behavior of replenishment and depletion showed in the example immune-compromised kinetics from Figure 3 is biological; we have modified the figure to avoid giving the impression that it is (see response to reviewer 1).

Reviewer #3:This is an interesting manuscript that uses mathematical modeling to argue that the timing of within-host selection is crucial to understanding antigenic evolution in influenza. In particular, it argues that selection by antibodies on mucosal surfaces during initial infection is the key driver of adaptation. This seems important, although I had a hard time following the argument in places.

We are pleased that the reviewer finds the work novel and important. We have worked to clarify the structure of the argument throughout, most notably by:

1) Expanding the initial model description, so that the reader is equipped with terminology and notation prior to encountering model results.

2) Adding subsection headings to the Results, with the aim of clarifying the structure of the argument.

Essential revisions:1) I would appreciate a clear, concise statement of which aspects of observed flu dynamics (e.g., reinfected hosts without escape mutants, global patterns of evolution, etc.) the model hits and which it doesn't, and which really distinguish it from what I understand to be the alternative model, that adaptation is driven by selection in partially immune hosts.

We have addressed this by moving some of the Appendix text out of the Appendix and making them the beginning of the main text Discussion section:

“Any explanation of influenza virus antigenic evolution—and why it is not even faster—must explain why population level antigenic selection is strong but within-host antigenic evolution is difficult to observe.”

“We hypothesized that antibodies present in the mucosa at the time of virus exposure can effectively block transmission, but have only a small effect on viral replication once cells become productively infected. Antigenic selection after successful infection therefore begins with the mounting of a recall response 48–72 hours post infection. In this case, selection pressure can be strong at the point of transmission, but subsequently weak until after 48 hours. This mechanistic paradigm reconciles strong but not perfect sterilizing homotypic immunity with rare observations of mutants in successfully reinfected experienced hosts.”

2) Related: I'm not sure I've understood what the point of the paper is. I thought from the Abstract that it was going to show that using a more realistic model of within-host evolution lets you get the right rate of global antigenic evolution, but after reading the paper I think that maybe that's not the case, and the authors are saying that there are also some interesting complications at the population level that would need to be taken into account. Putting it another way: I couldn't tell at times whether the point was that this was the first model of within-host evolution that included the known facts about immune dynamics, allowing it to say X about flu, or whether it was showing using influenza dynamics that a particular model of the immune system is correct.

We agree with the reviewer that the previous version of the manuscript failed to make our central point clear.

The point of our paper, first and foremost, is that influenza viruses possess both of the fundamental materials of adaptation within a single reinfected host: diversity and selection pressure. We propose that antigenic adaptation nonetheless fails to occur because of the temporal asynchrony between diversity and selection pressure.

This central finding then has implications for population level adaptation. Notably, existing models of influenza within-host evolution and the ready availability of large-effect antigenic substitutions would seem to imply that influenza adaptation should be rapid within reinfected experienced hosts, and that this could lead to rapid antigenic diversification at the population level. We are then able to explain why population-level diversification may be a slower process, with new variants only seen after a new antigenic variant has circulated long enough to generate substantial population immunity.

We have changed wording throughout the text to highlight that the central findings concern why within host adaptation is not more rapid, and that the implications for population level patterns, though of substantial interest, are just that – implications.

3) As can be seen from Table 1, there is a lot going on in the model. The authors spend a lot of time giving intuition about the role of key features, but it would be nice if they could do even more, or even present a more stripped-down version of the model in the main text and save more details for the Appendix.

The other two reviewers shared this concern, and we agree that it was an issue. As we explain in our response to reviewer 1, we have now added a more detailed model description, with examples, prior to the Results section.

[Editors’ note: what follows is the authors’ response to the second round of review.]

Essential revisions:1) The manuscript is presenting a thorough analysis of a model with qualitative comparison to data that establishes the plausibility of the model, rather than a quantitative comparison with alternative models by fitting data. The authors should frame the Abstract, Introduction and Discussion section to make this point more clear.

We have now revised the Abstract and Introduction to make it clear that our study is theoretical, and aimed at establishing the plausibility of a mechanism that could then be investigated empirically. In particular:

We have changed a key sentence in the Abstract to use more conditional language and emphasize that this is a modeling exercise: “Using a mathematical model, we show that the temporal asynchrony between within-host virus exponential growth and antibody-mediated selection could limit within-host antigenic evolution”.

Similarly, we have changed the summary of our results in the Abstract to: “Our results provide a theoretical explanation for how virus antigenic evolution can be highly selective at the global level but nearly neutral within host”.

We have also added the following final paragraph to the end of the Introduction:

“Our modeling results suggest a plausible mechanism that can explain otherwise poorly reconciled empirical patterns, and should motivate further experimental investigation of the mechanisms of immune protection and natural selection on influenza virus antigenic phenotypes at the point of transmission.”

2) The reviewers agreed that the comparison to data is limited, and they have a suggestion that could strengthen the paper. However, the authors could chose to ignore this suggestion and only discuss it as a potential test of the model:Figure 5 shows that there is a clear and large difference between the outcomes of the models involving or not involving an immediate recall response, which would seem to give scope for a quantitative comparison of antigenic evolution rates. As you know, the maps of antigenic change, first shown in Smith et al., 2004, are displayed within a quantitative metric space, where one square on the map corresponds to a specific change in titre. Is it possible to convert the antigenic change of Figure 5 to an approximate change in HI titre (e.g. per year) that might be compared with the antigenic maps of Smith et al. or an alternative measurement of the rate of antigenic change?

We agree with the reviewers that this would be an interesting comparison. We feel, however, that the single-chain to first-fixation model presented in Figure 5 is more appropriate for predicting the distribution of observed antigenic fixations rather than the yearly rate of antigenic change in the dominant variant, since the latter depends upon numbers of simultaneous chains, accumulating population immunity, and clonal interference amongst emerged types. In our opinion, this would require a principled population-level model.

Instead, we have attempted to highlight more clearly where the model does speak to Smith et al. style data: in predicting within-cluster backward fixations and drift alongside rarer forward jumps. We have added the following paragraph to the discussion of Figure 5:

“In particular, we note that whereas an immediate recall response would predict strong near-constant directed evolution of virus antigenic phenotypes away from existing immunity (Figure 5B,C), a realistically-timed recall response predicts that small-effect, drift-like, antigenic substitutions will be observed. […] Exact rates of antigenic evolution will depend upon how these emergence processes intersect with population-level epidemic dynamics and competition among variants.”

3) Your original submission made a distinction between two paths:i)"Transmission selection": This just means that a host with a variant-dominated infection will tend to transmit more, because they can transmit to hosts who are immune to an old variant.ii) "Inoculation selection": This is when there is competition within a diverse inoculum and immunity favors the new variant by clearing out the old type.One can expects that under inoculation selection, the absolute rate at which new antigenic-mutant infections arise increases as immunity to the old type increases, whereas under transmission selection they arise at a constant absolute rate. It would be good if the authors could add a discussion about the consequences of these selection paths to the manuscript and also discuss whether global patterns are consistent with one or the other.

We now call “transmission selection” simply “population-level selection”, as we think this increases clarity and avoids adding unnecessary new terminology. We retain the discussion of it in subsection “Tight bottlenecks lead to loss of generated diversity and mean new variants reach consensus through founder effects”: “Neutralization at the point of transmission thus not only gives new antigenic variant infections their transmission advantage (population-level selection) but may also increase the rate at which these new antigenic variant infections arise (inoculation selection).”

We agree with the reviewer about this empirical prediction, which we discussed in subsection “Population immunity sets the clock of antigenic evolution”. We have revised that section to highlight this point more clearly, including adding a reference to polyphyletic emergence events of new antigenic variants:

“If immunity to the old variant only gives new variants a population-level transmission advantage (population-level selection), we anticipate a constant rate of population-level antigenic diversification, with selective sweeps once a new variant has a sufficient population-level advantage over the old variant. […] That said, alternative explanations are possible, such as reduced rates of stochastic loss of new variants at higher scales (e.g. the epidemic scale) with increased population immunity.”